# ON THE DYNAMICS OF LEARNING LINEAR FUNCTIONS WITH NEURAL NETWORKS

## ABSTRACT

This paper studies the gradient descent training dynamics of fitting a one-hidden-layer network with multi-dimensional outputs to linear target functions. That is, we focus on a realizable model where the inputs are drawn i.i.d. from a Gaussian distribution and the labels are generated according to a planted linear model with multiple outputs. This framework serves as a good model for a variety of interesting problems including end-to-end training in inverse problems and various autoencoder models in machine learning. Despite the seemingly simple formulation, understanding training dynamics is a challenging unresolved problem. This is in part due to the fact that the training landscape contains multiple non-strict saddle points and it is completely unclear why gradient descent from random initialization is able to escape such bad stationary points. In this work, we develop the first comprehensive analysis of the gradient descent dynamics for learning linear target functions with ReLU networks. We provide a comprehensive characterization of the optimization landscape. Furthermore, we show that gradient descent with moderately small random initialization converges to a global minimizer at a linear rate with an order-wise optimal sample complexity. To rigorously show that GD avoids non-strict saddle points, we develop intricate techniques to decompose the loss and control the GD trajectory, which may have broader implications for the analysis of non-convex optimization problems involving non-strict saddles. We corroborate our theoretical results with extensive experiments with various configurations.

## 1 INTRODUCTION

### 1.1 MOTIVATION

End-to-end training of neural networks (NNs) via Gradient Descent (GD) has recently achieved remarkable success on many tasks. Of particular interest, these models have been adopted to solve inverse problems by taking the measurements as input and mapping them directly to the desired signal with successful scientific applications in computer vision (Ledig et al., 2017; Wang et al., 2018), MRI reconstruction (Sriram et al., 2020; Fabian et al., 2022), sparse-view computed tomography (CT) (Jin et al., 2017b), and phase retrieval (Hand et al., 2018). These models not only fit the training data but also appear to capture useful features and nuanced priors that enable them to generalize to unseen test examples. Despite this empirical success, the reasons behind the success of NNs for end-to-end training and how they can extract useful features from data remain unclear.

Perhaps the most classical form of end-to-end training is those arising in autoencoder type problems where the goal is to teach a neural network to learn a linear mapping (e.g. identity for autoencoders). Surprisingly, the dynamics of training of such a model is not well understood for nonlinear models. For linear networks, a classical result by Baldi and Hornik (1989) provided a complete characterization, showing how gradient descent recovers the principal components of the data. In contrast, understanding the dynamics of non-linear encoders has remained an open and challenging problem even for simple target functions. In this paper we aim to take a step towards a systematic understanding of the training dynamics of such problems by addressing the following question:

> *How do the dynamics of training ReLU neural networks with gradient descent starting from random initialization facilitate learning simple priors and structures such as linear target functions?*

Despite significant recent progress in understanding neural networks (especially shallow networks) (Chizat et al., 2019; Soltanolkotabi et al., 2018; Jacot et al., 2018; Du et al., 2018; Ongie et al., 2019), many aspects of the dynamics of GD and how it facilitates learning remain mysterious even in seemingly simple settings. A particularly simple one involves learning linear target functions via GD, that is, teaching a one-hidden-layer network to mimic the output of a simple linear model. Surprisingly, understanding the dynamics of GD in this simple setting has remained elusive. Although there are many results on learning specific target functions such as ReLUs (Xu and Du, 2023; Soltanolkotabi, 2017) and polynomials (Damian et al., 2022), these results typically exclude linear function classes. In fact, many of the existing papers use a pre-processing step or alter the early optimization trajectory to avoid complications arising from the dynamics of learning linear functions (Damian et al., 2022). This is in part due to the fact that the optimization landscape of learning linear target functions contains multiple non-strict saddle points (i.e. where the gradient vanishes and the Hessian is PSD but has a 0 eigenvalue) requiring a subtle trajectory analysis to ensure GD avoid these bad points (See Section 1.3 for further details). We note that despite the simple formulation, quite a few interesting scenarios, including autoencoder training dynamics, are captured in this framework.

Our main contributions are as follows:

- To gain insight into the inner working of nonlinear autoencoders, we focus our attention on learning linear target functions using one-hidden-layer Neural Networks (NNs) via Gradient Descent (GD). We empirically observe an interesting pattern in the weights of the neural network with exact parametrization (when the number of hidden units is exactly twice the number of target directions). We find that GD iterations converge to a solution where hidden units cluster into *pairs*: incoming and outcoming weights from these pairs are the negative of each other (Figure 3).

- Fixing the outer layer of the NN according to the said sign pattern, with exact parameterization, we develop theory for running GD on the NN with moderately small initialization, demonstrating exact convergence to the ground truth at a linear rate and with an optimal sample complexity that scales linearly in the number of parameters. That is, we show that the inner weights of the NN recover the target directions *exactly*.

- As detailed further in Section 1.3 the training landscape studied in this paper contains multiple non-strict saddles. To prove that the trajectory of GD from moderately small random initialization avoids these bad minima we develop new techniques to control the GD trajectory which we combine with intricate uniform concentration bounds. We believe our novel proof techniques may have broader implications for the analysis of non-convex optimization problems involving non-strict saddles.

- We further corroborate our results with various experimental investigations.

## 1.2 PROBLEM FORMULATION

We first state the general family of problems of interest in this paper.

**Data Model.** We assume there are $n$ pairs of training data consisting of input features $\boldsymbol{x}_i \in \mathbb{R}^d$ and corresponding targets $\boldsymbol{y}_i \in \mathbb{R}^r$. As mentioned before, we consider the class of linear models where the relationship between $\boldsymbol{x}_i$ and $\boldsymbol{y}_i$ is given by the equation: $\boldsymbol{y}_i = \boldsymbol{A}\boldsymbol{x}_i$ where $\boldsymbol{A} \in \mathbb{R}^{r \times d}$ is the labeling matrix. Conceptually, $\boldsymbol{A}$ contains $r$ target *directions* $(\boldsymbol{a}_1, \cdots, \boldsymbol{a}_r)$ that our predictor should *learn*. In the special case when $r = 1$ (there is a *single* direction to be learned), we use $\boldsymbol{a}^T$ instead of $\boldsymbol{A}$ to emphasize this single direction $\boldsymbol{a}$. For our theoretical analysis we assume the data points $\boldsymbol{x}_i$ are drawn i.i.d. according to a standard normal distribution $\mathcal{N}(\boldsymbol{0}, \boldsymbol{I}_d)$.

**Network Model.** We consider one-hidden-layer neural networks of the form $\hat{\boldsymbol{y}} = f(\boldsymbol{\theta}, \boldsymbol{x}) = f(\boldsymbol{V}, \boldsymbol{W}, \boldsymbol{x}) = \boldsymbol{V}^T \phi(\boldsymbol{W}\boldsymbol{x})$ as our predictor. Here $k$ denotes the number of hidden-units, $\boldsymbol{V} \in \mathbb{R}^{k \times r}$ is the outer layer of the neural network, $\boldsymbol{W} \in \mathbb{R}^{k \times d}$ is the inner layer of the neural network, and $\phi(\boldsymbol{z})$ is the activation function. We refer to individual rows of $\boldsymbol{V}/\boldsymbol{W}$ as $\boldsymbol{v}_i/\boldsymbol{w}_i$ respectively, In this paper,

we specifically consider neural networks with ReLU activation functions i.e. $\phi(z) = \text{ReLU}(z) = \max(0, z)$, where $\max$ is applied to the input vector $z$ element-wise.

**Training Loss.** We minimize the squared loss between the target and the prediction

$$\mathcal{L}(\boldsymbol{\theta}) = \frac{1}{2n} \sum_{i=1}^{n} \|\hat{\boldsymbol{y}}_i - \boldsymbol{y}_i\|^2 = \frac{1}{2n} \sum_{i=1}^{n} \left\| \boldsymbol{V}^T \phi(\boldsymbol{W}\boldsymbol{x}) - \boldsymbol{A}\boldsymbol{x} \right\|^2 \tag{1}$$

using gradient descent. For part of our theoretical analysis of GD, we also consider the population loss (i.e. infinite data asymptotics as $n \to \infty$) with $\boldsymbol{x}$ drawn randomly from an isotropic Gaussian distribution $\boldsymbol{x} \sim \mathcal{N}(\boldsymbol{0}, \boldsymbol{I_d})$. Concretely, the population loss is given by

$$\mathcal{L}(\boldsymbol{\theta}) = \frac{1}{2} \mathbb{E}_{\boldsymbol{x}} \left[ \left\| \sum_{i=1}^{k} \boldsymbol{v}_i \phi\left(\boldsymbol{w}_i^T \boldsymbol{x}\right) - \boldsymbol{A}\boldsymbol{x} \right\|^2 \right]. \tag{2}$$

### 1.3 WHY IS LEARNING LINEAR FUNCTIONS WITH ReLU NETWORKS CHALLENGING?

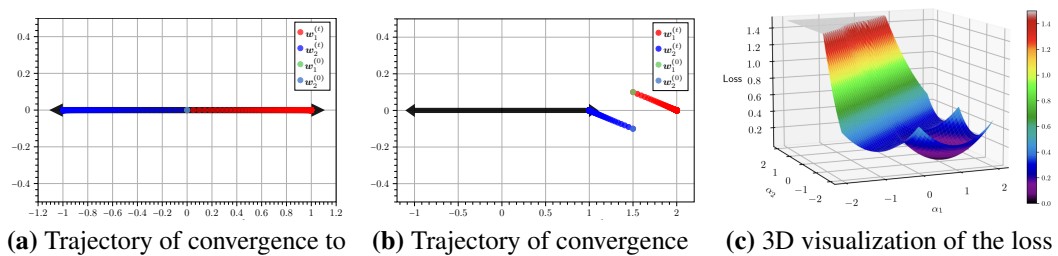

**(a)** Trajectory of convergence to the global optimum.

**(b)** Trajectory of convergence to one of the non-strict saddle points.

**(c)** 3D visualization of the loss landscape.

Figure 1: **Characterization of the Loss landscape.** We run gradient descent updates on the population loss. Here, the fitted model is a one-hidden-layer ReLU network with two hidden units and the outer layer is fixed to $\boldsymbol{v} = [1, -1]^T$. We use two different choices of initializations for $\boldsymbol{w}_1^{(0)}$ and $\boldsymbol{w}_2^{(0)}$ in parts (a) and (b). Note that in both figures black arrows indicate the $\pm \boldsymbol{a}$ direction and a randomly selected orthogonal direction to $\boldsymbol{a}$ is shown for the y-axis in order to visualize the neurons in 2D. In part (a) on the left, we initialize the weights small, i.e. near the origin, and observe that the weights converge to the global optimum. However, when we initialize the weights near $1.5\boldsymbol{a}$ as depicted in part (b) on the right, the weights converge to $\boldsymbol{a}$ and $2\boldsymbol{a}$. This corresponds to one of the non-strict saddle points of the population loss (2). To corroborate this further, in part (c) we visualize the loss landscape when $\boldsymbol{w}_1 = \alpha_1 \boldsymbol{a}$ and $\boldsymbol{w}_2 = \alpha_2 \boldsymbol{a}$.

Given the simple nature of the target function it is natural to wonder about what makes GD analysis in this setting challenging. The main challenge arises from the fact that the loss landscape of the problem with ReLU NNs has many non-strict saddles–in fact infinitely many! For instance, when the output is one dimensional, if we fix the outer layer to be $\boldsymbol{v} = [v_1, -v_2]^T$ with $v_1, v_2 > 0$, and the target function is given by $\boldsymbol{x} \mapsto \boldsymbol{a}^T \boldsymbol{x}$ it is easy to see that any $\boldsymbol{w}_1$ and $\boldsymbol{w}_2$ of the form $\boldsymbol{w}_1 = \frac{(c+1)\boldsymbol{a}}{v_1}$ and $\boldsymbol{w}_2 = \frac{c\boldsymbol{a}}{v_2}$ for any $c > 0$ or $c < -1$ is a non-strict saddle (See Appendix C for comprehensive characterization of the loss landscape). In Figure 1 (c) we visualize this by setting $v_1 = v_2 = 1$ and drawing the loss as a 3D plot when we restrict student neurons to be $\boldsymbol{w}_1 = \alpha_1 \boldsymbol{a}$ and $\boldsymbol{w}_2 = \alpha_2 \boldsymbol{a}$. The gradient vanishes around the $\alpha_1 - \alpha_2 = 1$ *valley*, but the loss is greater than 0.

As a concrete example, in Figure 1 we show that the initialization of the network directly influences whether GD will converge to the global optimum or a non-strict saddle. This experiment gives us a hint that as long as $\|\boldsymbol{w}_1 + \boldsymbol{w}_2\|$ ($\|v_1 \boldsymbol{w}_1 + v_2 \boldsymbol{w}_2\|$ for general $v_1, v_2$) is sufficiently small (which is satisfied for small initialization), we are far away from bad stationary points. This observation will play a crucial role in our analysis.

## 2 Theoretical Results in the Population Setting

We begin by stating our main result when the output is one dimensional i.e. $r = 1$ and defer the general case to the appendix.

**Theorem 1** *Suppose the feature vectors are distributed i.i.d. according to a Gaussian distribution $\boldsymbol{x} \sim \mathcal{N}(\boldsymbol{0}, \boldsymbol{I}_d)$. We assume the corresponding output are generated according to a linear target function of the form $\boldsymbol{y} = \boldsymbol{a}^T \boldsymbol{x}$ where $\boldsymbol{a} \in \mathbb{R}^d$ is an arbitrary weight vector. To learn this linear function we fit a one hidden layer ReLU network with two hidden nodes of the form*

$$\boldsymbol{x} \mapsto \boldsymbol{v}^T ReLU(\boldsymbol{W}\boldsymbol{x}) = v_1 ReLU(\boldsymbol{w}_1^T \boldsymbol{x}) - v_2 ReLU(\boldsymbol{w}_2^T \boldsymbol{x}).$$

*Here, we fix $\boldsymbol{v} = [v_1, -v_2]^T$ with $v_1, v_2 > 0$ and define $\boldsymbol{W} = [\boldsymbol{w}_1, \boldsymbol{w}_2]^T \in \mathbb{R}^{2 \times d}$. Consider the population loss*

$$\mathcal{L}(\boldsymbol{W}) = \frac{1}{2} \mathrm{E}_{\boldsymbol{x}} \left[ \left( \boldsymbol{v}^T ReLU(\boldsymbol{W}\boldsymbol{x}) - \boldsymbol{a}^T \boldsymbol{x} \right)^2 \right].$$

*To fit this model we run gradient updates of the form $\boldsymbol{W}^{(\tau+1)} = \boldsymbol{W}^{(\tau)} - \mu \, diag \, (\boldsymbol{v})^{-2} \, \nabla \mathcal{L}(\boldsymbol{W}^{(\tau)})$, starting from an initial estimate $\boldsymbol{W}^{(0)} = \begin{bmatrix} \boldsymbol{w}_1^{(0)} & \boldsymbol{w}_2^{(0)} \end{bmatrix}^T$ with step size obeying $\mu \leq c_1$.*

*Then, as long as the initialization obeys $\left\| v_1 \boldsymbol{w}_1^{(0)} + v_2 \boldsymbol{w}_2^{(0)} \right\| \leq \frac{1}{2} \|\boldsymbol{a}\|$, we have*

$$\left\| \boldsymbol{W}^{(\tau)} - \boldsymbol{W}^* \right\|_F^2 \leq \frac{\max(v_1^2, v_2^2)}{\min(v_1^2, v_2^2)} (1 - c_2 \mu)^\tau \left\| \boldsymbol{W}^{(0)} - \boldsymbol{W}^* \right\|_F^2$$

*for all iterations $\tau$. Here, $\boldsymbol{W}^* = \left[ \frac{\boldsymbol{a}}{v_1}, -\frac{\boldsymbol{a}}{v_2} \right]^T$ and all $c_j$'s are fixed numerical constants.*

This result shows that one can indeed use GD to train a one-hidden layer network with two hidden nodes to learn a linear function. We note that any linear function of the form $\boldsymbol{a}^T \boldsymbol{x}$ can also be written as a difference of two ReLUs: $v_1 \mathrm{ReLU} \left( \frac{1}{v_1} \boldsymbol{a}^T \boldsymbol{x} \right) - v_2 \mathrm{ReLU} \left( \frac{-1}{v_2} \boldsymbol{a}^T \boldsymbol{x} \right)$ for any $v_1, v_2 > 0$, so that two hidden nodes are necessary. We indeed show directly that the GD updates result in directional convergence: $\boldsymbol{w}_1^{(\infty)} = \frac{\boldsymbol{a}}{v_1}$ and $\boldsymbol{w}_2^{(\infty)} = -\frac{\boldsymbol{a}}{v_2}$.

Stated differently, with exact parametrization, GD indeed finds the underlying structure in the data (we verify this experimentally in Section 4.1). It is also worth noting that the dependence on the initialization scale is moderate and is only through ensuring that $\|v_1 \boldsymbol{w}_1 + v_2 \boldsymbol{w}_2\|$ is not *too large* at initialization. Finally, the training is rather fast enjoying a geometric (a.k.a. linear) rate of convergence. It is worth noting that this result is based on running GD with a *scaled* step size due to the $diag \, (\boldsymbol{v})^{-2}$ term. While this scaling is absent in standard GD updates, it significantly accelerates convergence which we demonstrate empirically in Section 4.3.

As discussed previously in Section 1.3 the optimization landscape in this problem is rather complex involving multiple non-strict saddles. Nevertheless, the above theorem demonstrates geometric convergence to the global optimum. As we explain in the proofs this is possible in part an interesting control of the trajectory of the iterates where we show that throughout the training dynamics $\left\| v_1 \boldsymbol{w}_1^{(\tau)} + v_2 \boldsymbol{w}_2^{(\tau)} \right\|$, continuously decrease. This facilitates a more refined control of the trajectory of GD, ensuring that GD can in fact avoid the bad stationary points.

Finally, we note that when running GD on both layers from small random initialization the outer weight will have opposite sign with roughly the same absolute value ($v_1 \approx v_2$) (see Section 4). This holds if the output layer weights have opposite sign at initialization. Perhaps to be expected if the outer weights are initialized with the same sign the model gets stuck at a local optimum.

## 3 Theoretical Results in the Empirical Setting

In the previous section, we stated our main results for the population setting. Now, we focus on the more practical empirical setting when the output is one dimensional i.e. $r = 1$.

**Theorem 2** *Suppose we have $n$ feature vectors $\{\boldsymbol{x}_1, \cdots, \boldsymbol{x}_n\}$ that are sampled i.i.d. according to a Gaussian distribution $\boldsymbol{x}_i \sim \mathcal{N}(\boldsymbol{0}, \boldsymbol{I}_d)$. We assume the corresponding output are generated according to a linear target function of the form $\boldsymbol{y}_i = \boldsymbol{a}^T \boldsymbol{x}_i$ where $\boldsymbol{a} \in \mathbb{R}^d$ is an arbitrary weight vector. To learn this linear function we fit a one hidden layer ReLU network with two hidden nodes of the form*

$$\boldsymbol{x} \mapsto \boldsymbol{v}^T ReLU(\boldsymbol{W}\boldsymbol{x}) = v_1 ReLU(\boldsymbol{w}_1^T \boldsymbol{x}) - v_2 ReLU(\boldsymbol{w}_2^T \boldsymbol{x}).$$

*Here, we fix the outer weights $\boldsymbol{v} = [v_1, -v_2]^T$ for $v_1, v_2 > 0$ and optimize the loss over the inner weights $\boldsymbol{W} = [\boldsymbol{w}_1, \boldsymbol{w}_2]^T \in \mathbb{R}^{2 \times d}$ on the empirical loss*

$$\widehat{\mathcal{L}}(\boldsymbol{W}) = \frac{1}{2n} \sum_{i=1}^n \left( \boldsymbol{v}^T ReLU\left(\boldsymbol{W}\boldsymbol{x}_i\right) - \boldsymbol{a}^T \boldsymbol{x}_i \right)^2.$$

*To fit this model we run gradient updates of the form $\boldsymbol{W}^{(\tau+1)} = \boldsymbol{W}^{(\tau)} - \mu_\tau \, diag\left(\boldsymbol{v}\right)^{-2} \nabla \mathcal{L}(\boldsymbol{W}^{(\tau)})$, with step size obeying $\mu_1 = 2$, $\mu_\tau = \mu \le c_5$, $\forall \tau \ge 2$. Furthermore, we assume a sufficiently small random initial estimate $\boldsymbol{W}^{(0)} = \begin{bmatrix} \boldsymbol{w}_1^{(0)} & \boldsymbol{w}_2^{(0)} \end{bmatrix}^T$ of the form $\boldsymbol{w}_1^{(0)} \sim \mathcal{N}\left(\boldsymbol{0}, \sigma_1^2 \boldsymbol{I}_d\right)$ and $\boldsymbol{w}_2^{(0)} \sim \mathcal{N}\left(\boldsymbol{0}, \sigma_2^2 \boldsymbol{I}_d\right)$ with the standard deviations obeying $\sqrt{v_1^2 \sigma_1^2 + v_2^2 \sigma_2^2} \sqrt{d} \le c_6 \|\boldsymbol{a}\|$. Then as long as the number of training samples satisfy $n \ge c_7 d$, we have*

$$\left\| \boldsymbol{W}^{(\tau)} - \boldsymbol{W}^* \right\|_F^2 \le \frac{\max(v_1^2, v_2^2)}{\min(v_1^2, v_2^2)} \left(1 - c_8 \mu\right)^\tau \left\| \boldsymbol{W}^{(0)} - \boldsymbol{W}^* \right\|_F^2$$

*with probability at least $1 - Ce^{-cd}$. Here, $\boldsymbol{W}^* = \left[\frac{\boldsymbol{a}}{v_1}, -\frac{\boldsymbol{a}}{v_2}\right]^T$ and all $c_j$'s are fixed numerical constants independent of any problem dimensions.*

This result demonstrates that gradient descent can indeed be used to train a one-hidden-layer network with two hidden nodes on a finite number of training samples to learn a linear function. Similar to the population setting, we show that student neurons $\boldsymbol{w}_1$ and $\boldsymbol{w}_2$ directionally converge in direction to the ground truth with high probability. Notably, the convergence is fast, exhibiting a geometric rate. Moreover, the sample complexity is optimal, scaling linearly with the problem dimension $d$. Finally, we note that the theorem above allows sample reuse across all iterations requiring the development and use of intricate uniform concentration inequalities in the proof.

We also highlight that the initialization radius required in the empirical setting, given by $\sqrt{v_1^2 \sigma_1^2 + v_2^2 \sigma_2^2} \sqrt{d} \le c_6 \|\boldsymbol{a}\|$, directly parallels the condition in the population setting, where $\left\| v_1 \boldsymbol{w}_1^{(0)} + v_2 \boldsymbol{w}_2^{(0)} \right\| \le \frac{1}{2} \|\boldsymbol{a}\|$. However, our analysis for the empirical case relies on random initialization, whereas in the population setting, any initialization in the specified ball is sufficient.

## 4 EXPERIMENTS

In this section we show experimental results in various output dimension ($r = 1$ vs. $r > 1$), and initialization scale (small vs large). We use PyTorch for experiments and unless mentioned otherwise, network weights are initialized with Xavier Normal initialization (for a matrix $\boldsymbol{W} \in \mathbb{R}^{r \times d}$, $\boldsymbol{W}_{ij} \sim \mathcal{N}\left(0, \frac{2}{r+d}\right)$). For visualization purposes, we set $v_1 = v_2 = 1$ (in fact the proof for arbitrary $v_1, v_2$ setting can be reduced to $v_1 = v_2 = 1$ which we discuss in detail in Appendix D). In order to change the initialization scale, we multiply the default initialization with a positive scalar $\alpha$. For *small* initialization experiments, we use $\alpha = 10^{-8}$, otherwise it is set to $\alpha = 1$. We set $d = 100$ and $\mu = 0.1$. All experiments are run on a server with an Intel Xeon Gold 5220R CPU. We would like to stress that even though the visualizations in this paper are based on a single trial, we ran these experiments for different random seeds and the behavior of the visualizations did not change.

### 4.1 LEARNING LINEAR TARGETS (SINGLE OUTPUT: $r = 1$)

In experiments w.l.o.g. we choose $\boldsymbol{a} = \boldsymbol{e}_1$ where $\boldsymbol{e}_1$ is the first standard basis in $\mathbb{R}^d$. This does not effect the results due to the rotational symmetry of isotropic Gaussian distribution of which $\boldsymbol{x}$ are drawn from. Note that this implies $\|\boldsymbol{a}\| = 1$ in our experiments. Finally, in this section we focus

our experiments on the population loss. Similar results continue to hold in the empirical case with moderate sample sizes i.e. when $n \geq crd$ with $c$ a sufficiently large constant.

When the model is exactly parameterized with two hidden nodes ($k = 2$), and outer layer is fixed as $\pm 1$ we show that the population loss decreases at a linear rate (Figure 5) and weights of the inner layer align themselves with $\pm a$ (Figure 2 (a)). If $v$ is also initialized randomly, we empirically see that the model cannot converge to the global optima consistently. When it does, $v^{(\infty)}$ indeed becomes $\pm 1$ and $w_1^{(\infty)}$ and $w_2^{(\infty)}$ recover $\pm a$ exactly. For the remaining time, the GD iterates converge to one of the many local optima of this problem similar to the depiction in Figure 1 (part b). We further observe that iterates get stuck only when $v_1^{(0)}$ and $v_2^{(0)}$ both have the same signs which happens with probability $\frac{1}{2}$.

When $k > 2$, the probability of *all* $v_i$'s having the same sign decreases rapidly. Therefore, iterates typically converge to the global optima. However, in this case global minima is not unique anymore. To demonstrate this, consider the case where there are four hidden units ($k = 4$) instead of two. We fix half of $v$ as $+1$ and remaining half as $-1$. The trajectory of the inner weights across GD iterations is depicted in Figure 2.

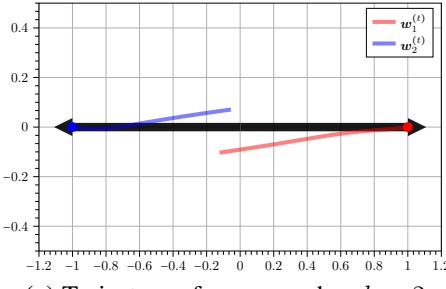
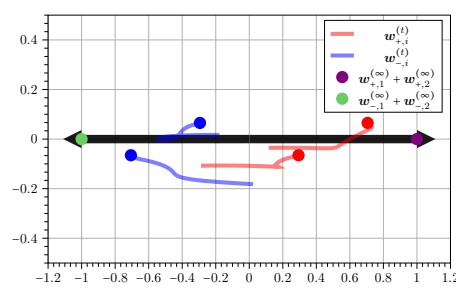

(a) Trajectory of neurons when $k = 2$.    (b) Trajectory of neurons when $k > 2$.

Figure 2: **Trajectory of neurons for different values of** $k$. We run gradient descent updates on the population loss after fixing $v$ with half $+1$'s and half $-1$'s. A randomly selected orthogonal direction to $a$ is shown for the y-axis in order to visualize the neurons in 2D. Black arrows indicate $\pm a$ direction. We use colors red and blue to indicate whether $v_i$ corresponding to $w_i$ is 1 or $-1$ respectively. Points at the end of each trajectory denotes the final weight GD converges to.

We observe that while no individual $w_i$ align itself with $\pm a$ direction, grouping hidden units based on their corresponding signs in $v$ and summing them recovers $\pm a$ *exactly* (purple and green points in Figure 2). Although not depicted here, we have tried various values for $k > 2$ and the observation that grouping weights recover $\pm a$ was consistent. This suggests that combining node aggregation technique from (Li et al., 2024) with our proof strategy may extend our results for the $k > 2$ setting. We leave this to future work.

### 4.2 LEARNING LINEAR TARGETS (MULTI-DIMENSIONAL OUTPUT: $r > 1$)

We only consider the case where the model is exactly parameterized i.e. $k = 2r$. We first show that an interesting pattern arises if both the inner and outer layers of the neural network are initialized sufficiently small. For visualization purposes in Figure 3, we pick $r = 3$ and $k = 6$ (see Appendix G.1 for additional figures with $r > 3$). As for the target function, we pick $a_1, a_2, a_3$ to be $e_1, e_2, e_3$ respectively which correspond to the standard basis vectors in $\mathbb{R}^d$. We plot the trajectory of both the inner and outer layer weights of the network across iterations and observe a peculiar pattern in both $v_i$'s and $w_i$'s. At convergence, weights can be grouped into pairs such that one of the weights is approximately negative of the other. As a concrete example, in Figure 3, we observe that $v_3^{(\infty)} \approx -v_4^{(\infty)}$, $v_1^{(\infty)} \approx -v_5^{(\infty)}$, and $v_2^{(\infty)} \approx -v_6^{(\infty)}$ which also holds similarly for $w_i$'s as well. This suggests that after a permutation of the hidden units, we get

$$V^{(\infty)} \approx [I_r, -I_r]^T \widetilde{V}, \quad W^{(\infty)} \approx [I_r, -I_r]^T \widetilde{W}.$$

This inspires us to fix $V$ according to the pattern above for our theoretical results (see Theorem 6 in the Appendix) as a natural extension to the $v_i = \pm 1$ pattern in the single output setting.

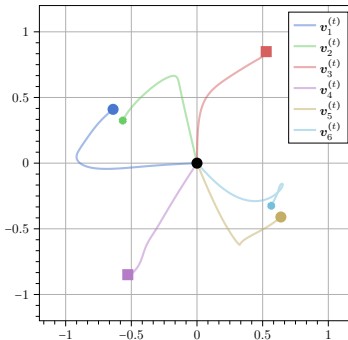 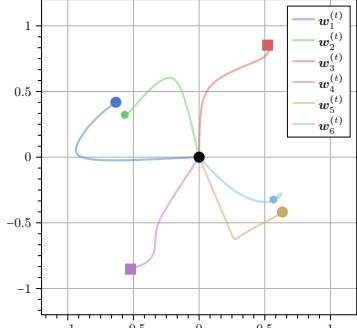

**(a)** Trajectory of $v_i$'s and their pairing behavior.  **(b)** Trajectory of $w_i$'s and their pairing behavior.

Figure 3: **Pairing pattern in multi-dimensional setting.** We train the network from small initialization when exactly parameterized ($k = 6$ and $r = 3$). On left (a), we depict the trajectories of individual weights in the outer layer ($v_i$'s) across iterations. We observe that the weights at convergence can be grouped into three pairs such that one of the weights is approximately negative of the other. For instance, we observe that $v_3^{(\infty)} \approx -v_4^{(\infty)}$. Which neurons end up pairing with each other is indicated by the usage of same symbol (square, circle, etc.). A similar pairing is observed for the inner layer weights as well (b). While these vectors all lie in a higher dimensional space, we pick an arbitrary two dimensional axis to plot them in 2D.

### 4.3 BENEFITS OF SCALING THE LEARNING RATE

In this section we illustrate why a scaled learning rate (i.e. $\mu_i = \frac{\mu}{v_i^2}$), which appears in Theorems 1 and 2, is a natural choice. While this scaling is absent in standard GD updates, we show that it significantly accelerates convergence in the fixed-$v$ setting.

Concretely, in this experiment we focus on a single output setting ($r = 1$) where $v_i$'s are fixed but with a big difference in their magnitudes. In particular, we set $v_1 = 100$ and $v_2 = 1$. We initialize $W$ moderately small with $\alpha = 0.01$ and compare the convergence speed of the loss in the population setting with and without the scaled step size. We separately find the best $\mu$ (via binary search) that achieves the fastest convergence. The loss across iterations is depicted in Figure 4.

As evident in this figure, the difference in convergence speed is rather significant. Without our scaling, about $10^4$ iterations are needed to bring the loss below $10^{-5}$, while with normalization it takes only **3** iterations. Our proposed rescaling, thus enables the use of a large $\mu$ throughout training.

### 5 RELATED WORK

There is a large body of work on developing global convergence guarantees for nonconvex problems. We briefly review this literature and compare the differences with the setting discussed in this paper.

Figure 4: **Effect of scaled step-size.** The population loss decreases significantly faster with a scaled step size compared to its fixed step size counterpart. Step size $\mu$ is tuned separately in both cases.

**Nonconvex low-rank matrix recovery**: In low-rank matrix recovery, numerous studies have shown that nonconvex gradient descent, when initiated with spectral initialization, can effectively solve low-rank reconstruction problems across various domains. This includes phase retrieval (Candès et al., 2015; Chen and Candès, 2017; Ma et al., 2020), matrix sensing Tu et al. (2016), blind deconvolution Li et al. (2019); Ling and Strohmer (2019), and matrix completion Chen et al. (2020). In practice, random initialization is frequently employed instead of specialized spectral initialization methods. As a result, more recent literature Sun et al. (2018); Ge et al. (2016); Zhang et al. (2019), have turned to analyzing the loss landscape. These studies demonstrate that, despite their non-convex nature,

these loss landscapes remain well-behaved under certain assumptions. Specifically, they contain no spurious local minima (i.e., all minimizers are global minima), and saddle points exhibit a strict direction of negative curvature (also known as strict saddle points) Sun et al. (2015). Then specialized truncation or saddle escaping algorithms such as trust region, cubic regularization Nesterov and Polyak (2006); Nocedal and Wright (2006) or noisy (stochastic) gradient-based methods Jin et al. (2017a); Ge et al. (2015); Raginsky et al. (2017); Zhang et al. (2017) are deployed to provably find a global optimum. In contrast to the above literature, the landscape of our loss contain non-strict saddle points. Furthermore, we do not seek any modification to the initialization or the GD updates. Indeed, our result holds with moderately small initialization. As mentioned earlier, we are able to establish this result by developing intricate control of the GD updates throughout the trajectory.

**Gradient-based analysis for neural networks:** A recent line of work is concerned with connecting the analysis of neural network training with the so-called neural tangent kernel (NTK) Jacot et al. (2018); Oymak and Soltanolkotabi (2019; 2020); Du et al. (2019); Arora et al. (2019). The core idea is that with sufficiently large initialization, a neural network can be approximated by its linearization around the origin. This approximation facilitates linking neural network analysis to the well-established theory of kernel methods. This approach is sometimes referred to as lazy training since, under such initialization, the network parameters remain close to their initial values throughout training. However, some research suggests that NTK-based analysis alone may not fully account for the practical success of neural networks. For instance, Chizat et al. (2019) presents empirical evidence indicating that reducing the initialization size can lead to lower test error. Similarly, Ghorbani et al. (2020) observes a performance gap between neural networks and their NTK counterparts, with the gap widening when the covariance matrix is isotropic. We note that in an NTK analysis the parameters stay close to the initialization which is not the case in our setting. Furthermore, an NTK analysis that relies on linearization can not deal with trajectory analysis that avoids local optima. Indeed, an NTK analysis will not yield the directional convergence established in this paper. So in this sense our result can be viewed as going beyond the lazy training in NTK theory. We demonstrate the lack of directional convergence and lack of generalization in the NTK regime empirically in Appendix G.2.

**Beyond NTK and learning of specific target functions.** Recent work carries out analysis of neural networks beyond NTK regime including Damian et al. (2022); Ba et al. (2022); Lee et al. (2024); Xu and Du (2023). Many of these results also focus on learning specific target functions such as ReLUs (Xu and Du, 2023), (Soltanolkotabi, 2017) and polynomials (Damian et al., 2022). These results however typically exclude linear function classes and do not directly involve analysis that requires avoiding bad stationary points explicitly. In fact, many of the existing papers use a pre-processing step or alter the early optimization trajectory to avoid complications arising from the dynamics of learning linear functions (Damian et al., 2022). In contrast, our focus is directly dealing with such intricacies.

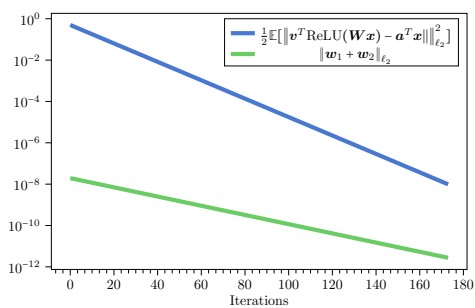

Figure 5: $\|\boldsymbol{w}_1 + \boldsymbol{w}_2\|$ **is monotonically decreasing.** Both the population loss $\mathcal{L}(\boldsymbol{\theta})$ and $\|\boldsymbol{w}_1 + \boldsymbol{w}_2\|$ go to zero at a linear rate.

Among these papers, perhaps the closest to ours in spirit is (Xu and Du, 2023) which studies the problem of fitting an overparameterized ReLU network to a single ReLU target function with a one dimensional output. Our one-dimensional result can be viewed as a generalization of this work (in particular their exact parametrization result) where the target function has two ReLUs with a particular pattern. This is due to the fact that any linear function of the form $\boldsymbol{a}^T\boldsymbol{x}$ can also be written as a difference of two ReLUs: $v_1\text{ReLU}\left(\frac{1}{v_1}\boldsymbol{a}^T\boldsymbol{x}\right) - v_2\text{ReLU}\left(\frac{-1}{v_2}\boldsymbol{a}^T\boldsymbol{x}\right)$ for any $v_1, v_2 > 0$. The addition of this new ReLU with a negative sign introduces non-strict saddle points and various intricacies in the landscape necessitating a completely different analysis. However, compared to (Xu and Du, 2023) we do not study the effect of overparameterization theoretically. Our empirical results in Section 4.1 suggest that such an extension may be possible.

We highlight that besides Xu and Du (2023), there are several other works on learning a single neuron Yehudai and Shamir (2022); Vardi et al. (2022); Chistikov et al. (2023) and variants Brutzkus and

Globerson (2017). As explained before, such results cannot be used to analyze linear targets due to the interaction terms between positive and negative ReLU neurons. Furthermore, we note that the landscape for fitting a single ReLU is fundamentally different as it contains only a single basin of attraction (albeitt a non-convex one). In contrast, as discussed earlier the landscape in our problem include non-strict saddle points significantly complicating gradient descent analysis.

We would also like to discuss the difference between our work and a few other papers Zhong et al. (2017); Zhang et al. (2018); Zhu et al. (2025) that have planted one-hidden layer models. These papers differ in at least one of three ways focusing on (1) local analysis, (2) have sub-optimal sample complexity, and/or (3) assume non-negative outer layer weights. For instance, Zhong et al. (2017) utilize tensor initialization, performing a local analysis rather than a global GD analysis. This local analysis however can not be used to analyze the linear target setting. Indeed, as noted in Remark 4.3 of their work, their analysis requires $\boldsymbol{W}^*$ to be full-rank which does not hold in the linear setting (where the rows of the weight matrix are negatives of each other leading to a minimum singular value is zero). Furthermore, this result also requires resampling the data points at each iteration to ensure convergence of gradient descent where as we use the same samples across all iterations. On a related note, their sample complexity has polynomial dependency on many problem parameters (Theorem 4.2) whereas our proof only requires sample size linear in input dimension $d$.

Similarly, Zhang et al. (2018) provide a local analysis of GD when the outer layer weights are fixed to be all ones. They also utilize results of Zhong et al. (2017) and share similar limitations in terms of the rank requirement on $W^*$. Thus this result can not be used in the linear target setting even for a local analysis. While they improve the sample complexity of (Zhang et al., 2018) by getting rid of the resampling trick, they still end up with a sample complexity polynomial in width of the network.

Wu et al. (2018) consider the setting when student and teacher networks both have 2 neurons. In particular, when the teachers are *orthogonal*, and the outer weights are all ones; they demonstrated an interesting result that the landscape is benign and all saddles are strict. In contrast, the landscape in our problem include non-strict saddle points significantly complicating gradient descent analysis. More recently, Zhu et al. (2025) also consider learning multiple *orthogonal* ReLU neurons in a teacher-student framework with outer layer weights fixed to all ones. As just discussed, having orthogonal teacher weights leads to a much more benign landscape. Moreover, assumptions in the aforementioned works strictly exclude the linear target setting, where the outer layer must contain negative coefficients. Furthermore, they impose strong restrictions on the initialization. Specifically, they look at the convergence after "weak alignment" where for each student neuron there exists only one teacher neuron that is not near perpendicular. Our population results on the other hand can handle initializations where both student neurons are perpendicular to the target direction as long as we have $\|\boldsymbol{w}_1 + \boldsymbol{w}_2\| \leq \frac{1}{2}\|\boldsymbol{a}\|$ (e.g. $\boldsymbol{w}_1 = \boldsymbol{w}_2 = \frac{\|\boldsymbol{a}\|}{4}\bar{\boldsymbol{a}_\perp}$). That said, their analysis can handle over-parametrization ($k \gg \bar{k}^*$) and teacher networks with more than 2 neurons.

In recent and independent work, Boursier and Flammarion (2025) also consider the problem of learning linear target functions. The authors demonstrate an interesting result: despite over-parametrization, the sum of positive (resp. negative) neurons aligns with the OLS estimator obtained from the "positive" (resp. negative) subset of the data. To prove this, the authors impose heavy restrictions on the data distribution (in particular, Conditions 3 and 4 in their paper) to essentially align the data with the target direction and avoid changes in the activation cone. We quote the authors:

"However, item 3 is quite restrictive: it is needed to ensure that the volume of the activation cone containing $\beta^*$ does not vanish when $n \to \infty$. A similar assumption is considered by Chistikov et al. (2023); Tsoy and Konstantinov (2024), for similar reasons. Additionally, Condition 4 ensures that $\mathbb{E}_x[xx^T]\beta^*$ and $\beta^*$ are in the same activation cone. This assumption allows the training dynamics to remain within a single cone after the early alignment phase, significantly simplifying our analysis."

In contrast, we demonstrate feature learning in the linear target setting by performing a full characterization of GD dynamics with a generic data distribution and initialization without any of the restrictive assumptions mentioned above.

## 6 OVERVIEW AND KEY IDEAS OF THE PROOF

In this section we provide an overview of our proof strategy. In the entire proof we focus on the $v_1 = v_2 = 1$ setting. As we demonstrate in Appendix D this is without any loss in generality.

**Key Proof Ideas in the Population Setting.**
Our proof technique relies on showing the following two inequalities:

1. **Correlation inequality:** $\langle W - W^*, \nabla_W \mathcal{L}(W) \rangle \geq \alpha \|W - W^*\|_F^2$

2. **Gradient smoothness (towards the global optima):** $\|\nabla_W \mathcal{L}(W)\|_F \leq \beta \|W - W^*\|_F$

These two inequalities combined imply that the GD iterates converge to a global optima at a linear rate. However, as stated earlier in Section 1.3 the loss landscape has many non-strict saddle points. This immediately implies that the first correlation inequality can not hold over the entire domain. To circumvent this, we will utilize the following key observation.

**Key Observation:** A critical idea towards proving the correlation inequality is showing that along the GD trajectory $\|w_1 + w_2\|$ decreases as also depicted in Figure 5. This is formalized below.

**Lemma 3** *For all iterations $\tau$, we have $\left\| w_1^{(\tau+1)} + w_2^{(\tau+1)} \right\| \leq \left\| w_1^{(\tau)} + w_2^{(\tau)} \right\|$ when $\mu \leq 1$.*

With this key observation in place, we turn our attention to the proof which follows these steps:

- Step 1 (Correlation inequality): We define

$$h(w_1, w_2, a) = \langle W - W^*, \nabla_W \mathcal{L}(W) \rangle - \alpha \|W - W^*\|_F^2, \tag{3}$$

  and compute $\tilde{h}(w_1, w_2) = \min_a h(w_1, w_2, a)$ assuming that $\|w_1 + w_2\| \leq \frac{1}{2}\|a\|$. We show that $\frac{1}{\|w_1\|^2}\tilde{h}(w_1, w_2)$ is only a function of $\theta$ (angle between $w_1$ and $w_2$) and $\frac{\|w_2\|}{\|w_1\|}$. This allows us to draw $\frac{1}{\|w_1\|^2}\tilde{h}$ in 2D and prove that $\tilde{h}$ is indeed non-negative. We provide the visualization in the Appendix. This leads us to the following lemma:

  **Lemma 4** $\langle W - W^*, \nabla_W \mathcal{L}(W) \rangle \geq \alpha \|W - W^*\|_F^2$ *holds with* $\alpha = 0.3$ *as long as* $\|w_1 + w_2\| \leq \frac{1}{2}\|a\|$.

- Step 2 (Gradient smoothness towards the global optima): This step is more straightforward and only requires algebraic manipulations. Concretely we have the following lemma.

  **Lemma 5** $\|\nabla_W \mathcal{L}(W)\|_F \leq \beta \|W - W^*\|_F$ *holds for all* $W \in \mathbb{R}^{2 \times d}$ *with* $\beta$ *a fixed constant.*

**Key Proof Ideas in the Empirical Setting.** To prove our results for the empirical case, we need only establish the empirical counterparts of Lemma 4 and Lemma 5. Here, we will focus on outlining the key ideas behind the proof for the counterpart to Lemma 4 which is more involved. A first idea may be to try to show that the empirical correlation concentrates around the population correlation. However, since we reuse samples across iterations and the correlation inequality involves complex and heavy tail nonlinear functions of the data points, this can be quite challenging. To circumvent this, our key idea is to observe that the empirical correlation can be decomposed into two terms

$$\left\langle W - W^*, \nabla_W \widehat{\mathcal{L}}(W) \right\rangle = \frac{1}{n}\sum_{i=1}^{n} r^2(x_i) + \frac{1}{n}\sum_{i=1}^{n} r(x_i)(a^T x_i)(1 - \phi'(w_1^T x_i) - \phi'(w_2^T x_i))$$

where $r(x) = v^T \text{ReLU}(Wx) - a^T x$ is the residual function. Critically we can show that the second term is dominated by the first term allowing us to conclude that

$$\left\langle W - W^*, \nabla_W \widehat{\mathcal{L}}(W) \right\rangle \geq c \left( \frac{1}{n}\sum_{i=1}^{n} r^2(x_i) \right)$$

holds for a fixed numerical constant (see appendix for detail). Thus to show the empirical version of the correlation inequality with parameter $\alpha$, it suffices to show that

$$\frac{1}{n}\sum_{i=1}^{n} r^2(x_i) \geq \frac{\alpha}{c} \|W - W^*\|_F^2, \tag{4}$$

holds with high probability. To prove this we will first show that $\mathbb{E}_x\left[r^2(x)\right] \geq \frac{\alpha}{c(1-\delta)}\|W - W^*\|_F^2$. Next, we show that as long as $n \geq c\frac{d}{\delta^2}$, then

$$\frac{1}{n}\sum_{i=1}^{n} r^2(x_i) \geq (1-\delta)\mathbb{E}_x\left[r^2(x)\right],$$

holds with high probability. The combination of the latter two results immediately implies (4) finishing the proof. This reduction may seem naive as the sum of the residuals above also involve uniform concentration of heavy-tail stochastic processes. However, critically, the sumands are now positive allowing us to utilize powerful one-sided uniform concentration results that hold despite their heavy-tail nature.

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

APPENDIX

# A  USEFUL CALCULATIONS

In this section we provide the derivation of several useful identities.

## A.1  POPULATION LOSS

Let $\boldsymbol{a}, \boldsymbol{b} \in \mathbb{R}^d$ be two arbitrary vectors. Define

$$
f(\boldsymbol{a}, \boldsymbol{b}) = \mathrm{E}_{\boldsymbol{x}}\left[\left[\boldsymbol{a}^T \boldsymbol{x}\right]_+ \left[\boldsymbol{b}^T \boldsymbol{x}\right]_+\right]
$$

$$
\overset{(a)}{=} \frac{1}{2\pi} \|\boldsymbol{a}\| \|\boldsymbol{b}\| \left(\sin\left(\theta_{\boldsymbol{a},\boldsymbol{b}}\right) + \left(\pi - \theta_{\boldsymbol{a},\boldsymbol{b}}\right) \cos\left(\theta_{\boldsymbol{a},\boldsymbol{b}}\right)\right) \tag{5}
$$

where $\theta_{\boldsymbol{a},\boldsymbol{b}} = \cos^{-1}\left(\frac{\boldsymbol{a}^T \boldsymbol{b}}{\|\boldsymbol{a}\| \|\boldsymbol{b}\|}\right)$, expectation is over $\boldsymbol{x} \sim \mathcal{N}(\boldsymbol{0}, \boldsymbol{I}_d)$ and inequality (a) follows from the Table 1 in (Daniely et al., 2016).

Using these we calculate the closed form for the population loss (2) as:

$$
\mathcal{L}(\boldsymbol{\theta}) = \frac{1}{2} \mathrm{E}_{\boldsymbol{x}}\left[\left\|\sum_{i=1}^{k} \boldsymbol{v}_i \phi\left(\boldsymbol{w}_i^T \boldsymbol{x}\right) - \boldsymbol{A}\boldsymbol{x}\right\|^2\right]
$$

$$
= \frac{1}{2}\sum_{i=1}^{k}\sum_{j=1}^{k} \langle \boldsymbol{v}_i, \boldsymbol{v}_j \rangle \, \mathrm{E}_{\boldsymbol{x}}\left[\phi\left(\boldsymbol{w}_i^T \boldsymbol{x}\right)\phi\left(\boldsymbol{w}_j^T \boldsymbol{x}\right)\right] - \sum_{i=1}^{k} \boldsymbol{v}_i^T \boldsymbol{A} \mathrm{E}_{\boldsymbol{x}}\left[\phi\left(\boldsymbol{w}_i^T \boldsymbol{x}\right)\boldsymbol{x}\right] + \frac{1}{2}\mathrm{E}_{\boldsymbol{x}}\left[\boldsymbol{x}^T \boldsymbol{A}^T \boldsymbol{A}\boldsymbol{x}\right]
$$

$$
\overset{(a)}{=} \frac{1}{2}\sum_{i=1}^{k}\sum_{j=1}^{k} \langle \boldsymbol{v}_i, \boldsymbol{v}_j \rangle f\left(\boldsymbol{w}_i, \boldsymbol{w}_j\right) - \sum_{i=1}^{k} \boldsymbol{v}_i^T \boldsymbol{A} \mathrm{E}_{\boldsymbol{x}}\left[\phi\left(\boldsymbol{w}_i^T \boldsymbol{x}\right)\boldsymbol{x}\right] + \frac{1}{2}\mathrm{E}_{\boldsymbol{x}}\left[\boldsymbol{x}^T \boldsymbol{A}^T \boldsymbol{A}\boldsymbol{x}\right]
$$

$$
\overset{(b)}{=} \frac{1}{2}\sum_{i=1}^{k}\sum_{j=1}^{k} \langle \boldsymbol{v}_i, \boldsymbol{v}_j \rangle f\left(\boldsymbol{w}_i, \boldsymbol{w}_j\right) - \sum_{i=1}^{k} \boldsymbol{v}_i^T \boldsymbol{A} \mathrm{E}_{\boldsymbol{x}}\left[\nabla_{\boldsymbol{x}}\phi\left(\boldsymbol{w}_i^T \boldsymbol{x}\right)\right] + \frac{1}{2}\mathrm{E}_{\boldsymbol{x}}\left[\boldsymbol{x}^T \boldsymbol{A}^T \boldsymbol{A}\boldsymbol{x}\right]
$$

$$
= \frac{1}{2}\sum_{i=1}^{k}\sum_{j=1}^{k} \langle \boldsymbol{v}_i, \boldsymbol{v}_j \rangle f\left(\boldsymbol{w}_i, \boldsymbol{w}_j\right) - \sum_{i=1}^{k} \boldsymbol{v}_i^T \boldsymbol{A}\boldsymbol{w}_i \mathrm{E}_{\boldsymbol{x}}\left[\phi'\left(\boldsymbol{w}_i^T \boldsymbol{x}\right)\right] + \frac{1}{2}\mathrm{E}_{\boldsymbol{x}}\left[\boldsymbol{x}^T \boldsymbol{A}^T \boldsymbol{A}\boldsymbol{x}\right]
$$

$$
\overset{(c)}{=} \frac{1}{2}\sum_{i=1}^{k}\sum_{j=1}^{k} \langle \boldsymbol{v}_i, \boldsymbol{v}_j \rangle f\left(\boldsymbol{w}_i, \boldsymbol{w}_j\right) - \frac{1}{2}\sum_{i=1}^{k} \boldsymbol{v}_i^T \boldsymbol{A}\boldsymbol{w}_i + \frac{1}{2}\mathrm{E}_{\boldsymbol{x}}\left[\boldsymbol{x}^T \boldsymbol{A}^T \boldsymbol{A}\boldsymbol{x}\right]
$$

$$
\overset{(d)}{=} \frac{1}{2}\sum_{i=1}^{k}\sum_{j=1}^{k} \langle \boldsymbol{v}_i, \boldsymbol{v}_j \rangle f\left(\boldsymbol{w}_i, \boldsymbol{w}_j\right) - \frac{1}{2}\sum_{i=1}^{k} \boldsymbol{v}_i^T \boldsymbol{A}\boldsymbol{w}_i + \frac{1}{2}\mathrm{Tr}\left(\boldsymbol{A}\boldsymbol{A}^T\right)
$$

$$
= \frac{1}{4\pi}\sum_{i=1}^{k}\sum_{j=1}^{k} \langle \boldsymbol{v}_i, \boldsymbol{v}_j \rangle \|\boldsymbol{w}_i\| \|\boldsymbol{w}_j\| \left(\sin\theta_{ij} + \left(\pi - \theta_{ij}\right)\cos\theta_{ij}\right) - \frac{1}{2}\sum_{i=1}^{k} \boldsymbol{v}_i^T \boldsymbol{A}\boldsymbol{w}_i + \frac{1}{2}\mathrm{Tr}\left(\boldsymbol{A}\boldsymbol{A}^T\right)
$$

$$
\tag{6}
$$

where equation (a) follows from the definition of $f(\boldsymbol{a}, \boldsymbol{b})$, (b) follows from the Stein's Lemma, (c) follows from the fact that derivative of ReLU activation is the step function and $\boldsymbol{w}_i^T \boldsymbol{x} > 0$ with probability $\frac{1}{2}$, and finally (d) follows from the cyclical property of the trace.

We also write this in a more compact matrix form as follows:

$$
\mathcal{L}(\boldsymbol{\theta}) = \frac{1}{4\pi}\mathrm{Tr}\left(\boldsymbol{V}^T diag(\boldsymbol{\omega})\left(\sin(\boldsymbol{\Theta}) + \left(\pi \mathbb{1}\mathbb{1}^T - \boldsymbol{\Theta}\right)\odot\cos(\boldsymbol{\Theta})\right)diag(\boldsymbol{\omega})\boldsymbol{V}\right) - \frac{1}{2}\mathrm{Tr}\left(\left(\boldsymbol{V}^T \boldsymbol{W} - \boldsymbol{A}\right)\boldsymbol{A}^T\right)
$$

where $\boldsymbol{\omega}_i = \|\boldsymbol{w}_i\|$ and $\boldsymbol{\theta}_{ij}$ is the angle between $\boldsymbol{w}_i$ and $\boldsymbol{w}_j$. When $r = 1$, the expression simplifies further to

$$
\mathcal{L}(\boldsymbol{\theta}) = \frac{1}{4\pi}\boldsymbol{u}^T\left(\sin(\boldsymbol{\Theta}) + \left(\pi \mathbb{1}\mathbb{1}^T - \boldsymbol{\Theta}\right)\odot\cos(\boldsymbol{\Theta})\right)\boldsymbol{u} - \frac{1}{2}\boldsymbol{a}^T \boldsymbol{W}^T \boldsymbol{v} + \frac{1}{2}\|\boldsymbol{a}\|^2
$$

where $\boldsymbol{u} = diag(\boldsymbol{\omega})\boldsymbol{v}$.

## A.2 POPULATION GRADIENT

Let us define,

$$
\begin{aligned}
g\left(\boldsymbol{a}, \boldsymbol{b}\right) &= \frac{\partial}{\partial \boldsymbol{a}} f\left(\boldsymbol{a}, \boldsymbol{b}\right) \\
&= \frac{1}{2\pi}\left(\|\boldsymbol{b}\| \sin\left(\theta_{\boldsymbol{a},\boldsymbol{b}}\right) \bar{\boldsymbol{a}} + \left(\pi - \theta_{\boldsymbol{a},\boldsymbol{b}}\right) \boldsymbol{b}\right) \quad \left(\bar{\boldsymbol{a}} = \frac{\boldsymbol{a}}{\|\boldsymbol{a}\|}, \quad \bar{\boldsymbol{b}} = \frac{\boldsymbol{b}}{\|\boldsymbol{b}\|}\right) \\
&= \frac{\|\boldsymbol{b}\|}{2\pi}\left(\sin\left(\theta_{\boldsymbol{a},\boldsymbol{b}}\right) \bar{\boldsymbol{a}} + \left(\pi - \theta_{\boldsymbol{a},\boldsymbol{b}}\right) \bar{\boldsymbol{b}}\right).
\end{aligned}
\tag{7}
$$

Taking the derivative of (6) with respect to $\boldsymbol{w}_i$, we get

$$
\begin{aligned}
\nabla_{\boldsymbol{w}_i}\mathcal{L}\left(\boldsymbol{\theta}\right) &= \frac{1}{2}\|\boldsymbol{v}_i\|^2 \boldsymbol{w}_i + \sum_{\substack{j=1 \\ i\neq j}}^{k} \langle \boldsymbol{v}_i, \boldsymbol{v}_j \rangle g\left(\boldsymbol{w}_i, \boldsymbol{w}_j\right) - \frac{1}{2}\boldsymbol{A}^T\boldsymbol{v}_i \\
&= \sum_{j=1}^{k} \langle \boldsymbol{v}_i, \boldsymbol{v}_j \rangle g\left(\boldsymbol{w}_i, \boldsymbol{w}_j\right) - \frac{1}{2}\boldsymbol{A}^T\boldsymbol{v}_i \\
&= \frac{1}{2\pi}\sum_{j=1}^{k} \langle \boldsymbol{v}_i, \boldsymbol{v}_j \rangle \|\boldsymbol{w}_j\| \left(\sin\left(\theta_{ij}\right)\bar{\boldsymbol{w}}_i + \left(\pi - \theta_{ij}\right)\bar{\boldsymbol{w}}_j\right) - \frac{1}{2}\boldsymbol{A}^T\boldsymbol{v}_i
\end{aligned}
$$

In matrix form:

$$
\nabla_{\boldsymbol{W}}\mathcal{L}\left(\boldsymbol{\theta}\right) = \frac{1}{2\pi}\left(\left(\boldsymbol{V}\boldsymbol{V}^T \odot \left(\pi\mathbb{1}\mathbb{1}^T - \boldsymbol{\Theta}\right)\right) diag\left(\boldsymbol{\omega}\right) + diag\left(\left(\boldsymbol{V}\boldsymbol{V}^T \odot \sin\boldsymbol{\Theta}\right)\boldsymbol{\omega}\right)\right)\bar{\boldsymbol{W}} - \frac{1}{2}\boldsymbol{V}\boldsymbol{A}
$$

where $\boldsymbol{\omega}_i = \|\boldsymbol{w}_i\|$. When $r = 1$, the expression simplifies further to

$$
\nabla_{\boldsymbol{W}}\mathcal{L}\left(\boldsymbol{\theta}\right) = \frac{1}{2\pi}diag\left(\boldsymbol{v}\right)\left(\left(\pi\mathbb{1}\mathbb{1}^T - \boldsymbol{\Theta}\right)diag\left(\boldsymbol{u}\right) + diag\left(\sin\left(\boldsymbol{\Theta}\right)\boldsymbol{u}\right)\right)\bar{\boldsymbol{W}} - \frac{1}{2}\boldsymbol{v}\boldsymbol{a}^T
\tag{8}
$$

where $\boldsymbol{u} = diag\left(\boldsymbol{\omega}\right)\boldsymbol{v}$.

## A.3 POPULATION HESSIAN

We calculate the population Hessian $\nabla^2_{\text{vect}(\boldsymbol{W}),\text{vect}(\boldsymbol{W})}\mathcal{L}\left(\boldsymbol{\theta}\right)$ below. Define $\bar{\boldsymbol{w}}_l = \frac{\boldsymbol{w}_l}{\|\boldsymbol{w}_l\|}$, $\boldsymbol{P}_{\boldsymbol{w}_l^\perp} = \left(\boldsymbol{I} - \bar{\boldsymbol{w}}_l\bar{\boldsymbol{w}}_l^T\right)$, and $\boldsymbol{w}_{\ell,m^\perp} = \boldsymbol{P}_{\boldsymbol{w}_m^\perp}\boldsymbol{w}_\ell$. Then,

$$
\nabla^2_{\boldsymbol{w}_\ell, \boldsymbol{w}_m}\mathcal{L}\left(\boldsymbol{\theta}\right) = \mathrm{E}_{\boldsymbol{x}}\left[\left(f\left(\boldsymbol{\theta};\boldsymbol{x}\right) - \boldsymbol{a}^T\boldsymbol{x}\right)\nabla^2_{\boldsymbol{w}_\ell,\boldsymbol{w}_m}f\left(\boldsymbol{\theta};\boldsymbol{x}\right) + \nabla_{\boldsymbol{w}_\ell}f\left(\boldsymbol{\theta};\boldsymbol{x}\right)\nabla_{\boldsymbol{w}_m}f\left(\boldsymbol{\theta};\boldsymbol{x}\right)^T\right]
\tag{9}
$$

$$
= \begin{cases}
\frac{\boldsymbol{v}_\ell^2}{2}\boldsymbol{I} + \frac{\boldsymbol{v}_\ell}{2\pi\|\boldsymbol{w}_\ell\|}\sum_{i=1}^{k}\boldsymbol{v}_i\|\boldsymbol{w}_i\|\sin\left(\theta_{\ell,i}\right)\left(\boldsymbol{P}_{\boldsymbol{w}_l^\perp} + \frac{\boldsymbol{P}_{\boldsymbol{w}_l^\perp}\bar{\boldsymbol{w}}_i\bar{\boldsymbol{w}}_i^T\boldsymbol{P}_{\boldsymbol{w}_l^\perp}}{\left\|\boldsymbol{P}_{\boldsymbol{w}_l^\perp}\bar{\boldsymbol{w}}_i\right\|^2}\right) & \ell = m \\
\frac{\boldsymbol{v}_\ell\boldsymbol{v}_m}{2\pi}\left(\bar{\boldsymbol{w}}_\ell\bar{\boldsymbol{w}}_{m,\ell^\perp}^T + \bar{\boldsymbol{w}}_m\bar{\boldsymbol{w}}_{\ell,m^\perp}^T + \left(\pi - \theta_{\ell,m}\right)\boldsymbol{I}\right) & \ell \neq m
\end{cases}
$$

for calculation of individual terms refer down below.

### A.3.1 CALCULATING $\mathrm{E}_{\boldsymbol{x}}\left[\nabla_{\boldsymbol{w}_\ell}f\left(\boldsymbol{\theta};\boldsymbol{x}\right)\nabla_{\boldsymbol{w}_m}f\left(\boldsymbol{\theta};\boldsymbol{x}\right)^T\right]$

$$
\begin{aligned}
\mathrm{E}_{\boldsymbol{x}}\left[\nabla_{\boldsymbol{w}_\ell}f\left(\boldsymbol{\theta};\boldsymbol{x}\right)\nabla_{\boldsymbol{w}_m}f\left(\boldsymbol{\theta};\boldsymbol{x}\right)^T\right] &= \mathrm{E}_{\boldsymbol{x}}\left[\boldsymbol{v}_\ell\phi'\left(\boldsymbol{w}_\ell^T\boldsymbol{x}\right)\boldsymbol{x}\boldsymbol{x}^T\phi'\left(\boldsymbol{w}_m^T\boldsymbol{x}\right)\boldsymbol{v}_m\right] \\
&= \boldsymbol{v}_\ell\boldsymbol{v}_m\mathrm{E}_{\boldsymbol{x}}\left[\phi'\left(\bar{\boldsymbol{w}}_\ell^T\boldsymbol{x}\right)\phi'\left(\bar{\boldsymbol{w}}_m^T\boldsymbol{x}\right)\boldsymbol{x}\boldsymbol{x}^T\right]
\end{aligned}
$$

To tackle the expectation term, we use second order Stein's Lemma, $\mathrm{E}_{\boldsymbol{x}}\left[g(\boldsymbol{x})\boldsymbol{x}\boldsymbol{x}^T\right] = \mathrm{E}_{\boldsymbol{x}}\left[\nabla^2_{\boldsymbol{x}}g\left(\boldsymbol{x}\right)\right] + \mathrm{E}_{\boldsymbol{x}}\left[g\left(\boldsymbol{x}\right)\right]\boldsymbol{I}$.

$$\mathrm{E}_{\boldsymbol{x}}\left[\phi'\left(\bar{\boldsymbol{w}}_{\ell}^T\boldsymbol{x}\right)\phi'\left(\bar{\boldsymbol{w}}_m^T\boldsymbol{x}\right)\boldsymbol{x}\boldsymbol{x}^T\right] = \mathrm{E}_{\boldsymbol{x}}\left[\nabla_{\boldsymbol{x}}^2\left(\phi'\left(\bar{\boldsymbol{w}}_{\ell}^T\boldsymbol{x}\right)\phi'\left(\bar{\boldsymbol{w}}_m^T\boldsymbol{x}\right)\right)\right] + \mathrm{E}_{\boldsymbol{x}}\left[\phi'\left(\bar{\boldsymbol{w}}_{\ell}^T\boldsymbol{x}\right)\phi'\left(\bar{\boldsymbol{w}}_m^T\boldsymbol{x}\right)\right]\boldsymbol{I}$$

First term is:

$$\begin{aligned}
\mathrm{E}_{\boldsymbol{x}}\left[\nabla_{\boldsymbol{x}}^2\left(\phi'\left(\bar{\boldsymbol{w}}_{\ell}^T\boldsymbol{x}\right)\phi'\left(\bar{\boldsymbol{w}}_m^T\boldsymbol{x}\right)\right)\right] &= \mathrm{E}_{\boldsymbol{x}}\left[\delta'\left(\bar{\boldsymbol{w}}_{\ell}^T\boldsymbol{x}\right)\phi'\left(\bar{\boldsymbol{w}}_m^T\boldsymbol{x}\right)\bar{\boldsymbol{w}}_{\ell}\bar{\boldsymbol{w}}_{\ell}^T\right] \\
&\quad + \mathrm{E}_{\boldsymbol{x}}\left[\delta\left(\bar{\boldsymbol{w}}_{\ell}^T\boldsymbol{x}\right)\delta\left(\bar{\boldsymbol{w}}_m^T\boldsymbol{x}\right)\left(\bar{\boldsymbol{w}}_{\ell}\bar{\boldsymbol{w}}_m^T + \bar{\boldsymbol{w}}_m\bar{\boldsymbol{w}}_{\ell}^T\right)\right] \\
&\quad + \mathrm{E}_{\boldsymbol{x}}\left[\phi'\left(\bar{\boldsymbol{w}}_{\ell}^T\boldsymbol{x}\right)\delta'\left(\bar{\boldsymbol{w}}_m^T\boldsymbol{x}\right)\bar{\boldsymbol{w}}_m\bar{\boldsymbol{w}}_m^T\right]
\end{aligned}$$

These terms can be grouped in two.

$$\begin{aligned}
\mathrm{E}_{\boldsymbol{x}}\left[\delta'\left(\bar{\boldsymbol{w}}_{\ell}^T\boldsymbol{x}\right)\phi'\left(\bar{\boldsymbol{w}}_m^T\boldsymbol{x}\right)\right] &= \mathrm{E}_{\boldsymbol{x},g}\left[\delta'\left(g\right)\phi'\left(\bar{\boldsymbol{w}}_m^T\boldsymbol{P}_{\boldsymbol{w}_{\boldsymbol{\ell}}^{\perp}}\boldsymbol{x} + \bar{\boldsymbol{w}}_m^T\bar{\boldsymbol{w}}_{\ell}g\right)\right] \\
&= -\frac{\cos\left(\theta_{\ell,m}\right)}{\sqrt{2\pi}}\mathrm{E}_{\boldsymbol{x}}\left[\delta\left(\bar{\boldsymbol{w}}_m^T\boldsymbol{P}_{\boldsymbol{w}_{\boldsymbol{\ell}}^{\perp}}\boldsymbol{x}\right)\right] \quad \left(\delta'\left(x\right)f\left(x\right) = -f'\left(0\right)\delta\left(x\right)\right) \\
&= -\frac{\cos\left(\theta_{\ell,m}\right)}{2\pi\left\|\boldsymbol{P}_{\boldsymbol{w}_{\boldsymbol{\ell}}^{\perp}}\bar{\boldsymbol{w}}_m\right\|} = -\frac{\cos\left(\theta_{\ell,m}\right)}{2\pi\sin\left(\theta_{\ell,m}\right)}
\end{aligned}$$

and the other one is

$$\begin{aligned}
\mathrm{E}_{\boldsymbol{x}}\left[\delta\left(\bar{\boldsymbol{w}}_{\ell}^T\boldsymbol{x}\right)\delta\left(\bar{\boldsymbol{w}}_m^T\boldsymbol{x}\right)\right] &= \mathrm{E}_{\boldsymbol{x},g}\left[\delta\left(g\right)\delta\left(\bar{\boldsymbol{w}}_m^T\boldsymbol{P}_{\boldsymbol{w}_{\boldsymbol{\ell}}^{\perp}}\boldsymbol{x} + \bar{\boldsymbol{w}}_m^T\bar{\boldsymbol{w}}_{\ell}g\right)\right] \\
&= \frac{1}{2\pi\left\|\boldsymbol{P}_{\boldsymbol{w}_{\boldsymbol{\ell}}^{\perp}}\bar{\boldsymbol{w}}_m\right\|} = \frac{1}{2\pi\sin\left(\theta_{\ell,m}\right)}
\end{aligned}$$

Therefore we get:

$$\mathrm{E}_{\boldsymbol{x}}\left[\nabla_{\boldsymbol{x}}^2\left(\phi'\left(\bar{\boldsymbol{w}}_{\ell}^T\boldsymbol{x}\right)\phi'\left(\bar{\boldsymbol{w}}_m^T\boldsymbol{x}\right)\right)\right] = \frac{\bar{\boldsymbol{w}}_{\ell}\bar{\boldsymbol{w}}_m^T + \bar{\boldsymbol{w}}_m\bar{\boldsymbol{w}}_{\ell}^T}{2\pi\sin\left(\theta_{\ell,m}\right)} - \frac{\cos\left(\theta_{\ell,m}\right)\left(\bar{\boldsymbol{w}}_{\ell}\bar{\boldsymbol{w}}_{\ell}^T + \bar{\boldsymbol{w}}_m\bar{\boldsymbol{w}}_m^T\right)}{2\pi\sin\left(\theta_{\ell,m}\right)}$$

Second term is:

$$\mathrm{E}_{\boldsymbol{x}}\left[\phi'\left(\bar{\boldsymbol{w}}_{\ell}^T\boldsymbol{x}\right)\phi'\left(\bar{\boldsymbol{w}}_m^T\boldsymbol{x}\right)\right]\boldsymbol{I} = \left(\frac{\pi - \theta_{\ell,m}}{2\pi}\right)\boldsymbol{I} \quad \text{(Dual activation of step function)}$$

Combining everything:

$$\mathrm{E}_{\boldsymbol{x}}\left[\nabla_{\boldsymbol{w}_{\ell}}f\left(\boldsymbol{\theta};\boldsymbol{x}\right)\nabla_{\boldsymbol{w}_m}f\left(\boldsymbol{\theta};\boldsymbol{x}\right)^T\right] = \boldsymbol{v}_{\ell}\boldsymbol{v}_m\left(\frac{\bar{\boldsymbol{w}}_{\ell}\bar{\boldsymbol{w}}_m^T + \bar{\boldsymbol{w}}_m\bar{\boldsymbol{w}}_{\ell}^T - \cos\left(\theta_{\ell,m}\right)\left(\bar{\boldsymbol{w}}_{\ell}\bar{\boldsymbol{w}}_{\ell}^T + \bar{\boldsymbol{w}}_m\bar{\boldsymbol{w}}_m^T\right)}{2\pi\sin\left(\theta_{\ell,m}\right)} + \left(\frac{\pi - \theta_{\ell,m}}{2\pi}\right)\boldsymbol{I}\right)$$

or alternatively (by substituting $\cos\left(\theta_{i,j}\right) = \bar{\boldsymbol{w}}_i^T\bar{\boldsymbol{w}}_j$):

$$\mathrm{E}_{\boldsymbol{x}}\left[\nabla_{\boldsymbol{w}_{\ell}}f\left(\boldsymbol{\theta};\boldsymbol{x}\right)\nabla_{\boldsymbol{w}_m}f\left(\boldsymbol{\theta};\boldsymbol{x}\right)^T\right] = \frac{\boldsymbol{v}_{\ell}\boldsymbol{v}_m}{2\pi}\left(\bar{\boldsymbol{w}}_{\ell}\bar{\boldsymbol{w}}_{m,\ell^{\perp}}^T + \bar{\boldsymbol{w}}_m\bar{\boldsymbol{w}}_{\ell,m^{\perp}}^T + \left(\pi - \theta_{\ell,m}\right)\boldsymbol{I}\right)$$

### A.3.2 Calculating $\mathrm{E}_{\boldsymbol{x}}\left[\left(f\left(\boldsymbol{\theta};\boldsymbol{x}\right) - \boldsymbol{a}^T\boldsymbol{x}\right)\nabla_{\boldsymbol{w}_{\ell},\boldsymbol{w}_m}^2 f\left(\boldsymbol{\theta};\boldsymbol{x}\right)\right]$

Note that $\nabla_{\boldsymbol{w}_{\ell},\boldsymbol{w}_m}^2 f\left(\boldsymbol{\theta};\boldsymbol{x}\right) = diag\left(\boldsymbol{v}\odot\phi''\left(\boldsymbol{W}\boldsymbol{x}\right)\right)_{\ell,m}\boldsymbol{x}\boldsymbol{x}^T$. This expectation is $\boldsymbol{0}$ when $\ell \neq m$.
Define $g = \bar{\boldsymbol{w}}_l^T\boldsymbol{x} \sim \mathcal{N}\left(0,1\right)$.

$$\begin{aligned}
\mathrm{E}_{\boldsymbol{x}}\left[\left(f\left(\boldsymbol{\theta};\boldsymbol{x}\right) - \boldsymbol{a}^T\boldsymbol{x}\right)\nabla_{\boldsymbol{w}_{\ell},\boldsymbol{w}_{\ell}}^2 f\left(\boldsymbol{\theta};\boldsymbol{x}\right)\right] &= \mathrm{E}_{\boldsymbol{x}}\left[r\left(\boldsymbol{x}\right)\boldsymbol{v}_{\ell}\delta\left(\boldsymbol{w}_{\ell}^T\boldsymbol{x}\right)\boldsymbol{x}\boldsymbol{x}^T\right] \\
&= \frac{\boldsymbol{v}_{\ell}}{\|\boldsymbol{w}_{\ell}\|}\mathrm{E}_{\boldsymbol{x}}\left[r\left(\boldsymbol{x}\right)\delta\left(\bar{\boldsymbol{w}}_{\ell}^T\boldsymbol{x}\right)\boldsymbol{x}\boldsymbol{x}^T\right] \\
&= \frac{\boldsymbol{v}_{\ell}}{\|\boldsymbol{w}_{\ell}\|}\mathrm{E}_{\boldsymbol{x},g}\left[r\left(\boldsymbol{P}_{\boldsymbol{w}_i^{\perp}}\boldsymbol{x} + \bar{\boldsymbol{w}}_l g\right)\delta\left(g\right)\left(\boldsymbol{P}_{\boldsymbol{w}_i^{\perp}}\boldsymbol{x} + \bar{\boldsymbol{w}}_l g\right)\left(\boldsymbol{P}_{\boldsymbol{w}_i^{\perp}}\boldsymbol{x} + \bar{\boldsymbol{w}}_l g\right)^T\right] \\
&= \frac{\boldsymbol{v}_{\ell}}{\sqrt{2\pi}\|\boldsymbol{w}_{\ell}\|}\boldsymbol{P}_{\boldsymbol{w}_i^{\perp}}\mathrm{E}_{\boldsymbol{x}}\left[r\left(\boldsymbol{P}_{\boldsymbol{w}_i^{\perp}}\boldsymbol{x}\right)\boldsymbol{x}\boldsymbol{x}^T\right]\boldsymbol{P}_{\boldsymbol{w}_i^{\perp}}
\end{aligned}$$

To tackle the expectation term, we use second order Stein's Lemma, $\mathrm{E}_{\boldsymbol{x}}\left[g(\boldsymbol{x})\boldsymbol{x}\boldsymbol{x}^T\right] = \mathrm{E}_{\boldsymbol{x}}\left[\nabla_{\boldsymbol{x}}^2 g\left(\boldsymbol{x}\right)\right] + \mathrm{E}_{\boldsymbol{x}}\left[g\left(\boldsymbol{x}\right)\right]\boldsymbol{I}$.

$$\mathrm{E}_{\boldsymbol{x}}\left[r\left(\boldsymbol{P}_{\boldsymbol{w}_l^\perp}\boldsymbol{x}\right)\boldsymbol{x}\boldsymbol{x}^T\right] = \mathrm{E}_{\boldsymbol{x}}\left[\nabla_{\boldsymbol{x}}^2 r\left(\boldsymbol{P}_{\boldsymbol{w}_l^\perp}\boldsymbol{x}\right)\right] + \mathrm{E}_{\boldsymbol{x}}\left[r\left(\boldsymbol{P}_{\boldsymbol{w}_l^\perp}\boldsymbol{x}\right)\right]\boldsymbol{I}$$

First term is:

$$\begin{aligned}
\mathrm{E}_{\boldsymbol{x}}\left[\nabla_{\boldsymbol{x}}^2 r\left(\boldsymbol{P}_{\boldsymbol{w}_l^\perp}\boldsymbol{x}\right)\right] &= \mathrm{E}_{\boldsymbol{x}}\left[\nabla_{\boldsymbol{x}}^2\left(\sum_{i=1}^k \boldsymbol{v}_i\phi\left(\boldsymbol{w}_i^T\boldsymbol{P}_{\boldsymbol{w}_l^\perp}\boldsymbol{x}\right)\right)\right] \quad (\boldsymbol{a}^T\boldsymbol{x} \text{ vanishes.})\\
&= \sum_{i=1}^k \boldsymbol{v}_i\mathrm{E}_{\boldsymbol{x}}\left[\nabla_{\boldsymbol{x}}^2\phi\left(\boldsymbol{w}_i^T\boldsymbol{P}_{\boldsymbol{w}_l^\perp}\boldsymbol{x}\right)\right]\\
&= \sum_{i=1}^k \boldsymbol{v}_i\mathrm{E}_{\boldsymbol{x}}\left[\delta\left(\boldsymbol{w}_i^T\boldsymbol{P}_{\boldsymbol{w}_l^\perp}\boldsymbol{x}\right)\boldsymbol{P}_{\boldsymbol{w}_l^\perp}\boldsymbol{w}_i\boldsymbol{w}_i^T\boldsymbol{P}_{\boldsymbol{w}_l^\perp}\right]\\
&= \sum_{i=1}^k \frac{\boldsymbol{v}_i}{\left\|\boldsymbol{P}_{\boldsymbol{w}_l^\perp}\boldsymbol{w_i}\right\|}\mathrm{E}_u\left[\delta\left(u\right)\right]\boldsymbol{P}_{\boldsymbol{w}_l^\perp}\boldsymbol{w}_i\boldsymbol{w}_i^T\boldsymbol{P}_{\boldsymbol{w}_l^\perp}\\
&= \frac{1}{\sqrt{2\pi}}\sum_{i=1}^k \frac{\boldsymbol{v}_i}{\|\boldsymbol{w}_i\|\sin\left(\theta_{l,i}\right)}\boldsymbol{P}_{\boldsymbol{w}_l^\perp}\boldsymbol{w}_i\boldsymbol{w}_i^T\boldsymbol{P}_{\boldsymbol{w}_l^\perp} \quad \text{(Delta integration)}\\
&= \frac{1}{\sqrt{2\pi}}\sum_{i=1}^k \frac{\boldsymbol{v}_i\|\boldsymbol{w}_i\|}{\sin\left(\theta_{l,i}\right)}\boldsymbol{P}_{\boldsymbol{w}_l^\perp}\bar{\boldsymbol{w}}_i\bar{\boldsymbol{w}}_i^T\boldsymbol{P}_{\boldsymbol{w}_l^\perp}
\end{aligned}$$

Second term is:

$$\begin{aligned}
\mathrm{E}_{\boldsymbol{x}}\left[r\left(\boldsymbol{P}_{\boldsymbol{w}_l^\perp}\boldsymbol{x}\right)\right] &= \sum_{i=1}^k \boldsymbol{v}_i\mathrm{E}_{\boldsymbol{x}}\left[\phi\left(\boldsymbol{w}_i^T\boldsymbol{P}_{\boldsymbol{w}_l^\perp}\boldsymbol{x}\right)\right]\\
&= \sum_{i=1}^k \boldsymbol{v}_i\left\|\boldsymbol{P}_{\boldsymbol{w}_l^\perp}\boldsymbol{w_i}\right\|\mathrm{E}_u\left[\phi\left(u\right)\right] \quad (u\sim N\left(0,1\right))\\
&= \frac{1}{\sqrt{2\pi}}\sum_{i=1}^k \boldsymbol{v}_i\|\boldsymbol{w}_i\|\sin\left(\theta_{l,i}\right) \quad \left(\text{Expectation of rectified Gaussian } f_{\boldsymbol{x}}(0) = \frac{1}{\sqrt{2\pi}}\right)
\end{aligned}$$

Combining both terms we get

$$\mathrm{E}_{\boldsymbol{x}}\left[r\left(\boldsymbol{P}_{\boldsymbol{w}_l^\perp}\boldsymbol{x}\right)\boldsymbol{x}\boldsymbol{x}^T\right] = \frac{1}{\sqrt{2\pi}}\sum_{i=1}^k \boldsymbol{v}_i\|\boldsymbol{w}_i\|\left(\sin\left(\theta_{l,i}\right)\boldsymbol{I} + \frac{\boldsymbol{P}_{\boldsymbol{w}_l^\perp}\bar{\boldsymbol{w}}_i\bar{\boldsymbol{w}}_i^T\boldsymbol{P}_{\boldsymbol{w}_l^\perp}}{\sin\left(\theta_{l,i}\right)}\right).$$

Finally we plug this back to get:

$$\begin{aligned}
\mathrm{E}_{\boldsymbol{x}}\left[r\left(\boldsymbol{x}\right)\nabla_{\boldsymbol{w}_\ell,\boldsymbol{w}_\ell}^2 f\left(\boldsymbol{\theta};\boldsymbol{x}\right)\right] &= \frac{\boldsymbol{v}_\ell}{2\pi\|\boldsymbol{w}_\ell\|}\boldsymbol{P}_{\boldsymbol{w}_l^\perp}\left(\sum_{i=1}^k \boldsymbol{v}_i\|\boldsymbol{w}_i\|\left(\sin\left(\theta_{l,i}\right)\boldsymbol{I} + \frac{\boldsymbol{P}_{\boldsymbol{w}_l^\perp}\bar{\boldsymbol{w}}_i\bar{\boldsymbol{w}}_i^T\boldsymbol{P}_{\boldsymbol{w}_l^\perp}}{\sin\left(\theta_{l,i}\right)}\right)\right)\boldsymbol{P}_{\boldsymbol{w}_l^\perp}\\
&= \frac{\boldsymbol{v}_\ell}{2\pi\|\boldsymbol{w}_\ell\|}\sum_{i=1}^k \boldsymbol{v}_i\|\boldsymbol{w}_i\|\left(\sin\left(\theta_{l,i}\right)\boldsymbol{P}_{\boldsymbol{w}_l^\perp} + \frac{\boldsymbol{P}_{\boldsymbol{w}_l^\perp}\bar{\boldsymbol{w}}_i\bar{\boldsymbol{w}}_i^T\boldsymbol{P}_{\boldsymbol{w}_l^\perp}}{\sin\left(\theta_{l,i}\right)}\right)\\
&= \frac{\boldsymbol{v}_\ell}{2\pi\|\boldsymbol{w}_\ell\|}\sum_{i=1}^k \boldsymbol{v}_i\|\boldsymbol{w}_i\|\sin\left(\theta_{l,i}\right)\left(\boldsymbol{P}_{\boldsymbol{w}_l^\perp} + \frac{\boldsymbol{P}_{\boldsymbol{w}_l^\perp}\bar{\boldsymbol{w}}_i\bar{\boldsymbol{w}}_i^T\boldsymbol{P}_{\boldsymbol{w}_l^\perp}}{\left\|\boldsymbol{P}_{\boldsymbol{w}_l^\perp}\bar{\boldsymbol{w}}_i\right\|^2}\right)
\end{aligned}$$

### A.4 POPULATION CORRELATION

Using the population gradient in Eq. 8, for $v_1 = v_2 = 1$ we calculate:

$$
\langle \boldsymbol{W} - \boldsymbol{W}^*, \nabla_{\boldsymbol{W}} \mathcal{L}(\boldsymbol{W}) \rangle = \langle \boldsymbol{w}_1 - \boldsymbol{a}, \nabla_{\boldsymbol{w}_1} \mathcal{L}(\boldsymbol{W}) \rangle + \langle \boldsymbol{w}_2 + \boldsymbol{a}, \nabla_{\boldsymbol{w}_2} \mathcal{L}(\boldsymbol{W}) \rangle
$$

$$
= \left\langle \boldsymbol{w}_1 - \boldsymbol{a}, -\frac{\boldsymbol{a}}{2} + \frac{1}{2\pi}(\pi \boldsymbol{w}_1 - \sin\theta \|\boldsymbol{w}_2\| \bar{\boldsymbol{w}}_1 - (\pi - \theta)\boldsymbol{w}_2) \right\rangle
$$

$$
+ \left\langle \boldsymbol{w}_2 + \boldsymbol{a}, \frac{\boldsymbol{a}}{2} - \frac{1}{2\pi}((\pi - \theta)\boldsymbol{w}_1 + \sin\theta\|\boldsymbol{w}_1\|\bar{\boldsymbol{w}}_2 - \pi\boldsymbol{w}_2) \right\rangle
$$

$$
= \frac{\|\boldsymbol{w}_1 - \boldsymbol{a}\|^2}{2} - \frac{\|\boldsymbol{w}_2\|}{2\pi}\langle \boldsymbol{w}_1 - \boldsymbol{a}, \sin\theta \bar{\boldsymbol{w}}_1 + (\pi - \theta)\bar{\boldsymbol{w}}_2 \rangle
$$

$$
+ \frac{\|\boldsymbol{w}_2 + \boldsymbol{a}\|^2}{2} - \frac{\|\boldsymbol{w}_1\|}{2\pi}\langle \boldsymbol{w}_2 + \boldsymbol{a}, \sin\theta \bar{\boldsymbol{w}}_2 + (\pi - \theta)\bar{\boldsymbol{w}}_1 \rangle
$$

$$
= \frac{\|\boldsymbol{w}_1 - \boldsymbol{a}\|^2}{2} - \frac{\|\boldsymbol{w}_2\|}{2\pi}\|\boldsymbol{w}_1\|(\sin\theta + (\pi - \theta)\cos\theta)
$$

$$
+ \frac{\|\boldsymbol{w}_2\|}{2\pi}\left(\frac{\sin\theta}{\|\boldsymbol{w}_1\|}\boldsymbol{w}_1^T\boldsymbol{a} + \frac{\pi - \theta}{\|\boldsymbol{w}_2\|}\boldsymbol{w}_2^T\boldsymbol{a}\right) + \frac{\|\boldsymbol{w}_2 + \boldsymbol{a}\|^2}{2}
$$

$$
- \frac{\|\boldsymbol{w}_1\|}{2\pi}\|\boldsymbol{w}_2\|(\sin\theta + (\pi - \theta)\cos\theta) - \frac{\|\boldsymbol{w}_1\|}{2\pi}\left(\frac{\sin\theta}{\|\boldsymbol{w}_2\|}\boldsymbol{w}_2^T\boldsymbol{a} + \frac{\pi - \theta}{\|\boldsymbol{w}_1\|}\boldsymbol{w}_1^T\boldsymbol{a}\right).
$$

Rearranging the terms, we obtain:

$$
\langle \boldsymbol{W} - \boldsymbol{W}^*, \nabla_{\boldsymbol{W}} \mathcal{L}(\boldsymbol{W}) \rangle = \frac{\|\boldsymbol{w}_1\|^2}{2} + \frac{\|\boldsymbol{w}_2\|^2}{2} - \left(\frac{(\pi - \theta)\cos\theta + \sin\theta}{\pi}\right)\|\boldsymbol{w}_1\|\|\boldsymbol{w}_2\|
$$

$$
- \left(\frac{3\pi - \sin\theta \frac{\|\boldsymbol{w}_2\|}{\|\boldsymbol{w}_1\|} - \theta}{2\pi}\right)\boldsymbol{w}_1^T\boldsymbol{a} + \left(\frac{3\pi - \sin\theta \frac{\|\boldsymbol{w}_1\|}{\|\boldsymbol{w}_2\|} - \theta}{2\pi}\right)\boldsymbol{w}_2^T\boldsymbol{a} + \|\boldsymbol{a}\|^2.
$$

## B ReLU NETWORKS WITH MULTI-DIMENSIONAL OUTPUTS

We now turn our attention to ReLU networks with multiple outputs. In this case running GD from small random initialization on both layers yields an interesting pattern. In particular, at convergence, weights can be grouped into pairs such that one of the weights in the pair is approximately negative of the other one (we discuss this further in Section 4.2).

Given the emergence of this interesting pattern in the outer layer, in our theory we fix the outer layer according to this pattern and focus on the trajectory of the input weights.

**Theorem 6** *Suppose the feature vectors are distributed i.i.d. according to a Gaussian distribution $\boldsymbol{x} \sim \mathcal{N}(\boldsymbol{0}, \boldsymbol{I}_d)$. We assume the corresponding output are generated according to a multi-dimensional linear target function of the form $\boldsymbol{y} = \boldsymbol{A}\boldsymbol{x} \in \mathbb{R}^r$ where $\boldsymbol{A} \in \mathbb{R}^{r \times d}$ is an arbitrary matrix. To learn this linear function we fit a one hidden layer ReLU network with $2r$ hidden nodes of the form*

$$
\boldsymbol{x} \mapsto \boldsymbol{V}^T ReLU(\boldsymbol{W}\boldsymbol{x})
$$

*Here, $\boldsymbol{W} \in \mathbb{R}^{2r \times d}$ and we fix $\boldsymbol{V}$ in the form of $[\boldsymbol{I}_r, -\boldsymbol{I}_r]^T \widetilde{\boldsymbol{V}}$. Here, we assume $\widetilde{\boldsymbol{V}} \in \mathbb{R}^{r \times r}$ is of the form $\boldsymbol{\Sigma}\boldsymbol{R}$ where $\boldsymbol{R} \in \mathbb{R}^{r \times r}$ is an orthonormal matrix and $\boldsymbol{\Sigma}$ is a diagonal matrix with positive entries. Now consider the population loss*

$$
\mathcal{L}(\boldsymbol{W}) = \frac{1}{2}\mathrm{E}_{\boldsymbol{x}}\left[\left\|\boldsymbol{V}^T ReLU(\boldsymbol{W}\boldsymbol{x}) - \boldsymbol{A}\boldsymbol{x}\right\|^2\right].
$$

*To fit this model we run gradient updates of the form $\boldsymbol{W}^{(\tau+1)} = \boldsymbol{W}^{(\tau)} - \mu\nabla\mathcal{L}(\boldsymbol{W}^{(\tau)})$, starting from an initial estimate $\boldsymbol{W}^{(0)}$ with step size obeying $\mu \leq c_3$. Furthermore, assume the initialization obeys*

$$
\left\|\boldsymbol{w}_\ell^{(0)} + \boldsymbol{w}_{\ell+r}^{(0)}\right\| \leq \frac{1}{2}\|\widetilde{\boldsymbol{a}}_\ell\| \quad \text{for all } \ell = 1, 2, \ldots, r,
$$

*where $\widetilde{\boldsymbol{a}}_\ell$ is the $\ell$th row of $\boldsymbol{\Sigma}^{-1}\boldsymbol{R}\boldsymbol{A}$, we have*

$$\left\|\boldsymbol{W}^{(\tau)} - \boldsymbol{W}^*\right\|_F^2 \leq (1 - c_4\mu)^\tau \left\|\boldsymbol{W}^{(0)} - \boldsymbol{W}^*\right\|_F^2$$

*for all iterations $\tau$. Here, $\boldsymbol{W}^* = \begin{bmatrix} \boldsymbol{\Sigma}^{-1}\boldsymbol{R}\boldsymbol{A} \\ -\boldsymbol{\Sigma}^{-1}\boldsymbol{R}\boldsymbol{A} \end{bmatrix}$ and all $c_j$'s are fixed numerical constants independent of any problem dimensions.*

This result directly generalizes our one dimensional result. The initialization requirement is similar showing that sums of pairs of rows of the inner weights $\boldsymbol{W}$ should be sufficiently small at initialization. Similarly, it shows that one can indeed use GD to train a one-hidden layer network with $2r$ hidden nodes to learn any linear target function with $r$ outputs. Our result also implies a directional convergence in the sense the rows of the first layer weights align themselves with the corresponding rows of $\boldsymbol{A}$ or its negative direction.

## C  LANDSCAPE CALCULATIONS

In this section, we verify that:

$$\boldsymbol{w}_1 = \frac{(c+1)\boldsymbol{a}}{v_1} \quad \text{and} \quad \boldsymbol{w}_2 = \frac{c\boldsymbol{a}}{v_2}$$

for arbitrary $c > 0$ or $c < -1$ are indeed stationary points of our optimization problem when $k = 2$, $r = 1$. We first show that the gradient vanishes. Plugging the $\boldsymbol{w}_1$ and $\boldsymbol{w}_2$ into (2):

$$
\begin{aligned}
\nabla_{\boldsymbol{W}}\mathcal{L}(\boldsymbol{\theta}) &= \frac{1}{2\pi}diag\left(\boldsymbol{v}\right)\left(\left(\pi\mathbb{1}\mathbb{1}^T - \boldsymbol{\Theta}\right)diag\left(\boldsymbol{u}\right) + diag\left(\sin\left(\boldsymbol{\Theta}\right)\boldsymbol{u}\right)\right)\bar{\boldsymbol{W}} - \frac{1}{2}\boldsymbol{v}\boldsymbol{a}^T \\
&\overset{(a)}{=} \frac{1}{2}diag\left(\boldsymbol{v}\right)\mathbb{1}\mathbb{1}^T diag\left(\boldsymbol{v}\right)\boldsymbol{W} - \frac{1}{2}\boldsymbol{v}\boldsymbol{a}^T \\
&= \frac{1}{2}\boldsymbol{v}\left(\boldsymbol{W}^T\boldsymbol{v} - \boldsymbol{a}\right)^T \\
&= \frac{1}{2}\boldsymbol{v}\left(v_1\boldsymbol{w}_1 - v_2\boldsymbol{w}_2 - \boldsymbol{a}\right)^T = \frac{1}{2}\boldsymbol{v}\left(\boldsymbol{a} - \boldsymbol{a}\right)^T = \boldsymbol{0}.
\end{aligned}
$$

where (a) follows from the fact that $\boldsymbol{\Theta} = \boldsymbol{0}$ at these points. Next we show that the Hessian at these points are PSD. Plugging the values into (9) we get:

$$\nabla^2_{\boldsymbol{w}_\ell, \boldsymbol{w}_m}\mathcal{L}(\boldsymbol{\theta}) = \begin{cases} \frac{v_\ell^2}{2}\boldsymbol{I} & \ell = m \\ \frac{-v_\ell v_m}{2}\boldsymbol{I} & \ell \neq m \end{cases}$$

which follows from the fact that $\theta_{\ell,i} = 0$ and $\bar{\boldsymbol{w}}_{m,\ell^\perp} = \bar{\boldsymbol{w}}_{\ell,m^\perp} = \boldsymbol{0}$ for any choice of $\ell, m, i \in [2]$. Since $k = 2$, the resulting Hessian matrix have $\frac{v_1^2}{2}\boldsymbol{I}$ and $\frac{v_2^2}{2}\boldsymbol{I}$ on its diagonal blocks and $\frac{-v_1 v_2}{2}\boldsymbol{I}$ on its off-diagonal blocks. Such a matrix have eigenvalues 0 and $\frac{v_1^2 + v_2^2}{2}$ each with multiplicity $d$. Therefore, all the stationary points are in fact non-strict saddle points of the problem.

## D  REDUCTION OF $v_1, v_2 > 0$ TO $v_1 = v_2 = 1$.

Let us define $\tilde{\boldsymbol{w}}_i = v_i\boldsymbol{w}_i$ as the *simulated* student neuron. We will show that the iterates of the $v_1, v_2 > 0$ with $\boldsymbol{w}_i$ as student neurons is identical to having $v_1 = v_2 = 1$ but with $\tilde{\boldsymbol{w}}_i$ as student neurons. First we note that due to homogeneity of ReLU function, we can combine $v_i$ and $\boldsymbol{w}_i$ without changing the loss function. Furthermore, gradient with respect to $\boldsymbol{w}_i$ and $\tilde{\boldsymbol{w}}_i$ are related via the following identity:

$$\nabla_{\boldsymbol{w}_i}\mathcal{L} = \mathcal{D}_{\boldsymbol{w}_i}\left(v_i\boldsymbol{w}_i\right)\nabla_{\tilde{\boldsymbol{w}}_i}\mathcal{L} = v_i\boldsymbol{I}\nabla_{\tilde{\boldsymbol{w}}_i}\mathcal{L} = v_i\nabla_{\tilde{\boldsymbol{w}}_i}\mathcal{L}.$$

where $\mathcal{D}$ denotes the derivative operation. Then, we can rewrite the GD iterates as:

$$\boldsymbol{w}_i^{(t+1)} = \boldsymbol{w}_i^{(t)} - \frac{\mu}{v_i^2}\nabla_{\boldsymbol{w}_i}\mathcal{L}$$

$$v_i\boldsymbol{w}_i^{(t+1)} = v_i\boldsymbol{w}_i^{(t)} - \mu\frac{\nabla_{\boldsymbol{w}_i}\mathcal{L}}{v_i}$$

$$\tilde{\boldsymbol{w}}_i^{(t+1)} = \tilde{\boldsymbol{w}}_i^{(t)} - \mu\nabla_{\tilde{\boldsymbol{w}}_i}\mathcal{L}.$$

Therefore, we first prove our theorems with $v_1 = v_2 = 1$ which directly implies the proof for the general case by plugging in $\tilde{w}_i = v_i w_i$ instead. Note that the direct substitution of $w_i \to v_i w_i$ only implies:

$$\left\| \text{diag}\left( \begin{bmatrix} v_1 \\ v_2 \end{bmatrix} \right) W^{(\tau)} - W^* \right\|_F^2 \leq (1 - c\mu)^\tau \left\| \text{diag}\left( \begin{bmatrix} v_1 \\ v_2 \end{bmatrix} \right) W^{(0)} - W^* \right\|_F^2,$$

where $W^* = [a \quad -a]^T$ and $c$ is picked appropriately for population and empirical settings separately. To complete the proof, we take the diag terms out, re-define the target to be $W^* = \left[ \frac{a}{v_1} \quad -\frac{a}{v_2} \right]^T$, and use a simple inequality to obtain:

$$\min\left( v_1^2, v_2^2 \right) \left\| W^{(\tau)} - W^* \right\|_F^2 \leq (1 - c\mu)^\tau \max(v_1^2, v_2^2) \left\| W^{(0)} - W^* \right\|_F^2.$$

Rearranging the terms completes the proof for the $v_1, v_2 > 0$ setting.

# E    PROOF OF KEY LEMMAS

## E.1    PROOF OF KEY LEMMAS IN THE POPULATION SETTING

### E.1.1    PROOF OF THE MONOTONIC DECREASING PROPERTY (LEMMA 3)

For simplicity of notation, let $w_1 = w_1^{(\tau)}$, $w_2 = w_2^{(\tau)}$ and $\theta$ to be the angle between $w_1$ and $w_2$. To derive the formula for $w_1^{(\tau+1)} + w_2^{(\tau+1)}$, we compute that

$$\nabla_{w_1} \mathcal{L}(\theta) + \nabla_{w_2} \mathcal{L}(\theta) = \frac{1}{2}\left( \frac{\theta}{\pi}(w_1 + w_2) - \frac{\sin(\theta)}{\pi}(\|w_2\| \overline{w}_1 + \|w_1\| \overline{w}_2) \right),$$

which means

$$w_1^{(\tau+1)} + w_2^{(\tau+1)} = w_1 + w_2 - \frac{1}{2}\mu\left( \frac{\theta}{\pi}(w_1 + w_2) - \frac{\sin(\theta)}{\pi}(\|w_2\| \overline{w}_1 + \|w_1\| \overline{w}_2) \right),$$

The square norm of which is

$$\left\| w_1^{(\tau+1)} + w_2^{(\tau+1)} \right\|^2$$

$$= \left\| w_1 + w_2 - \frac{1}{2}\mu\left( \frac{\theta}{\pi}(w_1 + w_2) - \frac{\sin(\theta)}{\pi}(\|w_2\| \overline{w}_1 + \|w_1\| \overline{w}_2) \right) \right\|^2$$

$$= \left\| \left( \left( 1 - \frac{\mu\theta}{2\pi} \right)\|w_1\| + \frac{\mu \sin(\theta)}{2\pi}\|w_2\| \right) \overline{w}_1 + \left( \left( 1 - \frac{\mu\theta}{2\pi} \right)\|w_2\| + \frac{\mu \sin(\theta)}{2\pi}\|w_1\| \right) \overline{w}_2 \right\|^2$$

$$= \left( \left( 1 - \frac{\mu\theta}{2\pi} \right)^2 + \left( \frac{\mu \sin(\theta)}{2\pi} \right)^2 + 2\cos(\theta)\left( 1 - \frac{\mu\theta}{2\pi} \right)\frac{\mu \sin(\theta)}{2\pi} \right) \left( \|w_1\|^2 + \|w_2\|^2 \right)$$

$$+ 2\left( 2\left( 1 - \frac{\mu\theta}{2\pi} \right)\frac{\mu \sin(\theta)}{2\pi} + \cos\theta\left( \left( 1 - \frac{\mu\theta}{2\pi} \right)^2 + \left( \frac{\mu \sin(\theta)}{2\pi} \right)^2 \right) \right) \|w_1\| \|w_2\|.$$

For simplicity of notation, call $\alpha = \left( 1 - \frac{\mu\theta}{2\pi} \right)^2 + \left( \frac{\mu \sin(\theta)}{2\pi} \right)^2$ and $\beta = 2\left( 1 - \frac{\mu\theta}{2\pi} \right)\frac{\mu \sin(\theta)}{2\pi}$. We have that

$$\left\| w_1^{(\tau+1)} + w_2^{(\tau+1)} \right\|^2 = (\alpha + \beta\cos(\theta))\left( \|w_1\|^2 + \|w_2\|^2 \right) + 2(\beta + \alpha\cos(\theta))\|w_1\| \|w_2\|.$$

To show $\left\| \boldsymbol{w}_1^{(\tau+1)} + \boldsymbol{w}_2^{(\tau+1)} \right\| \leq \left\| \boldsymbol{w}_1^{(\tau)} + \boldsymbol{w}_2^{(\tau)} \right\|$, we note that

$$\left\| \boldsymbol{w}_1^{(\tau)} + \boldsymbol{w}_2^{(\tau)} \right\|^2 - \left\| \boldsymbol{w}_1^{(\tau+1)} + \boldsymbol{w}_2^{(\tau+1)} \right\|^2$$

$$= \|\boldsymbol{w}_1 + \boldsymbol{w}_2\|^2 - \left\| \boldsymbol{w}_1^{(\tau+1)} + \boldsymbol{w}_2^{(\tau+1)} \right\|^2$$

$$= \|\boldsymbol{w}_1\|^2 + \|\boldsymbol{w}_2\|^2 + 2\cos\theta \|\boldsymbol{w}_1\| \|\boldsymbol{w}_2\|$$

$$\quad - (\alpha + \beta\cos(\theta)) \left( \|\boldsymbol{w}_1\|^2 + \|\boldsymbol{w}_2\|^2 \right) - 2(\beta + \alpha\cos(\theta)) \|\boldsymbol{w}_1\| \|\boldsymbol{w}_2\|$$

$$= (1 - \alpha - \beta\cos(\theta)) \left( \|\boldsymbol{w}_1\|^2 + \|\boldsymbol{w}_2\|^2 \right) + 2(\cos(\theta) - \beta - \alpha\cos(\theta)) \|\boldsymbol{w}_1\| \|\boldsymbol{w}_2\|$$

To show this quantity is nonnegative, we only need to prove that $1 - \alpha - \beta\cos(\theta) \geq |\cos(\theta) - \beta - \alpha\cos(\theta)|$. Since $\mu \leq 1$, we have $1 - \frac{\mu\theta}{2\pi} \geq 0$, which means $\beta \geq 0$. Note that $\alpha + \beta = \left( 1 - \frac{\mu\theta}{2\pi} + \frac{\mu\sin(\theta)}{2\pi} \right)^2 \leq 1$. We observe that

$$1 - \alpha - \beta\cos(\theta) - (\cos(\theta) - \beta - \alpha\cos(\theta))$$
$$= (1 - \cos(\theta))(1 + \beta - \alpha) \geq 0,$$

and

$$1 - \alpha - \beta\cos(\theta) + (\cos(\theta) - \beta - \alpha\cos(\theta))$$
$$= (1 + \cos(\theta))(1 - \beta - \alpha) \geq 0.$$

Combining both inequalities, we obtain that $1 - \alpha - \beta\cos(\theta) \geq |\cos(\theta) - \beta - \alpha\cos(\theta)|$. This finishes the proof.

### E.1.2 PROOF OF THE CORRELATION INEQUALITY IN THE POPULATION CASE (LEMMA 4)

As mentioned in the overview section, we define

$$h(\boldsymbol{w}_1, \boldsymbol{w}_2, \boldsymbol{a}) = \langle \boldsymbol{W} - \boldsymbol{W}^*, \nabla_{\boldsymbol{W}} \mathcal{L}(\boldsymbol{W}) \rangle - \alpha \|\boldsymbol{W} - \boldsymbol{W}^*\|_F^2.$$

Using the calculations in Appendix A.4, we can write it equivalently as

$$h(\boldsymbol{w}_1, \boldsymbol{w}_2, \boldsymbol{a}) = \underbrace{\left( \frac{1}{2} - \alpha \right) \|\boldsymbol{w}_1\|^2 + \left( \frac{1}{2} - \alpha \right) \|\boldsymbol{w}_2\|^2 - \left( \frac{(\pi - \theta)\cos\theta + \sin\theta}{\pi} \right) \|\boldsymbol{w}_1\| \|\boldsymbol{w}_2\|}_{:=g(\boldsymbol{w}_1, \boldsymbol{w}_2)}$$

$$\underbrace{- \left( \frac{3\pi - \sin\theta \frac{\|\boldsymbol{w}_2\|}{\|\boldsymbol{w}_1\|} - \theta}{2\pi} + 2\alpha \right) \boldsymbol{w}_1^T \boldsymbol{a}}_{:=\gamma_1} + \underbrace{\left( \frac{3\pi - \sin\theta \frac{\|\boldsymbol{w}_1\|}{\|\boldsymbol{w}_2\|} - \theta}{2\pi} - 2\alpha \right) \boldsymbol{w}_2^T \boldsymbol{a}}_{:=-\gamma_2}$$

$$+ (1 - 2\alpha) \|\boldsymbol{a}\|^2$$

$$= g(\boldsymbol{w}_1, \boldsymbol{w}_2) - (\gamma_1 \boldsymbol{w}_1 + \gamma_2 \boldsymbol{w}_2)^T \boldsymbol{a} + (1 - 2\alpha) \|\boldsymbol{a}\|^2$$

$$\geq g(\boldsymbol{w}_1, \boldsymbol{w}_2) - \|\gamma_1 \boldsymbol{w}_1 + \gamma_2 \boldsymbol{w}_2\| \|\boldsymbol{a}\| + (1 - 2\alpha) \|\boldsymbol{a}\|^2.$$

Noting that the expression above is quadratic in $\|\boldsymbol{a}\|$, we compute $\tilde{h}(\boldsymbol{w}_1, \boldsymbol{w}_2) = \min_{\boldsymbol{a}} h(\boldsymbol{w}_1, \boldsymbol{w}_2, \boldsymbol{a})$ with the assumption that $\|\boldsymbol{w}_1 + \boldsymbol{w}_2\| \leq \frac{1}{2} \|\boldsymbol{a}\|$. The choice of $\|\boldsymbol{a}\|$ that minimizes the expression depends on the location of parabola vertex. Then, we have

$$\|\boldsymbol{a}_{\min}\| = \begin{cases} \frac{\|\gamma_1 \boldsymbol{w}_1 + \gamma_2 \boldsymbol{w}_2\|}{2(1 - 2\alpha)}, & \text{if } \quad 2\|\boldsymbol{w}_1 + \boldsymbol{w}_2\| \leq \frac{\|\gamma_1 \boldsymbol{w}_1 + \gamma_2 \boldsymbol{w}_2\|}{2(1 - 2\alpha)} \\ 2\|\boldsymbol{w}_1 + \boldsymbol{w}_2\|, & \text{otherwise} \end{cases}.$$

Plugging it in, we get

$$\tilde{h}(\boldsymbol{w}_1, \boldsymbol{w}_2) = \begin{cases} g(\boldsymbol{w}_1, \boldsymbol{w}_2) - \frac{\|\gamma_1 \boldsymbol{w}_1 + \gamma_2 \boldsymbol{w}_2\|^2}{4(1 - 2\alpha)}, & \text{if } \quad 2\|\boldsymbol{w}_1 + \boldsymbol{w}_2\| \leq \frac{\|\gamma_1 \boldsymbol{w}_1 + \gamma_2 \boldsymbol{w}_2\|}{2(1 - 2\alpha)} \\ g(\boldsymbol{w}_1, \boldsymbol{w}_2) - 2\|\gamma_1 \boldsymbol{w}_1 + \gamma_2 \boldsymbol{w}_2\| \|\boldsymbol{w}_1 + \boldsymbol{w}_2\| + 4(1 - 2\alpha)\|\boldsymbol{w}_1 + \boldsymbol{w}_2\|^2, & \text{otherwise} \end{cases}.$$

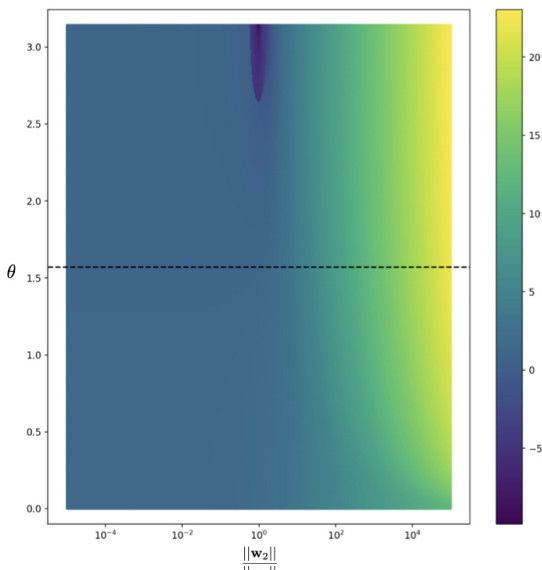

Figure 6: $\langle \boldsymbol{W} - \boldsymbol{W}^*, \nabla_{\boldsymbol{W}} \mathcal{L}(\boldsymbol{W}) \rangle - \alpha \|\boldsymbol{W} - \boldsymbol{W}^*\|_F^2$ **is non-negative**. We set $\alpha = 0.3$ and draw $\frac{1}{\|\boldsymbol{w}_1\|}\tilde{h}(\boldsymbol{w}_1, \boldsymbol{w}_2)$ for $\theta \in [0, \pi]$ and $\frac{\|\boldsymbol{w}_2\|}{\|\boldsymbol{w}_1\|} \in [10^{-5}, 10^5]$. Noting that the color bar is in log-scale, we show that $\frac{1}{\|\boldsymbol{w}_1\|}\tilde{h}(\boldsymbol{w}_1, \boldsymbol{w}_2)$ is non-negative everywhere in the domain.

Since we are only interested in the sign of $\tilde{h}$, we can take $\|\boldsymbol{w}_1\|$ outside. Defining the norm ratio $r$ as $r = \frac{\|\boldsymbol{w}_2\|}{\|\boldsymbol{w}_1\|}$, it is clear that the expression $\frac{1}{\|\boldsymbol{w}_1\|}\tilde{h}(\boldsymbol{w}_1, \boldsymbol{w}_2)$ is a function of only $\theta$ and $r$. To complete the proof, in Figure 6, we set $\alpha = 0.3$ and draw $\frac{1}{\|\boldsymbol{w}_1\|}\tilde{h}(\boldsymbol{w}_1, \boldsymbol{w}_2)$ as a 2D plot for a wide range of $r$ and $\theta$ empirically. The plot demonstrates that $\tilde{h}$ is non-negative everywhere. This finishes the proof of Lemma 4.

### E.1.3 PROOF OF GRADIENT SMOOTHNESS TOWARDS THE GLOBAL OPTIMA IN THE POPULATION CASE (LEMMA 5)

We first show that

$$\|\sin\theta\|\boldsymbol{w}_2\|\bar{\boldsymbol{w}}_1 + (\pi - \theta)\,\boldsymbol{w}_2\| \leq \pi\,\|\boldsymbol{w}_1 + \boldsymbol{w}_2\|. \tag{10}$$

Note that $0 \leq \theta \leq \pi$ since it is the angle between $\boldsymbol{w}_1$ and $\boldsymbol{w}_2$. We proceed by case analysis on the value of $\theta$. When $0 \leq \theta < \frac{\pi}{2}$, we have

$$
\begin{aligned}
\|\sin\theta\|\boldsymbol{w}_2\|\bar{\boldsymbol{w}}_1 + (\pi - \theta)\,\boldsymbol{w}_2\| &\overset{(a)}{\leq} \|\sin\theta\|\boldsymbol{w}_2\|\bar{\boldsymbol{w}}_1\| + \|(\pi - \theta)\,\boldsymbol{w}_2\| \\
&= \sin\theta\|\boldsymbol{w}_2\| + (\pi - \theta)\,\|\boldsymbol{w}_2\| \\
&\overset{(b)}{\leq} \pi\|\boldsymbol{w}_2\| \\
&\overset{(c)}{\leq} \pi\|\boldsymbol{w}_1 + \boldsymbol{w}_2\|.
\end{aligned}
$$

In Inequality (a) we use the triangle inequality. Inequality (b) follows from the fact that $\sin\theta \leq \theta$ when $\theta \geq 0$. Inequality (c) follows from the fact that $\theta \leq \frac{\pi}{2}$.

When $\theta \geq \frac{\pi}{2}$, we observe that

$$
\begin{aligned}
\|\sin\theta\|\boldsymbol{w}_2\|\bar{\boldsymbol{w}}_1 + (\pi - \theta)\,\boldsymbol{w}_2\| &\overset{(a)}{\leq} \|\sin\theta\|\boldsymbol{w}_2\|\bar{\boldsymbol{w}}_1\| + \|(\pi - \theta)\,\boldsymbol{w}_2\| \\
&= \sin\theta\|\boldsymbol{w}_2\| + (\pi - \theta)\,\|\boldsymbol{w}_2\| \\
&= \left(1 + \frac{\pi - \theta}{\sin\theta}\right)\sin\theta\,\|\boldsymbol{w}_2\| \\
&\overset{(b)}{\leq} \left(1 + \frac{\pi - \theta}{\sin\theta}\right)\|\boldsymbol{w}_1 + \boldsymbol{w}_2\| \\
&\overset{(c)}{\leq} \left(1 + \frac{\pi}{2}\right)\|\boldsymbol{w}_1 + \boldsymbol{w}_2\| \\
&\overset{(d)}{\leq} \pi\|\boldsymbol{w}_1 + \boldsymbol{w}_2\|.
\end{aligned}
$$

In Inequality (a) we use the triangle inequality. Inequality (b) follows from the fact that $\boldsymbol{w}_1 + \boldsymbol{w}_2$ has a component with magnitude $\sin\theta\,\|\boldsymbol{w}_2\|$ perpendicular to $\boldsymbol{w}_1$. Inequality (c) follows since $\frac{\pi - \theta}{\sin\theta}$ attains its maximum at $\theta = \frac{\pi}{2}$ when restricted to the range $\theta \geq \frac{\pi}{2}$. Finally, (d) follows because $1 \leq \frac{\pi}{2}$. This finishes the proof of Ineq. 10. Note that due to symmetry we get the following as a corollary:

$$
\|\sin\theta\|\boldsymbol{w}_1\|\bar{\boldsymbol{w}}_2 + (\pi - \theta)\,\boldsymbol{w}_1\| \leq \pi\,\|\boldsymbol{w}_1 + \boldsymbol{w}_2\|. \tag{11}
$$

Now recall the population gradient in Eq. 8. The individual rows, i.e. gradient with respect to $\boldsymbol{w}_1$ or $\boldsymbol{w}_2$, can be written separately as:

$$
\nabla_{\boldsymbol{w}_1}\mathcal{L} = -\frac{\boldsymbol{a}}{2} + \frac{1}{2\pi}\left(\pi\boldsymbol{w}_1 - \sin\theta\|\boldsymbol{w}_2\|\bar{\boldsymbol{w}}_1 - (\pi - \theta)\,\boldsymbol{w}_2\right)
$$

$$
\nabla_{\boldsymbol{w}_2}\mathcal{L} = \frac{\boldsymbol{a}}{2} - \frac{1}{2\pi}\left((\pi - \theta)\,\boldsymbol{w}_1 + \sin\theta\|\boldsymbol{w}_1\|\bar{\boldsymbol{w}}_2 - \pi\boldsymbol{w}_2\right).
$$

Using Ineq. 10, we can write

$$
\begin{aligned}
\|\nabla_{\boldsymbol{w}_1}\mathcal{L}\|^2 &= \left\|-\frac{\boldsymbol{a}}{2} + \frac{1}{2\pi}\left(\pi\boldsymbol{w}_1 - \sin\theta\|\boldsymbol{w}_2\|\bar{\boldsymbol{w}}_1 - (\pi - \theta)\,\boldsymbol{w}_2\right)\right\|^2 \\
&= \left\|\frac{\boldsymbol{w}_1 - \boldsymbol{a}}{2} - \frac{1}{2\pi}\left(\sin\theta\|\boldsymbol{w}_2\|\bar{\boldsymbol{w}}_1 + (\pi - \theta)\,\boldsymbol{w}_2\right)\right\|^2 \\
&\leq 2\left\|\frac{\boldsymbol{w}_1 - \boldsymbol{a}}{2}\right\|^2 + 2\left\|\frac{1}{2\pi}\left(\sin\theta\|\boldsymbol{w}_2\|\bar{\boldsymbol{w}}_1 + (\pi - \theta)\,\boldsymbol{w}_2\right)\right\|^2 \\
&\leq \frac{1}{2}\|\boldsymbol{w}_1 - \boldsymbol{a}\|^2 + \frac{1}{2}\|\boldsymbol{w}_1 + \boldsymbol{w}_2\|^2 \\
&= \frac{1}{2}\|\boldsymbol{w}_1 - \boldsymbol{a}\|^2 + \frac{1}{2}\|\boldsymbol{w}_1 - \boldsymbol{a} + \boldsymbol{a} + \boldsymbol{w}_2\|^2 \\
&\leq \frac{3}{2}\|\boldsymbol{w}_1 - \boldsymbol{a}\|^2 + \|\boldsymbol{w}_2 + \boldsymbol{a}\|^2.
\end{aligned}
$$

Similarly, using Eq. 11 on the gradient for $\boldsymbol{w}_2$, we get

$$
\|\nabla_{\boldsymbol{w}_2}\mathcal{L}\|^2 \leq \frac{3}{2}\|\boldsymbol{w}_2 + \boldsymbol{a}\|^2 + \|\boldsymbol{w}_1 - \boldsymbol{a}\|^2.
$$

Combining these, we obtain

$$
\|\nabla_{\boldsymbol{w}_1}\mathcal{L}\|^2 + \|\nabla_{\boldsymbol{w}_2}\mathcal{L}\|^2 \leq \frac{5}{2}\left(\|\boldsymbol{w}_1 - \boldsymbol{a}\|^2 + \|\boldsymbol{w}_2 + \boldsymbol{a}\|^2\right).
$$

This completes the proof of Lemma 5 for $\beta = \sqrt{\frac{5}{2}}$.

### E.2 KEY IDENTITIES AND PROOFS IN THE EMPIRICAL SETTING

#### E.2.1 GUARANTEES FOR THE FIRST ITERATION

In this section, we will show that with high probability, after the first step of gradient descent with step size 2, $\boldsymbol{W}$ is sufficiently close to the global optimum. Specifically, we have the following Lemma:

**Lemma 7** *Assume that $\boldsymbol{w}_1^{(0)} \sim \mathcal{N}\left(\boldsymbol{0}, \sigma_1^2 \boldsymbol{I}_d\right)$ and $\boldsymbol{w}_2^{(0)} \sim \mathcal{N}\left(\boldsymbol{0}, \sigma_2^2 \boldsymbol{I}_d\right)$ with the standard deviations obeying $\sqrt{\sigma_1^2 + \sigma_2^2}\sqrt{d} \leq c_6 \|\boldsymbol{a}\|$. Furthermore, assume $n \geq \frac{C}{\epsilon^2}d$. After one step of the gradient descent with step size $\mu = 2$, we have*

$$\|\boldsymbol{w}_1^{(1)} - \boldsymbol{a}\|^2 + \|\boldsymbol{w}_2^{(1)} + \boldsymbol{a}\|^2 \leq \epsilon^2,$$

*with probability at least $1 - Ce^{-cd}$. Here $c, C$ are fixed positive numerical constants.*

To prove the Lemma, it suffices to show $\|\boldsymbol{w}_1^{(1)} - \boldsymbol{a}\|^2 \leq \frac{\epsilon^2}{2}$ as proving $\|\boldsymbol{w}_2^{(1)} + \boldsymbol{a}\|^2 \leq \frac{\epsilon^2}{2}$ is the same by a symmetry argument. As a reminder the empirical gradient of $\boldsymbol{w}_1$ is

$$\nabla_{\boldsymbol{w}_1}\widehat{\mathcal{L}} = \frac{1}{n}\sum_{i=1}^{n} r\left(\boldsymbol{x}_i\right)\phi'\left(\boldsymbol{x}_i^T \boldsymbol{w}_1\right)\boldsymbol{x}_i.$$

we have

$$\begin{aligned}
\boldsymbol{w}_1^{(1)} =& \boldsymbol{w}_1^{(0)} - 2\nabla_{\boldsymbol{w}_1}\widehat{\mathcal{L}} \\
=& \boldsymbol{w}_1^{(0)} - \frac{2}{n}\sum_{i=1}^{n} r\left(\boldsymbol{x}_i\right)\phi'\left(\boldsymbol{x}_i^T \boldsymbol{w}_1^{(0)}\right)\boldsymbol{x}_i \\
=& \boldsymbol{w}_1^{(0)} - \frac{2}{n}\sum_{i=1}^{n} \left(\phi\left(\boldsymbol{x}_i^T \boldsymbol{w}_1^{(0)}\right) - \phi\left(\boldsymbol{x}_i^T \boldsymbol{w}_2^{(0)}\right) - \boldsymbol{a}^T \boldsymbol{x}_i\right)\phi'\left(\boldsymbol{x}_i^T \boldsymbol{w}_1^{(0)}\right)\boldsymbol{x}_i \\
=& \boldsymbol{w}_1^{(0)} - \frac{2}{n}\sum_{i=1}^{n} \left(\phi\left(\boldsymbol{x}_i^T \boldsymbol{w}_1^{(0)}\right) - \phi\left(\boldsymbol{x}_i^T \boldsymbol{w}_2^{(0)}\right) - \boldsymbol{a}^T \boldsymbol{x}_i\right)\mathbf{1}_{\{\boldsymbol{x}_i^T \boldsymbol{w}_1^{(0)} \geq 0\}}\boldsymbol{x}_i \\
=& \boldsymbol{w}_1^{(0)} - \frac{2}{n}\sum_{i=1}^{n} \left(\phi\left(\boldsymbol{x}_i^T \boldsymbol{w}_1^{(0)}\right) - \phi\left(\boldsymbol{x}_i^T \boldsymbol{w}_2^{(0)}\right) - \phi\left(\boldsymbol{x}_i^T \boldsymbol{a}\right) + \phi\left(\boldsymbol{x}_i^T(-\boldsymbol{a})\right)\right)\mathbf{1}_{\{\boldsymbol{x}_i^T \boldsymbol{w}_1^{(0)} \geq 0\}}\boldsymbol{x}_i.
\end{aligned}$$

To prove this vector is close to $\boldsymbol{a}$, we first show the following Lemma:

**Lemma 8** *$\{\boldsymbol{x}_i\}_{i=1}^{n}$ are i.i.d. Gaussian random vectors distributed as $\mathcal{N}(\boldsymbol{0}, \boldsymbol{I}_d)$. Furthermore, assume*

$$n \geq \frac{C}{\delta^2}d.$$

*For any fixed vector $\boldsymbol{p}, \boldsymbol{q}$, we have*

$$\left\|\frac{1}{n}\sum_{i=1}^{n} \mathbf{1}_{\{\boldsymbol{x}_i^T \boldsymbol{p} \geq 0\}}\boldsymbol{x}_i\phi(\boldsymbol{x}_i^T \boldsymbol{q}) - \mathbb{E}\left[\mathbf{1}_{\{\boldsymbol{x}^T \boldsymbol{p} \geq 0\}}\boldsymbol{x}\phi(\boldsymbol{x}^T \boldsymbol{q})\right]\right\| \leq \delta\|\boldsymbol{q}\|$$

*with probability at least $1 - 6e^{-\gamma d}$. Here, $\phi : \mathbb{R} \mapsto \mathbb{R}$ is the ReLU function.*

Without loss of generality, assume that $\boldsymbol{p} = \boldsymbol{e}_1$, $\text{span}(\boldsymbol{p}, \boldsymbol{q}) = (\boldsymbol{e}_1, \boldsymbol{e}_2)$ and $\|\boldsymbol{q}\| = 1$. Consider a random variable $Y = \mathbf{1}_{\{\boldsymbol{x}^T \boldsymbol{p} \geq 0\}}\boldsymbol{x}\phi(\boldsymbol{x}^T \boldsymbol{q})$. Assume that $\boldsymbol{x} = [g_1, g_2, \ldots, g_d]^T$. The last $d - 2$ entries of $Y$, defined as $Y_2$, is in the form of $\mathbf{1}_{\{g_1 \geq 0\}}\phi(\boldsymbol{x}^T \boldsymbol{q})[g_3, \ldots, g_d]^T$. We observe that $[g_3, \ldots, g_d]^T$ is independent with $\mathbf{1}_{\{g_1 \geq 0\}}\phi(\boldsymbol{x}^T \boldsymbol{q})$. For fixed $\mathbf{1}_{\{\boldsymbol{x}_i^T \boldsymbol{p} \geq 0\}}\phi(\boldsymbol{x}_i^T \boldsymbol{q}), i = 1, 2 \ldots, n$, $\frac{1}{n}\sum_{i=1}^{n} \mathbf{1}_{\{\boldsymbol{x}_i^T \boldsymbol{p} \geq 0\}}\boldsymbol{x}_i\phi(\boldsymbol{x}_i^T \boldsymbol{q})$ is a weighted sum of independent Gaussian random vectors, which means it is also a Gaussian random vector, the variance of which is

$$\frac{1}{n^2}\sum_{i=1}^{n}(\mathbf{1}_{\{\boldsymbol{x}_i^T \boldsymbol{p} \geq 0\}}\phi(\boldsymbol{x}_i^T \boldsymbol{q}))^2 \cdot \boldsymbol{I}_{d-2}.$$

Denote $\sigma = \frac{1}{n^2}\sum_{i=1}^{n}(\mathbf{1}_{\{\boldsymbol{x}_i^T \boldsymbol{p} \geq 0\}}\phi(\boldsymbol{x}_i^T \boldsymbol{q}))^2$. we have that

$$\Pr\left[\|Y_2\| \geq 2\sigma\sqrt{d}\right] \leq e^{-d/2}.$$

Let $Z = \mathbf{1}_{\{x^T p \geq 0\}} \phi(x^T q)$. Note that $Z$ is a sub-gaussian random variable, which implies that $Z^2$ is sub-exponential. Indeed, the sub-exponential norm of $Z^2$, defined as

$$\|Z^2\|_{\psi_1} = \sup_{p \geq 1} p^{-1} (\mathbb{E}|Z^2|^p)^{1/p},$$

is upper bounded by some numerical constant $\gamma_1$. By the Bernstein-type inequality, we have that

$$\Pr\left[\left|\sum_{i=1}^n Z_i^2 - \mathbb{E}[Z_i^2]\right| \geq t\right] \leq 2\exp\left[-c \min\left(\frac{t^2}{n\gamma_1^2}, \frac{t}{\gamma_1}\right)\right].$$

By union bound, we have that with probability at least $1 - 4e^{-\gamma d}$, $\|Y_2\| \leq \frac{\delta}{2}$. Similarly, the first two entries of $Y$ is also sub-exponential, the sub-exponential norm of which is upper bounded by some numerical constant $\gamma_2$. By the Bernstein-type inequality, we have that with probability at least $1 - 2e^{-\gamma d}$, $\|Y_1\| \leq \frac{\delta}{2}$. By union bound, we have that with probability at least $1 - 6e^{-\gamma d}$, $\|Y\| \leq \|Y_1\| + \|Y_2\| \leq \delta$, which finishes the proof of the Lemma.

We use this Lemma with $p = w_1^{(0)}$ and $q = w_1^{(0)}, w_2^{(0)}, -a, a$ and sum the four inequalities together. Combining with the triangle inequality, we obtain that

$$\left\|\frac{1}{n}\sum_{i=1}^n \left(\phi\left(x_i^T w_1^{(0)}\right) - \phi\left(x_i^T w_2^{(0)}\right) - a^T x_i\right) \mathbf{1}_{\{x_i^T w_1^{(0)} \geq 0\}} x_i \right.$$

$$\left. - \mathbb{E}\left[\left(\phi\left(x^T w_1^{(0)}\right) - \phi\left(x^T w_2^{(0)}\right) - a^T x\right) \mathbf{1}_{\{x^T w_1^{(0)} \geq 0\}} x\right]\right\|$$

$$\leq \delta(\|w_1^{(0)}\| + \|w_2^{(0)}\| + 2\|a\|), \tag{12}$$

with probability at least $1 - 24e^{-\gamma d}$. We compute that

$$\mathbb{E}\left[\left(\phi\left(x^T w_1^{(0)}\right) - \phi\left(x^T w_2^{(0)}\right) - a^T x\right) \mathbf{1}_{\{x^T w_1^{(0)} \geq 0\}} x\right]$$

$$= \mathbb{E}\left[\left(\phi\left(x^T w_1^{(0)}\right) - \phi\left(x^T w_2^{(0)}\right)\right) \mathbf{1}_{\{x^T w_1^{(0)} \geq 0\}} x\right] - \frac{a}{2}.$$

This implies that

$$w_1^{(1)} - a = w_1^{(0)} - \frac{2}{n}\sum_{i=1}^n \left(\phi\left(x_i^T w_1^{(0)}\right) - \phi\left(x_i^T w_2^{(0)}\right) - a^T x_i\right) \mathbf{1}_{\{x_i^T w_1^{(0)} \geq 0\}} x_i - a$$

$$= w_1^{(0)} - a - 2\left(\frac{1}{n}\sum_{i=1}^n \left(\phi\left(x_i^T w_1^{(0)}\right) - \phi\left(x_i^T w_2^{(0)}\right) - a^T x_i\right) \mathbf{1}_{\{x_i^T w_1^{(0)} \geq 0\}} x_i\right.$$

$$\left. - \mathbb{E}\left[\left(\phi\left(x^T w_1^{(0)}\right) - \phi\left(x^T w_2^{(0)}\right) - a^T x\right) \mathbf{1}_{\{x^T w_1^{(0)} \geq 0\}} x\right]\right)$$

$$- 2\mathbb{E}\left[\left(\phi\left(x^T w_1^{(0)}\right) - \phi\left(x^T w_2^{(0)}\right) - a^T x\right) \mathbf{1}_{\{x^T w_1^{(0)} \geq 0\}} x\right]$$

$$= w_1^{(0)} - 2\left(\frac{1}{n}\sum_{i=1}^n \left(\phi\left(x_i^T w_1^{(0)}\right) - \phi\left(x_i^T w_2^{(0)}\right) - a^T x_i\right) \mathbf{1}_{\{x_i^T w_1^{(0)} \geq 0\}} x_i\right.$$

$$\left. - \mathbb{E}\left[\left(\phi\left(x^T w_1^{(0)}\right) - \phi\left(x^T w_2^{(0)}\right) - a^T x\right) \mathbf{1}_{\{x^T w_1^{(0)} \geq 0\}} x\right]\right)$$

$$- 2\mathbb{E}\left[\left(\phi\left(x^T w_1^{(0)}\right) - \phi\left(x^T w_2^{(0)}\right)\right) \mathbf{1}_{\{x^T w_1^{(0)} \geq 0\}} x\right].$$

Plugging in the Ineq.12 and using the triangle inequality, we obtain that

$$\|w_1^{(1)} - a\| \leq \|w_1^{(0)}\| + 2\delta(\|w_1^{(0)}\| + \|w_2^{(0)}\| + 2\|a\|)$$

$$+ 2\left\|\mathbb{E}\left[\left(\phi\left(x^T w_1^{(0)}\right) - \phi\left(x^T w_2^{(0)}\right)\right) \mathbf{1}_{\{x^T w_1^{(0)} \geq 0\}} x\right]\right\|.$$

By computation, we have that

$$\left\| \mathbb{E}\left[ \left( \phi\left( \boldsymbol{x}^T \boldsymbol{w}_1^{(0)} \right) - \phi\left( \boldsymbol{x}^T \boldsymbol{w}_2^{(0)} \right) \right) \mathbf{1}_{\{\boldsymbol{x}^T \boldsymbol{w}_1^{(0)} \geq 0\}} \boldsymbol{x} \right] \right\| \leq \|\boldsymbol{w}_1^{(0)}\| + \|\boldsymbol{w}_2^{(0)}\|,$$

which implies that

$$\|\boldsymbol{w}_1^{(1)} - \boldsymbol{a}\| \leq (3 + 2\delta)(\|\boldsymbol{w}_1^{(0)}\| + \|\boldsymbol{w}_2^{(0)}\|) + 4\delta\|\boldsymbol{a}\|.$$

Remind that $\boldsymbol{w}_1^{(0)} \sim \mathcal{N}\left(\mathbf{0}, \sigma_1^2 \boldsymbol{I}_d\right)$ and $\boldsymbol{w}_2^{(0)} \sim \mathcal{N}\left(\mathbf{0}, \sigma_2^2 \boldsymbol{I}_d\right)$ with the standard deviations obeying $\sqrt{\sigma_1^2 + \sigma_2^2}\sqrt{d} \leq c_6 \|\boldsymbol{a}\|$, it is well-known that with probability at least $1 - 2e^{-\frac{d}{8}}$, $\|\boldsymbol{w}_1^{(0)}\| + \|\boldsymbol{w}_2^{(0)}\| \leq 2c_6\|\boldsymbol{a}\|$. Let $c_6 = \frac{1}{40}\epsilon, \delta \leq \frac{1}{40}\epsilon$. For any $\epsilon \leq 1$, we have

$$\begin{aligned}
\|\boldsymbol{w}_1^{(1)} - \boldsymbol{a}\| &\leq (3 + 2\delta)(\|\boldsymbol{w}_1^{(0)}\| + \|\boldsymbol{w}_2^{(0)}\|) + 4\delta\|\boldsymbol{a}\| \\
&\leq (3 + 2\delta) \cdot 2c_6\|\boldsymbol{a}\| + 4\delta\|\boldsymbol{a}\| \\
&\leq \frac{\epsilon}{\sqrt{2}},
\end{aligned}$$

with probability at least $1 - 24e^{-\gamma d} - 2e^{-\frac{d}{8}}$. Similarly, we have $\|\boldsymbol{w}_2^{(1)} - \boldsymbol{a}\| \leq \frac{\epsilon}{\sqrt{2}}$ with probability at least $1 - 24e^{-\gamma d} - 2e^{-\frac{d}{8}}$. By union bound, we conclude that $\|\boldsymbol{w}_1^{(1)} - \boldsymbol{a}\|^2 + \|\boldsymbol{w}_2^{(1)} + \boldsymbol{a}\|^2 \leq \epsilon^2$ with probability at least $1 - Ce^{-cd}$ with some positive numerical constant $c, C$. This finishes the proof of the Lemma 7.

### E.2.2 Proof of the Correlation Inequality in the Empirical Case

Without loss of generality, assume that $\|\boldsymbol{a}\| = 1$. For simplicity of notations, define $\phi(x) = ReLU(x)$. We observe that

$$\begin{aligned}
&\left\langle \nabla_{\boldsymbol{w}_1}\widehat{\mathcal{L}}, \boldsymbol{w}_1 - \boldsymbol{a} \right\rangle + \left\langle \nabla_{\boldsymbol{w}_2}\widehat{\mathcal{L}}, \boldsymbol{w}_2 + \boldsymbol{a} \right\rangle \\
&= \frac{1}{n}\sum_{i=1}^{n} r(\boldsymbol{x}_i)\phi'(\boldsymbol{w}_1^T\boldsymbol{x}_i)\boldsymbol{x}_i^T(\boldsymbol{w}_1 - \boldsymbol{a}) - \frac{1}{n}\sum_{i=1}^{n} r(\boldsymbol{x}_i)\phi'(\boldsymbol{w}_2^T\boldsymbol{x}_i)\boldsymbol{x}_i^T(\boldsymbol{w}_2 + \boldsymbol{a}) \\
&= \frac{1}{n}\sum_{i=1}^{n} r(\boldsymbol{x}_i)\left(\phi(\boldsymbol{x}_i^T\boldsymbol{w}_1) - \phi(\boldsymbol{x}_i^T\boldsymbol{w}_2)\right) - \frac{1}{n}\sum_{i=1}^{n} r(\boldsymbol{x}_i)(\boldsymbol{a}^T\boldsymbol{x}_i)\left(\phi'(\boldsymbol{w}_1^T\boldsymbol{x}_i) + \phi'(\boldsymbol{w}_2^T\boldsymbol{x}_i)\right) \\
&= \frac{1}{n}\sum_{i=1}^{n} r^2(\boldsymbol{x}_i) + \frac{1}{n}\sum_{i=1}^{n} r(\boldsymbol{x}_i)(\boldsymbol{a}^T\boldsymbol{x}_i)\left(1 - \left(\phi'(\boldsymbol{w}_1^T\boldsymbol{x}_i) + \phi'(\boldsymbol{w}_2^T\boldsymbol{x}_i)\right)\right).
\end{aligned}$$

By the Cauchy-Schwarz inequality, we have

$$\begin{aligned}
LHS &\geq \frac{1}{n}\sum_{i=1}^{n} r^2(\boldsymbol{x}_i) - \frac{1}{n}\sqrt{\sum_{i=1}^{n} r^2(\boldsymbol{x}_i)\sum_{i=1}^{n}(\boldsymbol{a}^T\boldsymbol{x}_i)^2\left(1 - \left(\phi'(\boldsymbol{w}_1^T\boldsymbol{x}_i) + \phi'(\boldsymbol{w}_2^T\boldsymbol{x}_i)\right)\right)^2} \\
&\geq \frac{1}{n}\sum_{i=1}^{n} r^2(\boldsymbol{x}_i) - \frac{1}{n}\sqrt{\sum_{i=1}^{n} r^2(\boldsymbol{x}_i)\sum_{i=1}^{n}(\boldsymbol{a}^T\boldsymbol{x}_i)^2\left(1 - \operatorname{sgn}(\boldsymbol{a}^T\boldsymbol{x}_i)\operatorname{sgn}(\boldsymbol{w}_1^T\boldsymbol{x}_i) + 1 - \operatorname{sgn}(-\boldsymbol{a}^T\boldsymbol{x}_i)\operatorname{sgn}(\boldsymbol{w}_2^T\boldsymbol{x}_i)\right)}.
\end{aligned}$$

where the last inequality is derived by decomposing $1$ as $\phi'\left(\boldsymbol{a}^T\boldsymbol{x}_i\right) + \phi'\left(-\boldsymbol{a}^T\boldsymbol{x}_i\right)$, using the identity $\operatorname{sgn}(\cdot) + 1 = 2\phi'(\cdot)$, and the fact that $(x + y)^2 \leq 2(x^2 + y^2)$.

The following inequality is established within the proof of (Soltanolkotabi, 2019, Lemma 7.17):

**Lemma 9** *Assume the measurement vectors $\{\boldsymbol{x}_i\}_{i=1}^{n}$ are i.i.d. Gaussian random vectors distributed as $\mathcal{N}(\mathbf{0}, \boldsymbol{I}_d)$, where $n \geq \frac{cd}{\delta}$. Furthermore, assume $\boldsymbol{a}$ is a fixed vector independent of the measurement vectors such that $\|\boldsymbol{a}\| = 1$. Define the set $E(\epsilon) = \{\boldsymbol{w} \in \mathbb{R}^d \mid \|\boldsymbol{w} - \boldsymbol{a}\| \leq \epsilon\}$. Assume that $\delta \leq \frac{\epsilon}{100}$.*

*We have*

$$\sqrt{\frac{1}{n}\sum_{i=1}^{n}(\boldsymbol{a}^T\boldsymbol{x}_i)^2\left(1-\operatorname{sgn}(\boldsymbol{a}^T\boldsymbol{x}_i)\operatorname{sgn}(\boldsymbol{w}^T\boldsymbol{x}_i)\right)}\leq\frac{\sqrt{2}}{1-\epsilon}\sqrt{\frac{1}{n}\sum_{i=1}^{n}\mathbf{1}_{\{(1-\epsilon)|\boldsymbol{x}_i^T\boldsymbol{a}|\leq|\boldsymbol{x}_i^T\boldsymbol{h}_\perp|\}}(\boldsymbol{h}_\perp^T\boldsymbol{x}_i)^2}$$

$$\leq\frac{\sqrt{2}}{1-\epsilon}\left(\delta+\sqrt{\frac{21}{20}}\epsilon\right)\|\boldsymbol{w}-\boldsymbol{a}\|$$

$$\leq\frac{3}{2}\cdot\frac{\epsilon}{1-\epsilon}\|\boldsymbol{w}-\boldsymbol{a}\|,$$

*holds for all $\boldsymbol{w}\in E(\epsilon)$ with probability at least $1-3e^{-\gamma n}$, where $\boldsymbol{h}_\perp$ is the part of $\boldsymbol{h}=\boldsymbol{w}+\boldsymbol{a}$ that is perpendicular to $\boldsymbol{a}$ and $\gamma$ is a fixed numerical constant.*

This allows us to proceed as follows:

$$LHS\geq\frac{1}{n}\sum_{i=1}^{n}r^2\left(\boldsymbol{x}_i\right)-\sqrt{\frac{1}{n}\sum_{i=1}^{n}r^2\left(\boldsymbol{x}_i\right)}\sqrt{\left(\frac{3}{2}\cdot\frac{\epsilon}{1-\epsilon}\|\boldsymbol{w}_1-\boldsymbol{a}\|\right)^2+\left(\frac{3}{2}\cdot\frac{\epsilon}{1-\epsilon}\|\boldsymbol{w}_2+\boldsymbol{a}\|\right)^2}$$

$$\geq\frac{1}{n}\sum_{i=1}^{n}r^2\left(\boldsymbol{x}_i\right)-\frac{1}{2}\left(\frac{1}{n}\sum_{i=1}^{n}r^2\left(\boldsymbol{x}_i\right)+\left(\frac{3}{2}\cdot\frac{\epsilon}{1-\epsilon}\|\boldsymbol{w}_1-\boldsymbol{a}\|\right)^2+\left(\frac{3}{2}\cdot\frac{\epsilon}{1-\epsilon}\|\boldsymbol{w}_2+\boldsymbol{a}\|\right)^2\right)$$

$$=\frac{1}{2n}\sum_{i=1}^{n}r^2\left(\boldsymbol{x}_i\right)-\frac{1}{2}\left(\frac{3}{2}\cdot\frac{\epsilon}{1-\epsilon}\right)^2\left(\|\boldsymbol{w}_1-\boldsymbol{a}\|^2+\|\boldsymbol{w}_2+\boldsymbol{a}\|^2\right),\tag{13}$$

holds for all $\boldsymbol{W}$ such that $\|\boldsymbol{W}-\boldsymbol{W}^*\|_F\leq\epsilon$ with probability at least $1-6e^{-\gamma n}$.

Next we will lower bound

$$\frac{1}{n}\sum_{i=1}^{n}r^2(\boldsymbol{x}_i)$$

To do this note that

$$\frac{1}{n}\sum_{i=1}^{n}r^2(\boldsymbol{x}_i)=\frac{1}{n}\sum_{i=1}^{n}\left(\mathbf{1}_{\{\boldsymbol{a}^T\boldsymbol{x}_i\geq0\}}r^2(\boldsymbol{x}_i)+\mathbf{1}_{\{\boldsymbol{a}^T\boldsymbol{x}_i<0\}}r^2(\boldsymbol{x}_i)\right).$$

We proceed by bounding the first term. The bound of the second term is the same by a symmetry argument. To bound the first term we proceed by the following chain of inequalities

$$\frac{1}{n}\sum_{i=1}^{n}\mathbf{1}_{\{\boldsymbol{a}^T\boldsymbol{x}_i\geq0\}}r^2(\boldsymbol{x}_i)\geq\frac{1}{n}\sum_{i=1}^{n}\mathbf{1}_{\{\boldsymbol{a}^T\boldsymbol{x}_i\geq0\}}r^2(\boldsymbol{x}_i)$$

$$\overset{(a)}{\geq}\frac{1}{2n}\sum_{i=1}^{n}\mathbf{1}_{\{\boldsymbol{a}^T\boldsymbol{x}_i\geq0\}}\left(\phi(\boldsymbol{w}_1^T\boldsymbol{x}_i)-\boldsymbol{a}^T\boldsymbol{x}_i\right)^2-\frac{1}{n}\sum_{i=1}^{n}\mathbf{1}_{\{\boldsymbol{a}^T\boldsymbol{x}_i\geq0\}}\phi^2(\boldsymbol{w}_2^T\boldsymbol{x}_i)$$

$$\overset{(b)}{\geq}\frac{1}{2n}\sum_{i=1}^{n}\mathbf{1}_{\{\boldsymbol{a}^T\boldsymbol{x}_i\geq0\}}\mathbf{1}_{\{\boldsymbol{w}_1^T\boldsymbol{x}_i\geq0\}}\left(\phi(\boldsymbol{w}_1^T\boldsymbol{x}_i)-\boldsymbol{a}^T\boldsymbol{x}_i\right)^2-\frac{1}{n}\sum_{i=1}^{n}\mathbf{1}_{\{\boldsymbol{a}^T\boldsymbol{x}_i\geq0\}}\phi^2(\boldsymbol{w}_2^T\boldsymbol{x}_i)$$

$$=\frac{1}{2n}\sum_{i=1}^{n}\mathbf{1}_{\{\boldsymbol{a}^T\boldsymbol{x}_i\geq0\}}\mathbf{1}_{\{\boldsymbol{w}_1^T\boldsymbol{x}_i\geq0\}}\left(\boldsymbol{h}_1^T\boldsymbol{x}_i\right)^2-\frac{1}{n}\sum_{i=1}^{n}\mathbf{1}_{\{\boldsymbol{a}^T\boldsymbol{x}_i\geq0\}}\phi^2(\boldsymbol{w}_2^T\boldsymbol{x}_i)$$

$$\overset{(c)}{\geq}c\|\boldsymbol{h}_1\|^2-\frac{1}{n}\sum_{i=1}^{n}\mathbf{1}_{\{\boldsymbol{a}^T\boldsymbol{x}_i\geq0\}}\phi^2(\boldsymbol{w}_2^T\boldsymbol{x}_i).\tag{14}$$

Here, (a) follows from $(x-y)^2\geq\frac{1}{2}x^2-y^2$, (b) from $1\geq\mathbf{1}_{\{\boldsymbol{w}_1^T\boldsymbol{x}_i\geq0\}}$, and (c) from the Lemma 11 proven in Appendix E.2.4 below. To bound the second term note that we can rewrite $\boldsymbol{w}_2$ as follows

$$\boldsymbol{w}_2=-(-\boldsymbol{w}_2^T\boldsymbol{a})\boldsymbol{a}+\boldsymbol{h}_{2,\perp},$$

where $h_{2,\perp}$ is the part of $h_2 = w_2 + a$ that is perpendicular to $a$. Now note that when $a^T x_i \geq 0$ and $\|h_2\| \leq \epsilon$ we have

$$w_2^T x_i = (-w_2^T a)(-a^T x_i) + h_{2,\perp}^T x_i \leq h_{2,\perp}^T x_i,$$

and

$$\sum_{i=1}^n \mathbf{1}_{\{a^T x_i \geq 0\}} \phi^2(w_2^T x_i) = \sum_{i=1}^n \mathbf{1}_{\{a^T x_i \geq 0\}} \mathbf{1}_{\{w_2^T x_i \geq 0\}} (w_2^T x_i)^2 \leq \sum_{i=1}^n \mathbf{1}_{\{a^T x_i \geq 0\}} \mathbf{1}_{\{w_2^T x_i \geq 0\}} (h_{2,\perp}^T x_i)^2.$$

(15)

Next note that when $a^T x_i \geq 0$ and $w_2^T x_i \geq 0$ we have

$$x_i^T h_{2,\perp} = x_i^T w_2 + (-w_2^T a)(a^T x_i) \geq (-w_2^T a)(a^T x_i) \geq (1 - \epsilon)(a^T x_i) \geq 0,$$

which implies that

$$|x_i^T h_{2,\perp}| \geq (1 - \epsilon)|a^T x_i|.$$

Thus we have

$$\mathbf{1}_{\{a^T x_i \geq 0\}} \mathbf{1}_{\{w_2^T x_i \geq 0\}} \leq \mathbf{1}_{\{(1-\epsilon)|x_i^T a| \leq |x_i^T h_{2,\perp}|\}}.$$

Plugging this inequality into (15) we conclude that

$$\frac{1}{n} \sum_{i=1}^n \mathbf{1}_{\{a^T x_i \geq 0\}} \phi^2(w_2^T x_i) \leq \frac{1}{n} \sum_{i=1}^n \mathbf{1}_{\{(1-\epsilon)|x_i^T a| \leq |x_i^T h_{2,\perp}|\}} (h_{2,\perp}^T x_i)^2.$$

Use Lemma 9 with $w = w_2$, we have that

$$\sqrt{\frac{1}{n} \sum_{i=1}^n \mathbf{1}_{\{(1-\epsilon)|x_i^T a| \leq |x_i^T h_{2,\perp}|\}} (h_{2,\perp}^T x_i)^2} \leq \left( \delta + \sqrt{\frac{21}{20}} \epsilon \right) \|w_2 + a\|$$

$$\leq 2\epsilon \|w_2 + a\|$$

holds for all $\|w_2 - a\| \leq \epsilon$ with probability at least $1 - 3e^{-\gamma n}$. Thus we have

$$\frac{1}{n} \sum_{i=1}^n \mathbf{1}_{\{a^T x_i \geq 0\}} \phi^2(w_2^T x_i) \leq \frac{1}{n} \sum_{i=1}^n \mathbf{1}_{\{(1-\epsilon)|x_i^T a| \leq |x_i^T h_{2,\perp}|\}} (h_{2,\perp}^T x_i)^2$$

$$\leq 4\epsilon^2 \|w_2 + a\|^2.$$

Plugging the latter into (14) we conclude that

$$\frac{1}{n} \sum_{i=1}^n \mathbf{1}_{\{a^T x_i \geq 0\}} r^2(x_i) \geq c \|h_1\|^2 - 4\epsilon^2 \|h_2\|^2$$

Similarly, we have

$$\frac{1}{n} \sum_{i=1}^n \mathbf{1}_{\{a^T x_i \leq 0\}} r^2(x_i) \geq c \|h_2\|^2 - 4\epsilon^2 \|h_1\|^2$$

Summing the latter two we conclude that

$$\frac{1}{n} \sum_{i=1}^n r^2(x_i) \geq (c - 4\epsilon^2) \left( \|h_1\|^2 + \|h_2\|^2 \right)$$

Finally, plugging in the above into (13) we conclude that

$$\langle \nabla \widehat{\mathcal{L}}(W), W - W^* \rangle \geq \left( \frac{c}{2} - 2\epsilon^2 - \frac{9\epsilon^2}{8(1-\epsilon)^2} \right) \|W - W^*\|_F^2$$

Thus, for $\epsilon$ a sufficiently small constant we have

$$\langle \nabla \widehat{\mathcal{L}}(W), W - W^* \rangle \geq \alpha \|W - W^*\|_F^2$$

holds with high probability with $\alpha = c/3$. This concludes the proof of the corelation inequality in the empirical case.

### E.2.3 PROOF OF GRADIENT SMOOTHNESS TOWARDS THE GLOBAL OPTIMA IN THE EMPIRICAL CASE

In this section we show that the empirical gradient also obeys a smoothness bound. Concretely we prove the following Lemma.

**Lemma 10 (smoothness of empirical gradient)** *Assume $n \geq d$. Then,*

$$\|\nabla \widehat{\mathcal{L}}(\boldsymbol{W})\|_F \leq \beta \|\boldsymbol{W} - \boldsymbol{W}^*\|_F$$

*with $\beta = 4\sqrt{2}$ holds for all $\boldsymbol{W}$ simultaneously with probability at least $1 - 2e^{-\gamma n}$.*

To prove this lemma note that

$$\|\nabla \widehat{\mathcal{L}}(\boldsymbol{W})\|_F^2 = \left\|\nabla_{\boldsymbol{w}_1} \widehat{\mathcal{L}}\right\|^2 + \left\|\nabla_{\boldsymbol{w}_2} \widehat{\mathcal{L}}\right\|^2$$

To continue note that

$$\left\|\nabla_{\boldsymbol{w}_1} \widehat{\mathcal{L}}\right\| = \sup_{\boldsymbol{u} \in \mathbb{S}^{d-1}} \langle \nabla_{\boldsymbol{w}_1} \widehat{\mathcal{L}}, \boldsymbol{u} \rangle = \sup_{\boldsymbol{u} \in \mathbb{S}^{d-1}} \frac{1}{n} \sum_{i=1}^{n} r(\boldsymbol{x}_i) \phi'(\boldsymbol{x}_i^T \boldsymbol{w}_1)(\boldsymbol{x}_i^T \boldsymbol{u})$$

Now applying Cauchy-Schwarz we conclude that for $n \geq d$ with probability at least $1 - e^{-\gamma d}$ we have

$$\left\|\nabla_{\boldsymbol{w}_1} \widehat{\mathcal{L}}\right\| \leq \sqrt{\frac{1}{n} \sum_{i=1}^{n} r^2(\boldsymbol{x}_i)} \sqrt{\sup_{\boldsymbol{u} \in \mathbb{S}^{d-1}} \frac{1}{n} \sum_{i=1}^{n} (\boldsymbol{x}_i^T \boldsymbol{u})^2}$$

$$\leq \sqrt{\frac{1}{n} \sum_{i=1}^{n} r^2(\boldsymbol{x}_i)} \sqrt{\frac{2(d+n)}{n}}$$

$$\leq 2\sqrt{\frac{1}{n} \sum_{i=1}^{n} r^2(\boldsymbol{x}_i)},$$

where in the penultimate step we used the fact that the spectral norm of a Gaussian matrix is bounded by $\sqrt{2(d+n)}$ with probability at least $1 - 2e^{-\gamma d}$ and in the last step we used the fact that $n \geq d$. Using a similar identity for $\left\|\nabla_{\boldsymbol{w}_2} \widehat{\mathcal{L}}\right\|$ we thus conclude that with probability at least $1 - 2e^{-\gamma d}$

$$\|\nabla \widehat{\mathcal{L}}(\boldsymbol{W})\|_F^2 \leq \frac{4}{n} \sum_{i=1}^{n} r^2(\boldsymbol{x}_i) \tag{16}$$

To proceed note that we have

$$\frac{1}{n} \sum_{i=1}^{n} r^2(\boldsymbol{x}_i) \overset{(a)}{\leq} \frac{2}{n} \sum_{i=1}^{n} \left(\phi(\boldsymbol{w}_1^T \boldsymbol{x}_i) - \phi(\boldsymbol{a}^T \boldsymbol{x}_i)\right)^2 + \frac{2}{n} \sum_{i=1}^{n} \left(\phi(\boldsymbol{w}_2^T \boldsymbol{x}_i) - \phi(-\boldsymbol{a}^T \boldsymbol{x}_i)\right)$$

$$\overset{(b)}{\leq} \frac{2}{n} \sum_{i=1}^{n} (\boldsymbol{x}_i^T \boldsymbol{h}_1)^2 + \frac{2}{n} \sum_{i=1}^{n} (\boldsymbol{x}_i^T \boldsymbol{h}_2)^2$$

$$\overset{(c)}{\leq} 8(\|\boldsymbol{h}_1\|^2 + \|\boldsymbol{h}_2\|^2)$$

where (a) follows from the simple identity $(a + b)^2 \leq 2(a^2 + b^2)$, (b) from the fact that ReLU is 1-Lipschitz, and (c) from the fact that the spectral norm of a Gaussian matrix is bounded by $\sqrt{2(d+n)}$ with probability at least $1 - 2e^{-\gamma d}$ and $n \geq d$. Plugging the latter into (16) we conclude that for $n \geq d$,

$$\|\nabla \widehat{\mathcal{L}}(\boldsymbol{W})\|_F^2 \leq 32 \|\boldsymbol{W} - \boldsymbol{W}^*\|_F^2$$

holds with probability at least $1 - 2e^{-\gamma d}$ concluding the proof with $\beta = 4\sqrt{2}$.

### E.2.4 PROOF OF LOWER-BOUND LEMMA VIA UNIFORM CONCENTRATION

We prove the following lemma

**Lemma 11** *Assume $\boldsymbol{x}_i$ are distributed i.i.d. $\mathcal{N}(\mathbf{0}, \boldsymbol{I}_d)$. Furthermore, assume*

$$n \geq Cd.$$

*Then assuming $\boldsymbol{a}$ a unit norm vector, for all $\boldsymbol{w}$ obeying $\|\boldsymbol{w} - \boldsymbol{a}\| \leq \epsilon$ we have*

$$\frac{1}{n} \sum_{i=1}^{n} \mathbf{1}_{\{\boldsymbol{a}^T \boldsymbol{x}_i \geq 0\}} \mathbf{1}_{\{\boldsymbol{w}^T \boldsymbol{x}_i \geq 0\}} \left(\boldsymbol{x}_i^T (\boldsymbol{w} - \boldsymbol{a})\right)^2 \geq \frac{1}{100\pi} \|\boldsymbol{w} - \boldsymbol{a}\|^2.$$

*holds with probability at least $1 - 4e^{-\gamma n}$.*

To prove this lemma, for all $\boldsymbol{w}$ obeying $\|\boldsymbol{w} - \boldsymbol{a}\| \leq \epsilon$ we divide the region

$$\mathcal{H} = \{\boldsymbol{h} : \|\boldsymbol{h}\| \leq \epsilon\}$$

into the following two regions

$$\mathcal{H}_1 = \left\{\boldsymbol{h} : \|\boldsymbol{h}\| \leq \epsilon \quad \text{and} \quad \frac{\boldsymbol{a}^T \boldsymbol{h}}{\|\boldsymbol{h}\|} \geq -\rho\right\}$$

and

$$\mathcal{H}_2 = \left\{\boldsymbol{h} : \|\boldsymbol{h}\| \leq \epsilon \quad \text{and} \quad \frac{\boldsymbol{a}^T \boldsymbol{h}}{\|\boldsymbol{h}\|} \leq -\rho\right\}$$

with $\rho = \frac{1}{\sqrt{2}}$. To prove the lemma it suffices to prove

$$\frac{1}{n} \sum_{i=1}^{n} \mathbf{1}_{\{\boldsymbol{a}^T \boldsymbol{x}_i \geq 0\}} \mathbf{1}_{\{\boldsymbol{h}^T \boldsymbol{x}_i + \boldsymbol{a}^T \boldsymbol{x}_i \geq 0\}} \left(\boldsymbol{x}_i^T \boldsymbol{h}\right)^2 \geq c \|\boldsymbol{h}\|^2.$$

when $\boldsymbol{h}$ belongs to each of the regions $\mathcal{H}_1$ and $\mathcal{H}_2$ separately.

**Case I: $\boldsymbol{h} \in \mathcal{H}_1$:**

In this case first note that when $\boldsymbol{h}^T \boldsymbol{x}_i \geq 0$ and $\boldsymbol{a}^T \boldsymbol{x}_i \geq 0$ we have $\boldsymbol{h}^T \boldsymbol{x}_i + \boldsymbol{a}^T \boldsymbol{x}_i \geq 0$. Thus,

$$\frac{1}{n} \sum_{i=1}^{n} \mathbf{1}_{\{\boldsymbol{a}^T \boldsymbol{x}_i \geq 0\}} \mathbf{1}_{\{\boldsymbol{h}^T \boldsymbol{x}_i + \boldsymbol{a}^T \boldsymbol{x}_i \geq 0\}} \left(\boldsymbol{x}_i^T \boldsymbol{h}\right)^2 \geq \frac{1}{n} \sum_{i=1}^{n} \mathbf{1}_{\{\boldsymbol{a}^T \boldsymbol{x}_i \geq 0\}} \mathbf{1}_{\{\boldsymbol{h}^T \boldsymbol{x}_i \geq 0\}} \left(\boldsymbol{x}_i^T \boldsymbol{h}\right)^2.$$

In this case we will prove that for all $\boldsymbol{u} \in \mathcal{U}_1 := \{\boldsymbol{u} \in \mathbb{S}^{d-1} : \boldsymbol{a}^T \boldsymbol{u} \geq -\rho\}$ we have

$$\frac{1}{n} \sum_{i=1}^{n} \mathbf{1}_{\{\boldsymbol{a}^T \boldsymbol{x}_i \geq 0\}} \phi^2 \left(\boldsymbol{x}_i^T \boldsymbol{u}\right) \geq \frac{1}{10\sqrt{\pi}} \tag{17}$$

holds with probability at least $1 - 2e^{-\gamma n}$. By taking $\boldsymbol{u} = \frac{\boldsymbol{h}}{\|\boldsymbol{h}\|}$ this immediately implies that

$$\frac{1}{n} \sum_{i=1}^{n} \mathbf{1}_{\{\boldsymbol{a}^T \boldsymbol{x}_i \geq 0\}} \mathbf{1}_{\{\boldsymbol{h}^T \boldsymbol{x}_i + \boldsymbol{a}^T \boldsymbol{x}_i \geq 0\}} \left(\boldsymbol{x}_i^T \boldsymbol{h}\right)^2 \geq \frac{1}{n} \sum_{i=1}^{n} \mathbf{1}_{\{\boldsymbol{a}^T \boldsymbol{x}_i \geq 0\}} \mathbf{1}_{\{\boldsymbol{h}^T \boldsymbol{x}_i \geq 0\}} \left(\boldsymbol{x}_i^T \boldsymbol{h}\right)^2 \geq \frac{1}{100\pi} \|\boldsymbol{h}\|^2,$$

holds for all $\boldsymbol{h} \in \mathcal{H}_1$ with the same probability completing the proof of Case I. We thus turn our attention to proving (17). To this aim, first note that using Jensen's inequality we have

$$\frac{1}{n} \sum_{i=1}^{n} \mathbf{1}_{\{\boldsymbol{a}^T \boldsymbol{x}_i \geq 0\}} \phi^2 \left(\boldsymbol{x}_i^T \boldsymbol{u}\right) \geq \left(\frac{1}{n} \sum_{i=1}^{n} \mathbf{1}_{\{\boldsymbol{a}^T \boldsymbol{x}_i \geq 0\}} \phi \left(\boldsymbol{x}_i^T \boldsymbol{u}\right)\right)^2$$

Next, define the stochastic process

$$\mathcal{X}_i(\boldsymbol{u}) := \mathbf{1}_{\{\boldsymbol{a}^T \boldsymbol{x}_i \geq 0\}} \phi \left(\boldsymbol{x}_i^T \boldsymbol{u}\right) - \mathrm{E}_{\boldsymbol{x}_i} \left[\mathbf{1}_{\{\boldsymbol{a}^T \boldsymbol{x}_i \geq 0\}} \phi \left(\boldsymbol{x}_i^T \boldsymbol{u}\right)\right].$$

We next prove that this stochastic process has sub-Gaussian increments. In particular using a well-known centering argument for the $\| \cdot \|_{\Phi_2}$ (denoting the Orlitz or sub-Guassian norm) we have

$$
\begin{aligned}
\|\mathcal{X}_i(\boldsymbol{u}) - \mathcal{X}_i(\boldsymbol{v})\|_{\Psi_2} &\leq 2\|\mathbf{1}_{\{\boldsymbol{a}^T \boldsymbol{x}_i \geq 0\}} \left(\phi\left(\boldsymbol{x}_i^T \boldsymbol{u}\right) - \phi\left(\boldsymbol{x}_i^T \boldsymbol{v}\right)\right)\|_{\Psi_2} \\
&\leq 2\|\left(\phi\left(\boldsymbol{x}_i^T \boldsymbol{u}\right) - \phi\left(\boldsymbol{x}_i^T \boldsymbol{v}\right)\right)\|_{\Psi_2} \\
&\leq 2\|\boldsymbol{x}_i^T (\boldsymbol{u} - \boldsymbol{v})\|_{\Psi_2} \\
&\leq c\,\|\boldsymbol{u} - \boldsymbol{v}\|
\end{aligned}
$$

where in the penultimate step we used the fact that ReLU is 1-Lipschitz. As a result using the fact that weighted sums of sub-Gaussians are also sub-Guassian, the stochastic process

$$
\begin{aligned}
\mathcal{X}(\boldsymbol{u}) :=&\frac{1}{n}\sum_{i=1}^{n}\mathbf{1}_{\{\boldsymbol{a}^T \boldsymbol{x}_i \geq 0\}}\phi\left(\boldsymbol{x}_i^T \boldsymbol{u}\right) - \mathrm{E}_{\boldsymbol{x}}\left[\mathbf{1}_{\{\boldsymbol{a}^T \boldsymbol{x} \geq 0\}}\phi\left(\boldsymbol{x}^T \boldsymbol{u}\right)\right] \\
=&\frac{1}{n}\sum_{i=1}^{n}\mathbf{1}_{\{\boldsymbol{a}^T \boldsymbol{x}_i \geq 0\}}\phi\left(\boldsymbol{x}_i^T \boldsymbol{u}\right) - \frac{1 + \boldsymbol{a}^T \boldsymbol{u}}{\sqrt{8\pi}}
\end{aligned}
$$

also has sub-Guassian increments i.e.

$$
\|\mathcal{X}(\boldsymbol{u}) - \mathcal{X}(\boldsymbol{v})\|_{\Psi_2} \leq \frac{c}{\sqrt{n}}\,\|\boldsymbol{u} - \boldsymbol{v}\|
$$

Thus using Talagrand's majorizing theorem (e.g. see (Vershynin, Exercise 8.6.5)) we have

$$
\sup_{\boldsymbol{u} \in \mathcal{U}_1} |\mathcal{X}(\boldsymbol{u})| \leq c\frac{\sqrt{d}}{\sqrt{n}} + t
$$

holds with probability at least $1 - 2e^{-\gamma t^2 n}$. Thus for all $\boldsymbol{u} \in \mathcal{U}_1$ using $t = \delta/2$ and picking $n$ such that $c\frac{\sqrt{d}}{\sqrt{n}} \leq \frac{\delta}{2}$ we have

$$
\frac{1}{n}\sum_{i=1}^{n}\mathbf{1}_{\{\boldsymbol{a}^T \boldsymbol{x}_i \geq 0\}}\phi\left(\boldsymbol{x}_i^T \boldsymbol{u}\right) \geq \frac{1 - \rho}{\sqrt{8\pi}} - \delta,
$$

holds with probability at least $1 - 2e^{-\gamma \delta^2 n}$ as long as $n \geq C\frac{d}{\delta^2}$. Using

$$
\delta = \frac{5\sqrt{2} - 7}{20\sqrt{\pi}}
$$

concludes the proof of Case I.

**Case II: $\boldsymbol{h} \in \mathcal{H}_2$:**

In this case first note that

$$
\begin{aligned}
&\frac{1}{n}\sum_{i=1}^{n}\mathbf{1}_{\{\boldsymbol{a}^T \boldsymbol{x}_i \geq 0\}}\mathbf{1}_{\{\boldsymbol{h}^T \boldsymbol{x}_i + \boldsymbol{a}^T \boldsymbol{x}_i \geq 0\}}\left(\boldsymbol{x}_i^T \boldsymbol{h}\right)^2 \\
&=\frac{1}{n}\sum_{i=1}^{n}\mathbf{1}_{\{\boldsymbol{a}^T \boldsymbol{x}_i \geq 0\}}\mathbf{1}_{\{\boldsymbol{h}^T \boldsymbol{x}_i \geq -\boldsymbol{a}^T \boldsymbol{x}_i\}}\left(\boldsymbol{x}_i^T \boldsymbol{h}\right)^2 \\
&=\left(\frac{1}{n}\sum_{i=1}^{n}\mathbf{1}_{\{\boldsymbol{a}^T \boldsymbol{x}_i \geq 0\}}\mathbf{1}_{\{\boldsymbol{h}^T \boldsymbol{x}_i \geq -\boldsymbol{a}^T \boldsymbol{x}_i\}}\left(\boldsymbol{x}_i^T \frac{\boldsymbol{h}}{\|\boldsymbol{h}\|}\right)^2\right)\|\boldsymbol{h}\|^2 \\
&\geq \left(\frac{1}{n}\sum_{i=1}^{n}\mathbf{1}_{\{\boldsymbol{a}^T \boldsymbol{x}_i \geq 0\}}\mathbf{1}_{\{\boldsymbol{x}_i^T \frac{\boldsymbol{h}}{\|\boldsymbol{h}\|} \geq -\frac{\boldsymbol{a}^T \boldsymbol{x}_i}{\epsilon}\}}\left(\boldsymbol{x}_i^T \frac{\boldsymbol{h}}{\|\boldsymbol{h}\|}\right)^2\right)\|\boldsymbol{h}\|^2 \\
&\geq \left(\frac{1}{n}\sum_{i=1}^{n}\mathbf{1}_{\{\boldsymbol{a}^T \boldsymbol{x}_i \geq 0\}}\mathbf{1}_{\{0 \geq \boldsymbol{x}_i^T \frac{\boldsymbol{h}}{\|\boldsymbol{h}\|} \geq -\frac{\boldsymbol{a}^T \boldsymbol{x}_i}{\epsilon}\}}\left(\boldsymbol{x}_i^T \frac{\boldsymbol{h}}{\|\boldsymbol{h}\|}\right)^2\right)\|\boldsymbol{h}\|^2
\end{aligned}
$$

where in the penultimate line we used the fact that $\|\boldsymbol{h}\| \le \epsilon$ and in the last line we added the indicator of $\boldsymbol{x}_i^T \boldsymbol{h} \le 0$. Therefore, to complete the proof of this part it suffices to show that

$$\frac{1}{n} \sum_{i=1}^{n} \mathbf{1}_{\{\boldsymbol{a}^T \boldsymbol{x}_i \ge 0\}} \mathbf{1}_{\{0 \ge \boldsymbol{x}_i^T \boldsymbol{u} \ge -\frac{\boldsymbol{a}^T \boldsymbol{x}_i}{\epsilon}\}} \left( \boldsymbol{x}_i^T \boldsymbol{u} \right)^2 \ge \frac{1}{100\pi}$$

holds with high probability for all $\boldsymbol{u} \in \mathcal{U}_2 := \{\boldsymbol{u} \in \mathbb{S}^{d-1} : \boldsymbol{u}^T \boldsymbol{a} \le -\rho\}$. Or equivalently by flipping the sign of $\boldsymbol{u}$

$$\frac{1}{n} \sum_{i=1}^{n} \mathbf{1}_{\{\boldsymbol{a}^T \boldsymbol{x}_i \ge 0\}} \mathbf{1}_{\{0 \le \boldsymbol{x}_i^T \boldsymbol{u} \le \frac{\boldsymbol{a}^T \boldsymbol{x}_i}{\epsilon}\}} \left( \boldsymbol{x}_i^T \boldsymbol{u} \right)^2 \ge \frac{1}{100\pi}$$

holds with high probability for all $\boldsymbol{u} \in \mathcal{U}_2 := \{\boldsymbol{u} \in \mathbb{S}^{d-1} : \boldsymbol{u}^T \boldsymbol{a} \ge \rho\}$.

To do this we once again applying Jensen's inequality

$$\frac{1}{n} \sum_{i=1}^{n} \mathbf{1}_{\{\boldsymbol{a}^T \boldsymbol{x}_i \ge 0\}} \mathbf{1}_{\{0 \le \boldsymbol{x}_i^T \boldsymbol{u} \le \frac{\boldsymbol{a}^T \boldsymbol{x}_i}{\epsilon}\}} \left( \boldsymbol{x}_i^T \boldsymbol{u} \right)^2 \ge \left( \frac{1}{n} \sum_{i=1}^{n} \mathbf{1}_{\{\boldsymbol{a}^T \boldsymbol{x}_i \ge 0\}} \mathbf{1}_{\{0 \le \boldsymbol{x}_i^T \boldsymbol{u} \le \frac{\boldsymbol{a}^T \boldsymbol{x}_i}{\epsilon}\}} \boldsymbol{x}_i^T \boldsymbol{u} \right)^2$$

$$\ge \left( \frac{1}{n} \sum_{i=1}^{n} \mathbf{1}_{\{\boldsymbol{a}^T \boldsymbol{x}_i \ge 0\}} \mathcal{S} \left( \boldsymbol{x}_i^T \boldsymbol{u}, \frac{\boldsymbol{x}_i^T \boldsymbol{a}}{\epsilon} \right) \right)^2$$

where

$$S(v; w) := \begin{cases} 0, & v < 0, \\ v, & 0 \le v \le \dfrac{w}{2}, \\ w - v, & \dfrac{w}{2} \le v \le w, \\ 0, & v \ge w. \end{cases}$$

Next, define the stochastic process

$$\mathcal{X}_i(\boldsymbol{u}) := \mathbf{1}_{\{\boldsymbol{a}^T \boldsymbol{x}_i \ge 0\}} \mathcal{S} \left( \boldsymbol{x}_i^T \boldsymbol{u}, \frac{\boldsymbol{x}_i^T \boldsymbol{a}}{\epsilon} \right) - \mathrm{E}_{\boldsymbol{x}_i} \left[ \mathbf{1}_{\{\boldsymbol{a}^T \boldsymbol{x}_i \ge 0\}} \mathcal{S} \left( \boldsymbol{x}_i^T \boldsymbol{u}, \frac{\boldsymbol{x}_i^T \boldsymbol{a}}{\epsilon} \right) \right].$$

We next prove that this stochastic process has sub-Gaussian increments. In particular using a well-known centering argument for the $\|\cdot\|_{\Phi_2}$ (denoting the Orlitz or sub-Guassian norm) we have

$$\|\mathcal{X}_i(\boldsymbol{u}) - \mathcal{X}_i(\boldsymbol{v})\|_{\Psi_2} \le 2\|\mathbf{1}_{\{\boldsymbol{a}^T \boldsymbol{x}_i \ge 0\}} \left( \mathcal{S} \left( \boldsymbol{x}_i^T \boldsymbol{u}, \frac{\boldsymbol{x}_i^T \boldsymbol{a}}{\epsilon} \right) - \mathcal{S} \left( \boldsymbol{x}_i^T \boldsymbol{v}, \frac{\boldsymbol{x}_i^T \boldsymbol{a}}{\epsilon} \right) \right) \|_{\Psi_2}$$

$$\le 2\|\mathcal{S} \left( \boldsymbol{x}_i^T \boldsymbol{u}, \frac{\boldsymbol{x}_i^T \boldsymbol{a}}{\epsilon} \right) - \mathcal{S} \left( \boldsymbol{x}_i^T \boldsymbol{v}, \frac{\boldsymbol{x}_i^T \boldsymbol{a}}{\epsilon} \right) \|_{\Psi_2}$$

$$\le 2\|\boldsymbol{x}_i^T (\boldsymbol{u} - \boldsymbol{v})\|_{\Psi_2}$$

$$\le c \|\boldsymbol{u} - \boldsymbol{v}\|$$

where in the penultimate step we used the fact that $S(v; w)$ is 1-Lipschitz in its first argument. As a result using the fact that weighted sums of sub-Gaussians are also sub-Guassian, the stochastic process

$$\mathcal{X}(\boldsymbol{u}) := \frac{1}{n} \sum_{i=1}^{n} \mathbf{1}_{\{\boldsymbol{a}^T \boldsymbol{x}_i \ge 0\}} \mathcal{S} \left( \boldsymbol{x}_i^T \boldsymbol{u}, \frac{\boldsymbol{x}_i^T \boldsymbol{a}}{\epsilon} \right) - \mathrm{E}_{\boldsymbol{x}} \left[ \mathbf{1}_{\{\boldsymbol{a}^T \boldsymbol{x} \ge 0\}} \mathcal{S} \left( \boldsymbol{x}_i^T \boldsymbol{u}, \frac{\boldsymbol{x}_i^T \boldsymbol{a}}{\epsilon} \right) \right]$$

$$= \frac{1}{n} \sum_{i=1}^{n} \mathbf{1}_{\{\boldsymbol{a}^T \boldsymbol{x}_i \ge 0\}} \mathcal{S} \left( \boldsymbol{x}_i^T \boldsymbol{u}, \frac{\boldsymbol{x}_i^T \boldsymbol{a}}{\epsilon} \right) - f(\boldsymbol{a}^T \boldsymbol{u}, \epsilon)$$

where $f(\boldsymbol{a}^T \boldsymbol{u}, \epsilon) := E_x \left[ \mathcal{S} \left( \boldsymbol{x}_i^T \boldsymbol{u}, \frac{\boldsymbol{x}_i^T \boldsymbol{a}}{\epsilon} \right) \right]$, also has sub-Guassian increments i.e.

$$\|\mathcal{X}(\boldsymbol{u}) - \mathcal{X}(\boldsymbol{v})\|_{\Psi_2} \le \frac{c}{\sqrt{n}} \|\boldsymbol{u} - \boldsymbol{v}\|$$

Thus using Talagrand's majorizing theorem (e.g. see (Vershynin, Exercise 8.6.5)) we have

$$\sup_{\boldsymbol{u} \in \mathcal{U}_2} |\mathcal{X}(\boldsymbol{u})| \leq c\frac{\sqrt{d}}{\sqrt{n}} + t$$

holds with probability at least $1 - 2e^{-\gamma t^2 n}$. Thus for all $\boldsymbol{u} \in \mathcal{U}_2$ using $t = \delta/2$ and picking $n$ such that $c\frac{\sqrt{d}}{\sqrt{n}} \leq \frac{\delta}{2}$ we have

$$\frac{1}{n} \sum_{i=1}^{n} \mathbf{1}_{\{\boldsymbol{a}^T \boldsymbol{x}_i \geq 0\}} \mathcal{S}\left(\boldsymbol{x}_i^T \boldsymbol{u}, \frac{\boldsymbol{x}_i^T \boldsymbol{a}}{\epsilon}\right) \geq f(\boldsymbol{a}^T \boldsymbol{u}, \epsilon) - \delta,$$

holds with probability at least $1 - 2e^{-\gamma \delta^2 n}$ as long as $n \geq C\frac{d}{\delta^2}$. It is easy to check that $f(\boldsymbol{a}^T \boldsymbol{u}, \epsilon)$ is non-decreasing in its first argument so that for $\boldsymbol{u} \in \mathcal{U}_2$ we have $f(\boldsymbol{a}^T \boldsymbol{u}, \epsilon) \geq f(\rho, \epsilon)$. Furthermore, for $\rho = \frac{1}{\sqrt{2}}$ we have

$$f(\rho, \epsilon) \geq \frac{1}{8\sqrt{\pi}}$$

so that we can conclude that

$$\frac{1}{n} \sum_{i=1}^{n} \mathbf{1}_{\{\boldsymbol{a}^T \boldsymbol{x}_i \geq 0\}} \mathcal{S}\left(\boldsymbol{x}_i^T \boldsymbol{u}, \frac{\boldsymbol{x}_i^T \boldsymbol{a}}{\epsilon}\right) \geq \frac{1}{8\sqrt{\pi}} - \delta,$$

holds with probability at least $1 - 2e^{-\gamma \delta^2 n}$ as long as $n \geq C\frac{d}{\delta^2}$. Picking $\delta = \frac{1}{40\sqrt{\pi}}$ concludes the proof of Case II.

# F PROOF OF MAIN THEOREMS

## F.1 PROOF OF THEOREM 1 FOR ONE DIMENSIONAL OUTPUTS

Using the update rule $\boldsymbol{W}^{(\tau+1)} = \boldsymbol{W}^{(\tau)} - \mu \nabla_{\boldsymbol{W}} \mathcal{L}\left(\boldsymbol{W}^{(\tau)}\right)$,

$$
\begin{aligned}
\left\|\boldsymbol{W}^{(\tau+1)} - \boldsymbol{W}^*\right\|_F^2 &= \left\|\boldsymbol{W}^{(\tau)} - \mu \nabla_{\boldsymbol{W}} \mathcal{L}\left(\boldsymbol{W}^{(\tau)}\right) - \boldsymbol{W}^*\right\|_F^2 \\
&= \left\|\boldsymbol{W}^{(\tau)} - \boldsymbol{W}^*\right\|_F^2 - 2\mu \left\langle \boldsymbol{W}^{(\tau)} - \boldsymbol{W}^*, \nabla_{\boldsymbol{W}} \mathcal{L}\left(\boldsymbol{W}^{(\tau)}\right)\right\rangle + \mu^2 \left\|\nabla_{\boldsymbol{W}} \mathcal{L}\left(\boldsymbol{W}^{(\tau)}\right)\right\|_F^2 \\
&\overset{(a)}{\leq} \left\|\boldsymbol{W}^{(\tau)} - \boldsymbol{W}^*\right\|_F^2 - 2\mu\alpha \left\|\boldsymbol{W}^{(\tau)} - \boldsymbol{W}^*\right\|_F^2 + \mu^2 \left\|\nabla_{\boldsymbol{W}} \mathcal{L}\left(\boldsymbol{W}^{(\tau)}\right)\right\|_F^2 \\
&\overset{(b)}{\leq} \left\|\boldsymbol{W}^{(\tau)} - \boldsymbol{W}^*\right\|_F^2 - 2\mu\alpha \left\|\boldsymbol{W}^{(\tau)} - \boldsymbol{W}^*\right\|_F^2 + \mu^2 \beta^2 \left\|\boldsymbol{W}^{(\tau)} - \boldsymbol{W}^*\right\|_F^2 \\
&= \left(1 - 2\mu\alpha + \mu^2 \beta^2\right) \left\|\boldsymbol{W}^{(\tau)} - \boldsymbol{W}^*\right\|_F^2,
\end{aligned}
$$

where (a) follows from Lemma 4, and (b) follows from Lemma 5. Picking $\mu \leq \frac{\alpha}{\beta^2}$ yields

$$\left\|\boldsymbol{W}^{(\tau+1)} - \boldsymbol{W}^*\right\|_F^2 \leq (1 - \mu\alpha) \left\|\boldsymbol{W}^{(\tau)} - \boldsymbol{W}^*\right\|_F^2.$$

Repeating the steps above $\tau$ times completes the proof. The constants in the theorem statement are $c_1 = \frac{\alpha}{\beta^2}$ and $c_2 = \alpha$.

## F.2 PROOF OF THEOREM 6 FOR MULTIDIMENSIONAL OUTPUTS

Our proof for multiple outputs can be derived by repeated application of the single output case. To see this we remind the reader that the loss takes the form

$$\mathcal{L}(\boldsymbol{W}) = \frac{1}{2} \mathbb{E}_{\boldsymbol{x}} \left[\left\|\boldsymbol{V}^T \text{ReLU}(\boldsymbol{W}\boldsymbol{x}) - \boldsymbol{A}\boldsymbol{x}\right\|^2\right].$$

Plugging in the particular pattern for $\boldsymbol{V}$ this is equal to

$$
\begin{aligned}
\mathcal{L}\left(\boldsymbol{W}\right) &= \frac{1}{2}\mathrm{E}_{\boldsymbol{x}}\left[\left\|\widetilde{\boldsymbol{V}}^T\left[\boldsymbol{I}_r \quad -\boldsymbol{I}_r\right]\mathrm{ReLU}(\boldsymbol{W}\boldsymbol{x}) - \boldsymbol{A}\boldsymbol{x}\right\|^2\right] \\
&= \frac{1}{2}\mathrm{E}_{\boldsymbol{x}}\left[\left\|\boldsymbol{R}^T\boldsymbol{\Sigma}\left[\boldsymbol{I}_r \quad -\boldsymbol{I}_r\right]\mathrm{ReLU}(\boldsymbol{W}\boldsymbol{x}) - \boldsymbol{A}\boldsymbol{x}\right\|^2\right] \\
&= \frac{1}{2}\mathrm{E}_{\boldsymbol{x}}\left[\left\|\boldsymbol{\Sigma}\left[\boldsymbol{I}_r \quad -\boldsymbol{I}_r\right]\mathrm{ReLU}(\boldsymbol{W}\boldsymbol{x}) - \boldsymbol{R}\boldsymbol{A}\boldsymbol{x}\right\|^2\right] \\
&= \frac{1}{2}\sum_{\ell=1}^{r}\sigma_\ell^2\mathrm{E}_{\boldsymbol{x}}\left[\left(\mathrm{ReLU}(\boldsymbol{w}_\ell^T\boldsymbol{x}) - \mathrm{ReLU}(\boldsymbol{w}_{\ell+r}^T\boldsymbol{x}) - \widetilde{\boldsymbol{a}}_\ell^T\boldsymbol{x}\right)^2\right]
\end{aligned}
$$

where $\widetilde{\boldsymbol{a}}_\ell$ is the $\ell$th row of $\boldsymbol{\Sigma}^{-1}\boldsymbol{R}\boldsymbol{A}$ and $\sigma_\ell = \Sigma_{\ell\ell}$. Note that the loss decomposes into $r$ optimization problems of the form in the Theorem 1. Thus the result follows from applying Theorem 1 to the summands of the above loss.

### F.3 PROOF OF THEOREM 2 FOR THE EMPIRICAL SETTING

To prove this theorem first we note that after the first iteration using Lemma 7 from Section E.2.1 we have that with high probability.

$$
\|\boldsymbol{W}^{(1)} - \boldsymbol{W}^*\|_F \leq \epsilon\|\boldsymbol{a}\|
$$

We show inductively below that assuming $\|\boldsymbol{W}^{(\tau)} - \boldsymbol{W}^*\|_F \leq \epsilon\|\boldsymbol{a}\|$ the next iteration also obeys this inequality staying in the local neighborhood of the global optima. In this local neighborhood as shown in Sections E.2.2 and E.2.3 we have the correlation inequality

$$
\left\langle\boldsymbol{W} - \boldsymbol{W}^*, \nabla_{\boldsymbol{W}}\widehat{\mathcal{L}}(\boldsymbol{W})\right\rangle \geq \alpha\|\boldsymbol{W} - \boldsymbol{W}^*\|_F^2 \tag{18}
$$

and the smoothness bound

$$
\|\nabla\widehat{\mathcal{L}}(\boldsymbol{W})\|_F \leq \beta\|\boldsymbol{W} - \boldsymbol{W}^*\|_F \tag{19}
$$

holds with high probability simultaneously for all $\boldsymbol{W}$ obeying $\|\boldsymbol{W}^{(1)} - \boldsymbol{W}^*\|_F \leq \epsilon\|\boldsymbol{a}\|$ with $\alpha$ and $\beta$ fixed numerical constants. We would like to emphasize that the proof (18) in the empirical setting is quite intricate necessitating clever algebraic manipulations combined with sophisticated empirical processing theory tools. With these identities in place using the update rule $\boldsymbol{W}^{(\tau+1)} = \boldsymbol{W}^{(\tau)} - \mu\nabla_{\boldsymbol{W}}\mathcal{L}\left(\boldsymbol{W}^{(\tau)}\right)$, we have

$$
\begin{aligned}
\left\|\boldsymbol{W}^{(\tau+1)} - \boldsymbol{W}^*\right\|_F^2 &= \left\|\boldsymbol{W}^{(\tau)} - \mu\nabla_{\boldsymbol{W}}\widehat{\mathcal{L}}\left(\boldsymbol{W}^{(\tau)}\right) - \boldsymbol{W}^*\right\|_F^2 \\
&= \left\|\boldsymbol{W}^{(\tau)} - \boldsymbol{W}^*\right\|_F^2 - 2\mu\left\langle\boldsymbol{W}^{(\tau)} - \boldsymbol{W}^*, \nabla_{\boldsymbol{W}}\widehat{\mathcal{L}}\left(\boldsymbol{W}^{(\tau)}\right)\right\rangle + \mu^2\left\|\nabla_{\boldsymbol{W}}\widehat{\mathcal{L}}\left(\boldsymbol{W}^{(\tau)}\right)\right\|_F^2 \\
&\overset{(a)}{\leq} \left\|\boldsymbol{W}^{(\tau)} - \boldsymbol{W}^*\right\|_F^2 - 2\mu\alpha\left\|\boldsymbol{W}^{(\tau)} - \boldsymbol{W}^*\right\|_F^2 + \mu^2\left\|\nabla_{\boldsymbol{W}}\widehat{\mathcal{L}}\left(\boldsymbol{W}^{(\tau)}\right)\right\|_F^2 \\
&\overset{(b)}{\leq} \left\|\boldsymbol{W}^{(\tau)} - \boldsymbol{W}^*\right\|_F^2 - 2\mu\alpha\left\|\boldsymbol{W}^{(\tau)} - \boldsymbol{W}^*\right\|_F^2 + \mu^2\beta^2\left\|\boldsymbol{W}^{(\tau)} - \boldsymbol{W}^*\right\|_F^2 \\
&= \left(1 - 2\mu\alpha + \mu^2\beta^2\right)\left\|\boldsymbol{W}^{(\tau)} - \boldsymbol{W}^*\right\|_F^2,
\end{aligned}
$$

where (a) follows from inequality 18, and (b) follows from inequality 19. Picking $\mu \leq \frac{\alpha}{\beta^2}$ yields

$$
\left\|\boldsymbol{W}^{(\tau+1)} - \boldsymbol{W}^*\right\|_F^2 \leq \left(1 - \mu\alpha\right)\left\|\boldsymbol{W}^{(\tau)} - \boldsymbol{W}^*\right\|_F^2.
$$

Repeating the steps above $\tau$ times completes the proof. The constants in the theorem statement are $c_5 = \frac{\alpha}{\beta^2}$ and $c_8 = \alpha$.

## G  ADDITIONAL EXPERIMENTAL RESULTS

### G.1  PAIRING-UP BEHAVIOR FOR $r > 3$

In this section, we present additional results on the pairing behavior of $w_i$ and $v_i$ for different values of $r$.

Beyond the $r = 3$ case shown in Section 4.2, we illustrate the same behavior for $r = 5$ in Figure 7 and for $r = 10$ in Figure 8. While we also observe the pairing for $r > 10$, we omit those results here for visual clarity. In general, we note that the weights at convergence (indicated with *star* symbol in Figures 7 and 8) can be grouped into $r$ pairs such that one of the weights is approximately negative of the other. To aid with detecting the pairs visually, we draw the line determined by each pair with dashed lines.

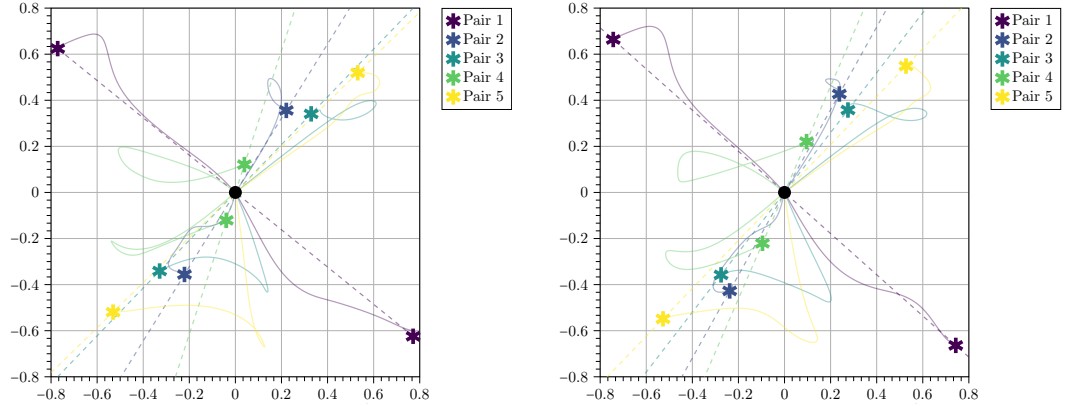

**(a)** Trajectory of $v_i$'s and their pairing behavior. **(b)** Trajectory of $w_i$'s and their pairing behavior.

Figure 7: **Pairing pattern for $r = 5$.** We train the network from small initialization when exactly parameterized ($k = 10$ and $r = 5$). On left (a), we depict the trajectories of individual weights in the outer layer ($v_i$'s) across iterations. Each pair is indicated by the same color and the dashed line. A similar pairing is observed for the inner layer weights as well (b). While these vectors all lie in a higher dimensional space, we pick an arbitrary two dimensional axis to plot them in 2D.

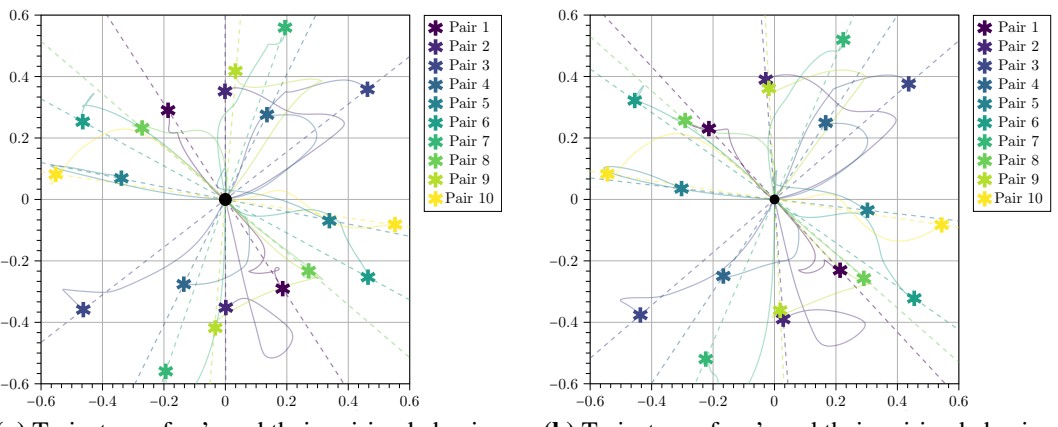

**(a)** Trajectory of $v_i$'s and their pairing behavior. **(b)** Trajectory of $w_i$'s and their pairing behavior.

Figure 8: **Pairing pattern for $r = 10$.** We train the network from small initialization when exactly parameterized ($k = 20$ and $r = 10$). On left (a), we depict the trajectories of individual weights in the outer layer ($v_i$'s) across iterations. Each pair is indicated by the same color and the dashed line. A similar pairing is observed for the inner layer weights as well (b). While these vectors all lie in a higher dimensional space, we pick an arbitrary two dimensional axis to plot them in 2D.

### G.2   LAZY VS. RICH REGIME COMPARISON

In this section, we compare training and generalization dynamics of, so called, "lazy" and "rich" regimes. In our experiments, we fix the learning rate and initialization scale. Therefore, the free parameter that controls the lazy vs. rich learning regime manifests itself through the model width. More specifically, we consider input dimension $d = 100$, fix the learning rate as $\mu = 0.001$ and the initialization scale as $\alpha = 1.0$ (default PyTorch initialization). We train exactly parametrized ($k = 2$), mildly over-parametrized ($k = 6$) and heavily over-parametrized ($k = 100$) models. Using $n = 1000$ data samples, we perform GD iterations until the training loss is below $10^{-4}$. The trajectory of neurons for all cases is given in Figure 9.

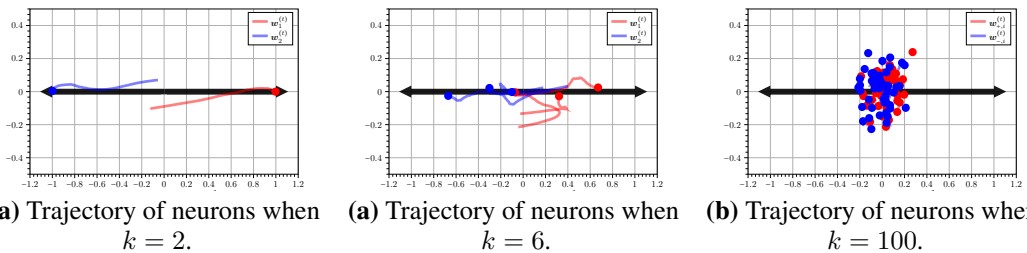

**(a)** Trajectory of neurons when $k = 2$.  **(a)** Trajectory of neurons when $k = 6$.  **(b)** Trajectory of neurons when $k = 100$.

Figure 9: **Trajectory of neurons for different values of** $k$. We run gradient descent updates on the empirical loss after fixing $v$ with half $+1$'s and half $-1$'s. Black arrows indicate $\pm a$ direction. A randomly selected orthogonal direction to $a$ is shown in y-axis in order to visualize the neurons in 2D. We use colors red and blue to indicate whether $v_i$ corresponding to $w_i$ is 1 or $-1$ respectively. Points at the end of each trajectory denotes the final weight GD converges to.

We observe two completely different behaviors. The exactly parametrized model recovers the target directions using $n \approx cd$ data samples as predicted by our theory. Similarly, the mildly over-parametrized model approximately aligns with the target direction. On the other hand, the heavily over-parametrized model has very little neuron movement from its initialization (as predicted by NTK theory). Consequently, the neurons are not necessarily aligned with the corresponding target directions. We measure the generalization performance by calculating the population loss given the final model weights. The exactly parametrized ($k = 2$) and mildly over-parametrized ($k = 6$) model achieves $\approx 2 \times 10^{-4}$ and $\approx 11 \times 10^{-4}$ test loss respectively that is in the same order as the training loss. The Heavily over-parametrized model ($k = 100$) on the other hand, does not generalize, achieving $\approx 4$ test loss. This experiment demonstrates that the existing theory of analyzing heavily over-parametrized models (such as Jacot et al. (2018); Oymak and Soltanolkotabi (2019; 2020); Du et al. (2019); Arora et al. (2019)) cannot be used to explain the generalization phenomenon present in exactly parametrized and mildly over-parametrized models.

## H   LIMITATIONS

While our work provides the first comprehensive analysis of the gradient descent dynamics for learning linear target functions with ReLU networks, it does come with a few limitations. First, our analysis is restricted to shallow networks with a single hidden layer. Extending these results to deeper architectures remains an important direction for future research. Second, our theoretical guarantees rely on the assumption that input data are drawn i.i.d from a Gaussian distribution. Although our techniques can potentially be adapted to broader classes of distributions, the i.i.d assumption is central to the current analysis. That said, we believe this work serves as a stepping stone toward understanding more complex data models, architectures, and training dynamics.

