# OpenReview forum: "On the Dynamics of Learning Linear Functions with Neural Networks"
_ICLR.cc/2026/Conference — Submitted to ICLR 2026_

### Official Review · Reviewer_57Ps · 2025-10-16

**Soundness:** 3
**Presentation:** 3
**Contribution:** 1
**Rating:** 2
**Confidence:** 4

**Summary:**

The paper is concerned with studying gradient descent on one-hidden layer ReLU networks, to learn linear functions.  The inputs are normally distributed, and the loss is mean square.  There are two main theoretical results, in both of which: the output dimension is 1; the width of the network is 2; the output layer is fixed, i.e., only the hidden layer is trained; and convergence to a global optimum with a linear rate is shown.  The two results differ in that the first is for population loss, and the second for empirical loss and also shows that a linear number of samples in the input dimension suffices.  The paper aso reports several experiments in similar or slightly extended settings compared to the two theoretical results.

**Strengths:**

The paper is generally well written.  The statements of the theoretical results and their discussion are clear.  The settings of the experiments are also clearly conveyed, and the plots are not difficult to understand.

**Weaknesses:**

The theoretical results are very restricted.  Learning a linear function by a width-2 one-hidden layer ReLU network does not seem substantially different from learning a single ReLU neuron by a single two-layer ReLU neuron.  In addition, only the hidden layer is trained, which substantially simplifies the dynamics.  And moreover, the gradient descent considered in the paper is non-standard due to the modification of multiplying by the reciprocal squares of the fixed output weights.

Most of the papers discussed as related work are several years old.  Also many relevant works are omitted, such as:
- Yehudai and Shamir "Learning a Single Neuron with Gradient Methods" COLT 2020;
- Vardi, Yehudai, and Shamir "Learning a Single Neuron with Bias Using Gradient Descent" NeurIPS 2021;
- Chistikov, Englert, and Lazic "Learning a Neuron by a Shallow ReLU Network: Dynamics and Implicit Bias for Correlated Inputs" NeurIPS 2023;
- Zhu, Liu, and Cevher "How Gradient descent balances features: A dynamical analysis for two-layer neural networks" ICLR 2025;
- Boursier and Flammarion "Simplicity bias and optimization threshold in two-layer ReLU networks" ICML 2025.

I also remark some typos etc.
- Grammatically correct is "optimum" for singluar and "optima" for plural.
- In the third last line of Theorem 1, $v_1$ and $v_2$ should not be bold.
- In section 4.1, there is a reference to Figure 1 (right), however Figure 1 has parts (a), (b), and (c).

**Questions:**

I do now see how the middle sentence in the paragraph after Theorem 1 shows that "two hidden nodes are necessary".

---

> ### Author Response · Authors · 2025-11-21
> **Response: Part 1**
>
> We sincerely thank the reviewer for their time and efforts in reviewing our manuscript. We are glad that the reviewer finds the paper to be “well written” and the results and discussion to be “clear”. Please find below our responses to your concerns.
>
> - **Re weakness 1 (problem setting)**: Thank you for bringing up this point. As discussed in page 8, learning a linear function can be thought of as a special case of learning “two” ReLU neurons which is a significantly more complicated class of problems than learning a “single” neuron. Due to interaction terms between these neurons (where they can even cancel each other), training dynamics are significantly different from learning the single neuron setting. Therefore, results from single neuron learning cannot be extended here. In fact, the optimization landscape is also substantially different. For learning a single neuron there is effectively one basin of attraction (albeit non-convex) whereas the landscape in our problem contains bad stationary points in the form of non-strict saddles.
>
>   Indeed we set the outer layer to be fixed in this work as it is common with many existing works (including but not limited to [1,2]). Our aim is to eventually handle training of both layers. Analyzing both layers requires dealing with additional challenges and is thus beyond the scope of this paper. We believe this work to be a stepping stone towards this goal.
>
>   Finally, our scaling of GD iterates is akin to Hessian preconditioning, accelerating the convergence of GD significantly as demonstrated in Figure 4 in line with many common practices. We also note that our analysis also applies to regular GD. We focused on this preconditioned variant due to the superior empirical performance as demonstrated in Figure 4. Therefore, we do not think our modification to GD iterations to be non-conventional.

---

> > ### Author Response · Authors · 2025-11-21
> > **Response: Part 2**
> >
> > - **Re weakness 2 (past work)**: We thank the reviewer for pointers to related work.  As explained before, results about learning a single neuron setting (Yehudai et. al., Vardi et. al. , Chistikov et. al.) cannot be used here due to interaction terms between neurons and the significantly more challenging landscape. We incorporated a discussion in our related work section.
> > About the more recent works: Zhu et. al. consider learning multiple \textit{orthogonal} ReLU neurons in a teacher-student framework with outer layer weights fixed to all ones. We note that having orthogonal teacher weights leads to a much more benign landscape as shown in [3] for $k=k^*=2$ case. Moreover, assumptions in the aforementioned work strictly exclude the linear target setting, where the outer layer must contain negative coefficients. Furthermore, they impose strong restrictions on the initialization. Specifically, they look at the convergence after ``weak alignment" where for each student neuron there exists only one teacher neuron that is not near perpendicular. Quoting from the authors:
> >
> >   “Since analyzing the entire training dynamics is still challenging and is an open problem, so we assume the weak recovery, where for each student neuron, exactly one teacher neuron exists that is not nearly perpendicular to it.“
> >
> >   Our population results on the other hand can handle initializations where both student neurons are perpendicular to the target direction as long as we have $||w_1 + w_2|| \leq \frac{1}{2} ||a||$ (e.g. $w_1 = w_2 = \frac{||a||}{4}\bar{a_{\perp}}$). That said, their analysis can handle over-parametrization ($k \gg k^*$) and teacher networks with more than $2$ neurons.
> >
> >   Finally, we were not aware about the recent work of Boursier et. al. where they show despite over-parametrization, the sum of positive (resp. negative) neurons align with the OLS estimator obtained from the ``positive'' (resp. negative) subset of the data. Indeed, this is an interesting result and there are similarities with our work in which we both consider learning linear target functions. However, Boursier et. al. imposes heavy restrictions on the data distribution (in particular Conditions 3 and 4 in page 5). We quote the authors:
> >
> >   “However, item 3 is quite restrictive: it is needed to ensure that the volume of the activation cone containing $\beta*$ does not vanish when $n \rightarrow \infty$ A similar assumption is considered by Chistikov et al. (2023); Tsoy and Konstantinov (2024), for similar reasons. Additionally, Condition 4 ensures that $\mathbb{E}_x[xx^T] \beta*$ and $\beta*$ are in the same activation cone. This assumption allows the training dynamics to remain within a single cone after the early alignment phase, significantly simplifying our analysis.”
> >
> >   In a sense, authors align the data distribution with the target direction to simplify their analysis. Therefore, the authors do not perform a full characterization of GD dynamics with a generic data distribution and initialization without any of the restrictive assumptions mentioned above.
> >
> >   Again, we thank the reviewer for the pointers and we have discussed the differences between these recent work and ours more in detail in the updated manuscript.
> > - **Re typos**: Thank you for catching them. We fix these in the revised manuscript.
> > - **Re question 1**: Intuitively, a single ReLU can only capture “half” of the linearity since it suppresses the signal in a halfspace ($w^T x \leq 0$). By representing the linear function $a^Tx$ as the difference of 2 ReLUs, we aimed to point out the necessity of using at least two student neurons to learn this target function class.
> >
> > We hope that our responses clarify the concerns of the reviewer. If any issues remain, we are happy to discuss it further during the rebuttal period.
> > ***
> > ***References:***
> >
> > [1] Zhong, Kai, et al. "Recovery guarantees for one-hidden-layer neural networks." International conference on machine learning. PMLR, 2017.
> >
> > [2] Zhu, Liu, and Cevher "How Gradient descent balances features: A dynamical analysis for two-layer neural networks" ICLR, 2025.
> >
> > [3] Wu, Chenwei, et al. “No Spurious Local Minima in a Two Hidden Unit ReLU Network”. ICLR Workshop, 2018.

---

### Official Review · Reviewer_pG4V · 2025-10-23

**Soundness:** 3
**Presentation:** 3
**Contribution:** 1
**Rating:** 4
**Confidence:** 4

**Summary:**

This work studies the learning of a linear teacher by a one hidden layer ReLU network, with exact parametrization (ie 2 neurons in the theoretically studied case of univariate output). For this problem, the authors show that both population GD and GD (with $n\gtrsim d$) with empirical data recovers the teacher function.

**Strengths:**

The question of whether a one hidden layer ReLU network can learn a linear teacher remains an open question from a general aspect. The authors provide a nice answer to this question in the case of exact parametrization and no label noise, by giving a nice characterization of the training dynamics.

**Weaknesses:**

My main concern about this work is its relation with the existing literature and how novel it is. In particular, the authors omit to mention the papers [1] and [2] which characterize the training dynamics of a two neurons two layer network when the teacher is itself a two neuron ReLU network. Since a linear function can be seen as the difference of two ReLUs, these two papers clearly fit the scope of learning a linear teacher. In consequence, they deserve to be mentioned, but also to be discussed in detail.

Even if the results of [1] and [2] might not be applied directly to the linear case, I believe that the proof techniques are very similar. In consequence, the authors would also need to discuss to what extent their proof techniques are new and what are the major challenges that were not addressed in previous works.

My other concern is about the limited aspect of the problem. The authors consider strong assumptions that strongly simplify the analysis (and even denature the final message) : no label noise, exact parametrization, fixed second layer and $r=1$. In particular, the no label noise assumption allows a perfect recovery of the teacher possible even in the presence of a finite number of data points, which would clearly not be possible with label noise. In consequence, I find the abstract/introduction particularly overstating.

-----------------
# Minor remarks
- [3] also studied how a two layer ReLU newtork can learn from a linear teacher. Although their exact setting and analysis is somehow different, I think it deserves to be cited.
- Figures 1. (a) and (b) are not very clear. The axes are not labeled
- the multiplication by $\text{diag}(v)^-2$ in the GD step makes me think of the Hessian preconditioning. Is there any relation to it?
- I disagree that this work can be seen as a generalization of Xu and Du (2023) (line 413). Their paper precisely aims at studying the overparameterized case, not the exactly parameterized one
- line 405: you mention local optima, but in the problem here, it seems there is no suboptimal local optima, only saddle points


----------
# References
[1] Zhong, Kai, et al. "Recovery guarantees for one-hidden-layer neural networks." International conference on machine learning. PMLR, 2017.

[2] Zhang, Xiao, et al. "Learning one-hidden-layer relu networks via gradient descent." The 22nd international conference on artificial intelligence and statistics. PMLR, 2019.

[3] Boursier, Etienne, and Nicolas Flammarion. "Simplicity Bias and Optimization Threshold in Two-Layer ReLU Networks." Forty-second International Conference on Machine Learning, 2025.

**Questions:**

How novel are your result and analysis wrt [1] and [2] ?

---

> ### Author Response · Authors · 2025-11-21
> **Response: Part 1**
>
> We thank the reviewer for their time and effort reviewing our work. We are glad that the reviewer thinks our analysis nicely answers “the open problem of learning linear targets” by characterizing the training dynamics. We did our best to address your questions in depth.
>
>  - **Re weakness 1 (relation to existing work) and question 1**: Thank you for the pointers to the related work. We would like to highlight the difference between our work and a few other papers (such as [1,2]) that have planted one-hidden layer models. These papers differ in at least one of three ways focusing on (1) local analysis, (2) have sub-optimal sample complexity, and/or (3) assume non-negative outer layer weights. For instance, [1] utilizes tensor initialization, performing a local analysis rather than a global GD analysis.  This local analysis however can not be used to analyze the linear target setting. Indeed, as noted in Remark 4.3 of their work, their analysis requires $W^*$ to be full-rank which does not hold in the linear setting (where the rows of the weight matrix are negatives of each other leading to a minimum singular value is zero). We quote the authors:
>
>     > If $W^*$ is rank deficient, $\lambda \rightarrow \infty$, $\kappa \rightarrow \infty$ and we don’t have PD property.
>
>     Furthermore, this result also requires resampling the data points at each iteration to ensure convergence of gradient descent whereas we use the same samples across all iterations. On a related note, their sample complexity has polynomial dependency on many problem parameters (Theorem 4.2) whereas our proof only requires sample size linear in input dimension $d$.
>
>     Similarly, [2] provide a local analysis of GD when the outer layer weights are fixed to be all ones. They also utilize results of [1] and share similar limitations in terms of the rank requirement on $W^*$. Thus this result can not be used in the linear target setting even for a local analysis. While they improve the sample complexity of [1] by getting rid of the resampling trick, they still end up with a sample complexity polynomial in width of the network.
>
>     Again, we thank the reviewer for the pointers and we have discussed the differences between these recent work and ours more in detail in the updated manuscript.
>
> - **Re weakness 2 (problem setting)**: We wish to emphasize that analyzing gradient descent dynamics of ReLU networks, even in controlled/simplified settings, is a challenging task. In particular, nearly all papers in the literature explicitly exclude learning linear setting (e.g. Damian et. al. [4]).
>
>   To contextualize our work better, we note that the theoretical analysis of linear autoencoders was already addressed by Baldi and Hornik in 1989. In contrast, understanding the dynamics of non-linear autoencoders has remained an open and challenging problem. Since the autoencoder objective involves learning the identity mapping, it represents a specific case of learning a linear target function. Our work can be seen as taking a step toward this broader goal by first carefully analyzing how gradient descent learns linear functions with ReLU networks with a fixed second layer. We aim to build a foundational understanding that may eventually contribute to the theory of non-linear autoencoders even beyond two layers and linear targets. In this work, to the best of our knowledge, we develop the first comprehensive analysis of gradient descent dynamics of learning linear target functions with ReLU networks using isotropic Gaussian inputs. Furthermore, one can think of learning linear target functions as a special case of learning a “planted” ReLU network (${v^*}^T ReLU(W^*x)$) with two hidden units. As such, we believe our analysis takes us one step further in understanding learning non-linear target functions such as planted ReLU networks. We note that learning a two hidden node target function in the general case is still an open problem and believe our analysis takes an important step towards resolving this challenge.

---

> ### Author Response · Authors · 2025-11-21
> **Response: Part 2**
>
> - **Re minor remark 1**: Thank you for bringing the recent work of Boursier et al. [3] to our attention where they show despite over-parametrization, the sum of positive (resp. negative) neurons align with the OLS estimator obtained from the "positive" (resp. negative) subset of the data. Indeed, this is an interesting result and there are similarities with our work in which we both consider learning linear target functions. However, Boursier et al. imposes heavy restrictions on the data distribution (in particular Conditions 3 and 4 in page 5). We quote the authors:
>
>   "However, item 3 is quite restrictive: it is needed to ensure that the volume of the activation cone containing $\beta*$ does not vanish when $n \rightarrow \infty$. A similar assumption is considered by Chistikov et al. (2023), Tsoy and Konstantinov (2024) for similar reasons. Additionally, Condition 4 ensures that $\mathbb{E}_{x}[xx^T] \beta*$ and $\beta*$ are in the same activation cone. This assumption allows the training dynamics to remain within a single cone after the early alignment phase, significantly simplifying our analysis."
>
>   In a sense, authors align the data distribution with the target direction to simplify their analysis. Therefore, the authors do not perform a full characterization of GD dynamics with a generic data distribution and initialization without any of the restrictive assumptions mentioned above.
>
>   We have discussed the differences between these recent work [1,2,3] and ours more in detail in the updated manuscript.
>
> - **Re minor remark 2**: In the figure, we pick an appropriate 2D subspace to visualize the target direction and the movement of student neurons. We clarify the axes in the revised manuscript.
>
> - **Re minor remark 3**: Great observation. Indeed our scaling is akin to Hessian preconditioning, accelerating the convergence of GD significantly as demonstrated in Figure 4.
>
> - **Re minor remark 4**: We would like to clarify that our work is a generalization of the Xu and Du in the sense that it analyzes a specific configuration of learning two target neurons rather than a single neuron. Note that learning two neuron problems is significantly harder than a single one due to interaction terms (and potential to cancel the effect of each other). However, the reviewer is correct that it is not a strict generalization due to the fact that we do not analyze overparametrization yet. We clarify this in the updated manuscript.
>
> - **Re minor remark 5**: Thank you for catching the typo, we fixed it in the updated version.
>
> We hope we have managed to clarify and address all the points the reviewer has raised with respect to our work. If any issues remain, please let us know. We are happy to discuss further.
>
> ***
> ***References:***
>
> [1] Zhong, Kai, et al. "Recovery guarantees for one-hidden-layer neural networks." International conference on machine learning. PMLR, 2017.
>
> [2] Zhang, Xiao, et al. "Learning one-hidden-layer relu networks via gradient descent." The 22nd international conference on artificial intelligence and statistics. PMLR, 2019.
>
> [3] Boursier, Etienne, and Nicolas Flammarion. "Simplicity Bias and Optimization Threshold in Two-Layer ReLU Networks." Forty-second International Conference on Machine Learning, 2025.
>
> [4] Damian, Alexandru, Jason Lee, and Mahdi Soltanolkotabi. "Neural networks can learn representations with gradient descent." Conference on Learning Theory. PMLR, 2022.

---

> > ### Comment · Reviewer_pG4V · 2025-11-24
> >
> > I have read the authors answer and other reviews. I thank the authors for their answer.
> >
> > The additional discussion of related work makes it now clearer how novel are the results presented here. As a last point, I am still unsure how the proof technique here differs from the one of [1] and [2] (cited in my original review) and I would like some clarification about that.

---

> > > ### Author Response · Authors · 2025-11-25
> > >
> > > We are glad to hear that the additional discussion clarified the concerns regarding the novelty of the results. Below, we further clarify how our proof technique differs from [1] and [2].
> > >
> > > In [1,2], the authors use tensor initialization introduced in [1] to get close to $W^{\star}$, and primarily focus on locally bounding the singular values of the Hessian (i.e., when $W$ is near $W^{\star}$). Unlike [1], [2] derives a uniform convergence bound on the difference between the gradient of the empirical loss and the gradient of the population loss, thereby eliminating the need for the resampling trick (drawing a fresh batch of samples in each iteration). As we have clarified in the previous response to the reviewers, even this local analysis is not directly applicable in our case due to the fact that the planted model would have a zero eigenvalue i.e. $\sigma_{\text{min}}([a;-a])=0$.
> > >
> > > In contrast, our population analysis focuses on a complete trajectory analysis starting from moderately small initialization. This relies on two components:
> > >
> > > 1) the correlation inequality: $<W-W^{\star}, \nabla_{W}\mathcal{L}> \geq \alpha ||W-W^{\star}||^2$,
> > > and
> > > 2) directional gradient smoothness: $||\nabla_{W}\mathcal{L}||_F \leq \beta ||W-W^*||_F$.
> > >
> > > Crucially the correlation inequality cannot hold uniformly over the entire domain due to non-strict saddle points. Instead, we make the key observation that $||w_1+w_2||$ is a non-increasing quantity. This allows us a more refined control of the trajectory enabling us to show that (1) holds as long as $||w_1+w_2|| \leq \frac{1}{2}||a||$. Combining (1) and (2), we conclude that if the initialization satisfies $||w_1+w_2|| \leq \frac{1}{2}||a||$, $w_1,w_2$ converge linearly to $\pm a$.
> > >
> > > For the empirical results, we show that the empirical correlation concentrates around the population correlation and we establish the empirical analogue of the directional smoothness. Importantly, our proof technique does not rely on lower and upper bounding the singular values of the Hessian, as is done in [1,2]. As mentioned earlier this technique would fail even in a local neighborhood.
> > >
> > > We thank the reviewer for their engagement. Please don’t hesitate to reach out if you have any lingering questions.

---

> ### Comment · Reviewer_pG4V · 2025-11-28
>
> I thank the authors for their answer. Overall, they addressed my concerns and I thus raise my score in consequence.
>
> Edit: I will do it when allowed on Openreview again...

---

### Official Review · Reviewer_s1xV · 2025-10-29

**Soundness:** 3
**Presentation:** 3
**Contribution:** 2
**Rating:** 4
**Confidence:** 4

**Summary:**

The paper studies the dynamics of the generalization error resulting from learning a linear function using a one-hidden layer neural network with two neurons and ReLU activation.

**Strengths:**

The paper addresses an important question by studying how the generalization error evolves with the number of steps for two-layer neural networks.

**Weaknesses:**

The setting appears too restrictive. First of all, if the target function is linear, does it even make sense to learn it using a neural network. Second, usually the width of the hidden layer is comparable with the input dimension, while in this case the hidden width is 2. Third, the last layer is not trainable and is just frozen instead. I would suggest adding much more motivation to explain why this setting is interesting.

**Questions:**

1) What is the justification for considering the setting presented in the paper?
2) What are the assumptions regarding c_7 in Theorem 2 and how does it depend on the other parameters from the statement? I find it counterintuitive that the generalization error would converge towards 0 for **any** c_7 without extra quantitative assumptions.
3)  Can one generalize the results to an arbitrary number of neurons and / or more general activation functions?

---

> ### Author Response · Authors · 2025-11-21
>
> Thank you for your time reviewing our work. We are glad that the reviewer thinks our work addresses “an important question”. Please find our responses to your comments and concerns below.
>
> - **Re weakness 1 and question 1 (problem setting)**:  We wish to emphasize that analyzing gradient descent dynamics of ReLU networks, even in controlled/simplified settings, is a challenging task. In particular, nearly all papers in the literature explicitly exclude learning linear setting (e.g. Damian et. al. [1]).  Reviewers pG4V and 57Ps, brought the recent work Boursier et. al. [2] to our attention which also considers linear target functions but imposes heavy restrictions on the data distribution (in particular Conditions 3 and 4 in page 5). While this simplifies the analysis, it fails to characterize the entire GD process. Therefore, the general problem of learning linear target functions remains open.
>
>   To contextualize our work better, we note that the theoretical analysis of linear autoencoders was already addressed by Baldi and Hornik in 1989. In contrast, understanding the dynamics of non-linear autoencoders has remained an open and challenging problem. Since the autoencoder objective involves learning the identity mapping, it represents a specific case of learning a linear target function. Our work can be seen as taking a step toward this broader goal by first carefully analyzing how gradient descent learns linear functions with ReLU networks with a fixed second layer. We aim to build a foundational understanding that may eventually contribute to the theory of non-linear autoencoders even beyond two layers and linear targets. In this work, to the best of our knowledge, we develop the first comprehensive analysis of gradient descent dynamics of learning linear target functions with ReLU networks using isotropic Gaussian inputs. Furthermore, one can think of learning linear target functions as a special case of learning a “planted” ReLU network (${v^*}^T ReLU(W^*x)$) with two hidden units. As such, we believe our analysis takes us one step further in understanding learning non-linear target functions such as planted ReLU networks. We note that learning a two hidden node target function in the general case is still an open problem and believe our analysis takes an important step towards resolving this challenge.
>
> - **Re question 2**:  Thank you for pointing this out. Here $c_7$ is a fixed numerical constant that does not depend on any other parameters.
>
> - **Re question 3**: Great question. Understanding exact parametrization is typically necessary as a stepping stone to understanding overparametrization. For instance, for low-rank matrix recovery the exact parameterization results $r=r^* (\text{true rank})$ predates the overparameterized results by 5 years and in particular the overparameterized results heavily build upon the exact parameterization case (yet requiring significant additional novelty). Nevertheless, in section 4.1, figure 2.b, we visualize an overparametrized scenario (for $r=1$, $k=4$) empirically. We observe that grouping hidden units based on their corresponding signs in $v$ and summing them recovers planted directions $\pm a$ exactly. We observe such phenomena for arbitrary $k>2$. While we have a clear understanding of the behavior of the model empirically, our theoretical analysis cannot capture such a setting yet. In future work, we hope to cover overparametrized settings as well. We anticipate that our analysis would work with other well behaved activation functions (i.e. monotonic activations). We work with ReLU non-linearity as it is the most widely used yet technically most difficult one to analyze due to non-differentiability at 0.
>
> Please don’t hesitate to reach out in the discussion period if there are any lingering issues. We are happy to discuss further.
>
> ***
> ***References:***
>
> [1] Damian, Alexandru, Jason Lee, and Mahdi Soltanolkotabi. "Neural networks can learn representations with gradient descent." Conference on Learning Theory. PMLR, 2022.
>
> [2] Boursier, Etienne, and Nicolas Flammarion. "Simplicity Bias and Optimization Threshold in Two-Layer ReLU Networks." Forty-second International Conference on Machine Learning, 2025.

---

### Official Review · Reviewer_wEhk · 2025-11-02

**Soundness:** 3
**Presentation:** 2
**Contribution:** 2
**Rating:** 4
**Confidence:** 4

**Summary:**

The paper analyzes gradient-descent (GD) training dynamics of one-hidden-layer ReLU networks when the ground-truth mapping is linear. The authors mainly focus on the one dimensional case where the ground truth is a single vector and identify the existence of saddle points in this regime. They prove global convergence of GD for the small initialization and extend the result from population loss to empirical loss. Experiments are conducted to demonstrate the problem’s optimization landscape. They also briefly discuss the multi-dimensional output case in the appendix.

**Strengths:**

1. This paper proves global convergence of two layer ReLU networks for learning linear mappings. It also discusses how GD avoids saddle points along the optimization trajectory.
2. Extensive experiments are conducted on the optimization landscape. The observed pairing behaviour in the multi-dimensional setting might be helpful for understanding feature learning of neural networks.

**Weaknesses:**

The main result considers learning a single linear vector as the teacher, which is overly simple. The single neuron learning problem is extensively studied in previous literature (e.g., see \[1\], \[2\]). Although these results cannot be directly transferred to the linear teacher setting, I believe a widely-used early-alignment technique (e.g., see \[3\], \[4\]) can still be applied. When initialization is small, the gradient (equation (8)) is almost parallel with the ground truth, forcing students to align with the teacher and converge globally. Therefore, the analysis techniques used in the paper seem not very novel to me. That being said, I am happy to change my mind if I missed anything.

References:

1. Yehudai, Gilad and Ohad Shamir. “Learning a Single Neuron with Gradient Methods.” ArXiv abs/2001.05205 (2020): n. Pag.
2. Wu, Chenwei, Jiajun Luo, and Jason D. Lee. "No spurious local minima in a two hidden unit relu network." (2018).
3. Soltanolkotabi, Mahdi. "Learning relus via gradient descent." Advances in neural information processing systems 30 (2017).
4. Brutzkus, Alon, and Amir Globerson. "Globally optimal gradient descent for a convnet with gaussian inputs." International conference on machine learning. PMLR, 2017\.

**Questions:**

The multi-dimensional case in Appendix B seems interesting, but the assumption on initialization is restricted. What is the initialization scheme used in Theorem 6? Can a convergence proof be constructed for the random initialization setting?

---

> ### Author Response · Authors · 2025-11-21
>
> We thank the reviewer for their time and constructive feedback. We are encouraged that the reviewer finds the observed pairing behavior useful for understanding feature learning in neural networks and appreciates the extensive experiments included in the paper. Please see below for our detailed responses.
>
> - **Re weakness 1 (past work)**: While single-neuron learning problems are well-studied, extending the analysis to even two neurons (of which learning linear targets is a special case) is significantly more challenging due to the interaction terms between the neurons. As the reviewer noted, single-neuron results (e.g., [1]) do not directly translate to our setting. Fundamentally the landscape of [1,3,4] is significantly simpler containing only one basin of attraction (albeit nonconvex). On the other hand, Wu et. al. [2], investigate a target function with two perpendicular ReLU neurons and non-negative outer weights. Their landscape consists exclusively of strict saddle points, whereas our problem involves non-strict saddles that must be evaded to reach the global minimum, presenting a distinct and more difficult optimization challenge. Having an analysis for both positive and negative weights is a known challenge in the literature exactly due to the more difficult landscape.
>
>   We thank the reviewer for the pointers and we have discussed the differences between these recent work and ours more in detail in the updated manuscript.
>
> - **Re weakness 2 (early alignment ideas)**: First, we would like to stress that our population analysis requires only “moderately” small initialization. Specifically, provided that $||w_1+w_2|| \leq \frac{1}{2}||a||$, we demonstrate that GD iterates converge linearly to the global optimum. Crucially, this region encompasses configurations where the initial gradient is not yet well-aligned with the target direction (e.g. $a = e_1$ and $w_1 = w_2 = \frac{1}{4} e_2$ where $e_1$ and $e_2$ are from the standard basis).
>
>   In our empirical analysis, we indeed use early alignment (similar to [3,4]) to get close to the global minimizer. However, establishing the empirical counterpart of the correlation inequality (Lemma 4) to prove local convergence is non-trivial.  In fact our novelty in the empirical result comes exactly from the local analysis. We do not resort to resampling new data as is common in the literature which requires rather subtle and novel empirical process techniques. For these reasons, our conclusions have novelty that cannot be obtained solely by using existing proof techniques (e.g. [1,2,3,4]).
> - **Re question 1**: In the multidimensional output setting, we first empirically analyzed the behavior of the weights when running GD from small initialization. We observe an interesting pairing behavior at convergence. Given this behavior, we fix the outer layer $V$ according to the aforementioned pattern ($V = [I; -I] \tilde{V}$) and analyze the GD dynamics when $W$ is randomly initialized. So indeed it does apply to random initialization. Conceptually, this is a generalization of the one dimensional setting. While we have a clear understanding of the behavior of the model empirically, our theoretical analysis cannot handle training the outer layer from random initialization. In future work, we hope to resolve this limitation.
>
> We hope that we have sufficiently addressed your questions. Please let us know in case there are any remaining questions, we are glad to answer them in the discussion period.

---

> ### Comment · Reviewer_wEhk · 2025-11-28
>
> I would like to thank the authors for their detailed response. I agree that the initialization basin in this work extends beyond the scope of previous works. However, as noted in my review and also by other reviewers (pG4V, 57Ps), this paper studies a well-adopted model utilizing similar ideas from a long line of previous analysis, thus limiting the novelty of theoretical results obtained here. Therefore, my evaluation remains the same.

---

> > ### Author Response · Authors · 2025-11-28
> >
> > We thank the reviewer for their continued engagement during the rebuttal process.
> >
> > We would like to kindly request that the reviewer examine our response to the other reviewers (including Reviewer pG4V, who has mentioned that they raise their score), which clarify why the techniques of prior work are not applicable here and the ideas/analysis are quite different. We agree the model is fundamental and thus well established (as discussed in the extended related work section of our updated manuscript); however, the scarcity of theoretical guarantees for such a widely used model highlights the complexity and novelty of our contribution. This is similar to solving a famous conjecture: the fact that the problem is well-known makes the derivation of a proof *more* significant, not *less*.

---

### Official Review · Reviewer_QmsM · 2025-11-10

**Soundness:** 3
**Presentation:** 3
**Contribution:** 2
**Rating:** 4
**Confidence:** 3

**Summary:**

This paper analyzes the gradient descent dynamics of training a single-hidden-layer ReLU network to fit linear target functions under Gaussian inputs. Despite the simplicity of the setup, the optimization landscape includes numerous non-strict saddle points, making convergence uncertain. Under this setting, the authors prove that gradient descent with small random initialization converges with a linear rate to the global minimum in two cases: population and empirical loss. Under the empirical loss, they recover the optimal sample complexity. The theoretical findings are supported by experiments.

**Strengths:**

This paper studies an interesting question of learning a simple function by a complex model that is over-expressive. This scenario arises, for example, in training autoencoders, where the end-to-end scheme should implement the identity transform. In such cases, the natural question is whether the learning algorithm can recover the simple function that we are after.

**Weaknesses:**

Under the setting considered in the paper, there is only one minimum, which corresponds to the underlying (ground truth) linear transform. Many works in the past showed that GD can minimize the loss while training neural networks, e.g., [R1]. So, it is not surprising that it happens here as well.

Moreover, the particular setting studied in the paper does not match the practice. For example:
1. Only the weights of the first layer are optimized.
2. The paper considers a preconditioner that uses the weights of the second layer as normalization.
3. The width of the hidden layer is exactly two

Finally, the two settings considered in the paper, empirical and population loss, don't add much more information and look repetitive to me (even though the proof technique is different, and we also get the sample complexity in the empirical setting).

**References**:
[R1] - Gradient Descent Finds Global Minima of Deep Neural Networks

**Questions:**

1. Could we derive **similar** results from prior work? For example, using the same architecture but with SiLU activation and applying [R1].
2. Can we extend the analysis to fully connected layers with bias?

---

> ### Author Response · Authors · 2025-11-21
> **Response: Part 1**
>
> We thank the reviewer for their time and feedback on our manuscript. We are glad to hear that the reviewer finds the problem setting interesting. Please see below for our responses to the reviewer's concerns.
> - **Re weakness 1 (past work) and question 1**: The analysis in the aforementioned paper [1] primarily addresses heavily overparametrized networks with Lipschitz and smooth activations, which explicitly excludes ReLU. More significantly, their results are restricted to the so-called "lazy regime," where the network width is sufficiently large (in this case polynomially large in the number of data points). Consequently, such analysis is inapplicable to settings with mild overparametrization or exact parametrization. Specifically, in our case the number of hidden neurons is 2 and a practical model would have a mild amount of overparameterization with the number of hidden neurons of the order of a constant not scaling polynomially with the number of samples. As discussed further below we believe that our results are an important first step towards this mild overparameterization setting.
> We emphasize that the analysis in [1] is inapplicable here because it operates in the lazy regime to demonstrate that training error approaches zero. Consequently, that analysis provides no insight into generalization performance (test loss) or the specific features learned by the model (e.g., alignment with the target direction). In contrast, we rigorously prove that the neurons align with the target direction, thereby achieving generalization. We added additional experiments to demonstrate the different generalization behavior of heavy overparametrization and exact parametrization in Appendix G.2. In particular, we show that when $kd \gg n$ even though the training error is small, the neurons are not aligned with the target directions and the corresponding test error is high. This demonstrates that the analysis in [1] is not sufficient to explain the feature learning setting of our paper.
> - **Re weakness 2 (problem setting)**: We wish to emphasize that analyzing gradient descent dynamics of ReLU networks, even in controlled/simplified settings, is a challenging task. In particular, nearly all papers in the literature explicitly exclude learning linear setting (e.g. Damian et. al. [2]).  Reviewers pG4V and 57Ps, brought the recent work Boursier et. al. [3] to our attention which also considers linear target functions but imposes heavy restrictions on the data distribution (in particular Conditions 3 and 4 in page 5). While this simplifies the analysis, it fails to characterize the entire GD process. Therefore, the general problem of learning linear target functions remains open.
>
>   To contextualize our work better, we note that the theoretical analysis of linear autoencoders was already addressed by Baldi and Hornik in 1989. In contrast, understanding the dynamics of non-linear autoencoders has remained an open and challenging problem. Since the autoencoder objective involves learning the identity mapping, it represents a specific case of learning a linear target function. Our work can be seen as taking a step toward this broader goal by first carefully analyzing how gradient descent learns linear functions with ReLU networks with a fixed second layer. We aim to build a foundational understanding that may eventually contribute to the theory of non-linear autoencoders even beyond two layers and linear targets. In this work, to the best of our knowledge, we develop the first comprehensive analysis of gradient descent dynamics of learning linear target functions with ReLU networks using isotropic Gaussian inputs. Furthermore, one can think of learning linear target functions as a special case of learning a “planted” ReLU network (${v^*}^TReLU(W^*x)$) with two hidden units. As such, we believe our analysis takes us one step further in understanding learning non-linear target functions such as planted ReLU networks. We note that learning a two hidden node target function in the general case is still an open problem and believe our analysis takes an important step towards resolving this challenge.
> - **Re weakness 3 (empirical vs population)**: We respectfully maintain that empirical analysis provides distinct and necessary value. Theoretical papers on GD dynamics often omit analysis in the empirical setting due to technical complexities; however, while population analysis provides convergence rates, our empirical results offer a precise characterization of the sample complexity. The proofs while related are also substantially different, requiring a different strategy and very subtle use of empirical process theory to deal with correlations occurring from using the same samples across all iterations.

---

> > ### Author Response · Authors · 2025-11-21
> > **Response: Part 2**
> >
> > - **Re question 2**: This is an excellent point. In this work, our primary objective is to develop novel analytical tools to handle optimization in the presence of bad stationary points. To maintain clarity and focus, we analyzed the simplest case. While outside the scope of this specific paper, bias terms can be addressed in future work by appending a fixed feature of $1$ to the input $x$. Extending these results to train both layers (or deeper networks) presents a significantly harder problem; we view our current results and the associated proof techniques as a necessary stepping stone toward that broader goal.
> >
> > Please feel free to reach out during the discussion phase in case there are any further questions or concerns with respect to our work. We are happy to discuss further.
> >
> > ***
> > ***References:***
> >
> > [1] Du, Simon, et al. "Gradient descent finds global minima of deep neural networks." International conference on machine learning. PMLR, 2019.
> >
> > [2] Damian, Alexandru, Jason Lee, and Mahdi Soltanolkotabi. "Neural networks can learn representations with gradient descent." Conference on Learning Theory. PMLR, 2022.
> >
> > [3] Boursier, Etienne, and Nicolas Flammarion. "Simplicity Bias and Optimization Threshold in Two-Layer ReLU Networks." Forty-second International Conference on Machine Learning, 2025.

---

### Author Response · Authors · 2025-11-21
**Rebuttal by Authors**

We thank all reviewers for spending their time reviewing our paper. We did our best to address every concern that has been raised. We updated the manuscript based on several useful feedback and suggestions. We summarize the changes below.

- We add new experiments (Appendix G.2) to demonstrate lazy vs. rich learning behavior and a discussion on why results on heavy over-parametrization cannot explain the feature learning occurring in exactly parameterized or mildly over-parameterized networks. (Reviewer QmsM)
- We greatly expanded our “related work” section by distinguishing our work from others that were recommended by Reviewers wEhk, pG4V, 57Ps.
- We fix the typos and clarify axes in Figure 1 and 2. (Reviewers pG4V, 57Ps)

We are looking forward to addressing any lingering concerns in the rebuttal period.

---

### Meta-Review · Area_Chair_TKE8 · 2025-12-28

**Summary:**

The paper studies learning a linear teacher with a one-hidden-layer ReLU network under Gaussian inputs and squared loss, where the second layer is fixed and training is performed by gradient descent with a specific preconditioning. There are two main theoretical results, both in the setting of a hidden layer of width two, establishing linear convergence to the teacher for population and empirical loss, with the latter requiring only a linear number of samples in the input dimension. The theoretical analysis is complemented by experiments in mildly extended settings.


Reviews generally praised the clarity of presentation and technical soundness, but raised persistent concerns about scope due to the highly simplified setting (width two, linear target, unique global optimum, frozen output layer, and a preconditioned gradient descent variant). The rebuttal clarified the relationship to prior work and alleviated novelty concerns for pG4V, but wEhk explicitly maintained their original assessment, and 57Ps issued a reject score based on the restrictive nature of the theoretical results, which is central to the paper and unlikely to change after rebuttal. Consequently, it appears unlikely that sufficient score increases would occur for the submission to rise above the acceptance threshold. Overall, while the analysis is careful within a narrow regime and nicely corroborated by the experiments, the theoretical contribution is viewed as limited in generality, and the resulting insights are seen as either closely aligned with existing results on the topic or only weakly indicative of broader training dynamics.

**Reviewer Concerns:**

Addressed:

Improved related-work positioning and clearer distinctions from prior analyses (wEhk/pG4V/57Ps); pG4V’s novelty concern was alleviated and they indicated a score raise after the added clarification of proof strategy.

Added experiments on lazy vs. rich learning (partially addressing QmsM).

Outstanding:

Core scope/generalization concerns: the main theory remains in a highly simplified regime.

Novelty skepticism for some reviewers (notably wEhk).

**Reviewer Scores:**

pG4V: Raised their score; they explicitly stated that the additional clarification resolved their concerns.

QmsM: Possibly a minor positive update, but a substantial change is unlikely; core concerns about the simplicity of the setting largely remain.

wEhk: Unlikely to change their score; they explicitly maintained their original assessment.

s1xV: Unlikely to change their score; concerns about the restrictive setting.

57Ps: No change expected; the reject score is based on the inherently restrictive theoretical scope, which is unlikely to be affected by the rebuttal.

---

### Decision · Program_Chairs · 2026-01-26

Reject